# MMTEB: Massive Multilingual Text Embedding Benchmark

**Kenneth Enevoldsen**[1,†,‡]**, Isaac Chung**[2,‡]**, Imene Kerboua**[3,4,‡]**, Márton Kardos**[1,‡]**,
Ashwin Mathur**[2]**, David Stap**[5]**, Jay Gala**[6]**, Wissam Siblini**[2]**, Dominik Krzemiński**[2]**,
Genta Indra Winata**[2]**, Saba Sturua**[7]**, Saiteja Utpala**[8]**, Mathieu Ciancone**[9]**, Marion Schaeffer**[9]**,
Gabriel Sequeira**[2]**, Diganta Misra**[56,57]**, Shreeya Dhakal**[2]**, Jonathan Rystrøm**[11]**, Roman Solomatin**[12,‡]**,
Ömer Çağatan**[13]**, Akash Kundu**[14,15]**, Martin Bernstorff**[1]**, Shitao Xiao**[16]**, Akshita Sukhlecha**[2]**,
Bhavish Pahwa**[8]**, Rafał Poświata**[17]**, Kranthi Kiran GV**[18]**, Shawon Ashraf**[19]**, Daniel Auras**[19]**,
Björn Plüster**[19]**, Jan Philipp Harries**[19]**, Loïc Magne**[2]**, Isabelle Mohr**[7]**, Mariya Hendriksen**[5]**,
Dawei Zhu**[20]**, Hippolyte Gisserot-Boukhlef**[21,22]**, Tom Aarsen**[23,‡]**, Jan Kostkan**[1]**, Konrad Wojtasik**[24]**,
Taemin Lee**[25]**, Marek Šuppa**[27,28]**, Crystina Zhang**[29]**, Roberta Rocca**[1]**, Mohammed Hamdy**[30]**,
Andrianos Michail**[31]**, John Yang**[32]**, Manuel Faysse**[21,26]**, Aleksei Vatolin**[33]**, Nandan Thakur**[29]**,
Manan Dey**[34]**, Dipam Vasani**[2]**, Pranjal Chitale**[35]**, Simone Tedeschi**[36]**, Nguyen Tai**[37]**,
Artem Snegirev**[38]**, Michael Günther**[7]**, Mengzhou Xia**[39]**, Weijia Shi**[40]**, Xing Han Lù**[10]**, Jordan Clive**[41]**,
Gayatri Krishnakumar**[42]**, Anna Maksimova**[38]**, Silvan Wehrli**[43]**, Maria Tikhonova**[38,44]**,
Henil Panchal**[45]**, Aleksandr Abramov**[38]**, Malte Ostendorff**[46]**, Zheng Liu**[16]**, Simon Clematide**[31]**,
Lester James Miranda**[47]**, Alena Fenogenova**[38]**, Guangyu Song**[48]**, Ruqiya Bin Safi**[49]**, Wen-Ding Li**[50]**,
Alessia Borghini**[36]**, Federico Cassano**[51]**, Hongjin Su**[52]**, Jimmy Lin**[29]**, Howard Yen**[39]**, Lasse Hansen**[1]**,
Sara Hooker**[30]**, Chenghao Xiao**[53,‡]**, Vaibhav Adlakha**[10,54,‡]**, Orion Weller**[55,‡]**, Siva Reddy**[10,54,‡]**,
Niklas Muennighoff**[32,47,58,‡]

[1]Aarhus University, [2]Individual Contributor, [3]Esker, [4]INSA Lyon, LIRIS,
[5]University of Amsterdam, [6]MBZUAI, [7]Jina AI, [8]Microsoft Research,
[9]Wikit, [10]McGill University, [11]University of Oxford, [12]ITMO University,
[13]Koç University, [14]Heritage Institute of Technology, [15]Apart Research, [16]BAAI,
[17]National Information Processing Institute, [18]New York University, [19]Ellamind,
[20]Peking University, [21]CentraleSupélec, [22]Artefact Research Center, [23]Hugging Face,
[24]Wrocław University [25]Korea University, [26]Illuin Technology,
[27]Comenius University Bratislava, [28]Cisco Systems, [29]University of Waterloo,
[30]Cohere For AI, [31]University of Zurich, [32]Stanford University, [33]FRC CSC RAS,
[34]Salesforce, [35]IIT Madras, [36]Sapienza University of Rome,
[37]University of Pennsylvania, [38]SaluteDevices, [39]Princeton University,
[40]University of Washington, [41]Imperial College London, [42]R. V. College of Engineering,
[43]Robert Koch Institute, [44]HSE University, [45]Nirma University, [46]Occiglot,
[47]Allen Institute for AI, [48]Tano Labs, [49]The London Institute of Banking and Finance,
[50]Cornell University, [51]Northeastern University, [52]Hong Kong University
[53]Durham University, [54]ServiceNow Research, [55]Johns Hopkins University,
[56]ELLIS Institute Tübingen [57]MPI-IS Tübingen [58]Contextual AI

‡ Managing Team

†Correspondence: kenneth.enevoldsen@cas.au.dkme

## ABSTRACT

Text embeddings are typically evaluated on a limited set of tasks, which are constrained by language, domain, and task diversity. To address these limitations and provide a more comprehensive evaluation, we introduce the Massive Multilingual Text Embedding Benchmark (MMTEB) – a large-scale, community-driven expansion of MTEB, covering over 500 *quality-controlled* evaluation tasks across 250+ languages. MMTEB includes a diverse set of challenging, novel tasks such as instruction following, long-document retrieval, and code retrieval, representing the largest multilingual collection of evaluation tasks for embedding models to date. Using this collection, we develop several highly multilingual benchmarks, which we use to evaluate a representative set of models. We find that while large language models (LLMs) with billions of parameters can achieve state-of-the-art performance on certain language subsets and task categories, the best-performing publicly available model is multilingual-e5-large-instruct with only 560 million parameters. To facilitate accessibility and reduce computational cost, we introduce a novel downsampling method based on inter-task correlation, ensuring a diverse selection while preserving relative model rankings. Furthermore, we optimize tasks such as retrieval by sampling hard negatives, creating smaller but effective splits. These optimizations allow us to introduce benchmarks that drastically reduce computational demands. For instance, our newly introduced zero-shot English benchmark maintains a similar ranking order as the full-scale version but only requires 2% of the original documents vastly reducing the computational cost.[1]

## 1 INTRODUCTION

Text embeddings are used in many applications, such as semantic search (Reimers & Gurevych, 2019; Muennighoff, 2022; Hendriksen et al., 2023; Winata et al., 2023a; 2024b) and classification tasks (Wang et al., 2018; 2019). Additionally, text embeddings play a crucial role in retrieval-augmented generation (RAG; Borgeaud et al. 2022; Lewis et al. 2021), and often provide significant gains in performance on low- to mid-resource languages, enabling the incorporation of previously inaccessible information. Despite the wide range of applications, there's a lack of benchmarks that evaluate text embeddings across multiple domains, languages, and tasks. Existing benchmarks tend to focus on specific domains, demarcated by subject (e.g., medical, legal, fiction (Thorne et al., 2018b)), particular tasks (e.g., retrieval (Thakur et al., 2021)), literary type (e.g., fiction, and non-fiction) or form (e.g., spoken and written). Embeddings also tend to focus on a subset of languages (Nørregaard & Derczynski, 2021).

While recent efforts (Thakur et al., 2021; Muennighoff et al., 2023b; Zhang et al., 2022) have aimed to broaden the scope by encompassing more tasks, domains, or languages (Cohan et al., 2020a; Wrzalik & Krechel, 2021), a large gap in language coverage remains. This work bridges this gap by creating a benchmark that includes a much broader range of low- to mid-resource languages, along with broader coverage of domains and task categories. To create such an expansive benchmark, we initiated a large-scale, open collaboration. Contributors include native speakers from diverse linguistic backgrounds, NLP practitioners, academic and industry researchers, and enthusiasts. To ensure high-quality submissions, each dataset required systematic tests, detailed metadata, and a review.

The result of this extensive collaborative effort is MMTEB, the **M**assive **M**ultilingual **T**ext **E**mbedding **B**enchmark, which comprises more than 500 distinct tasks across 10 task categories, covering over 250 languages, and spans a wide array of domains such as fiction, social media, medical texts, and technical programming documentation. It also integrates recent, high-quality benchmarks that test a model's capabilities in following instructions (Winata et al., 2021; Weller et al., 2024), embedding long documents (Zhu et al., 2024), solving reasoning tasks (Xiao et al., 2024a; Su et al., 2024), and cross-lingual retrieval (Franco-Salvador et al., 2014). For an overview see Figure 1.

---

[1]MMTEB comes with open-source code available at `https://github.com/embeddings-benchmark/mteb` and a public leaderboard available at `https://huggingface.co/spaces/mteb/leaderboard`.

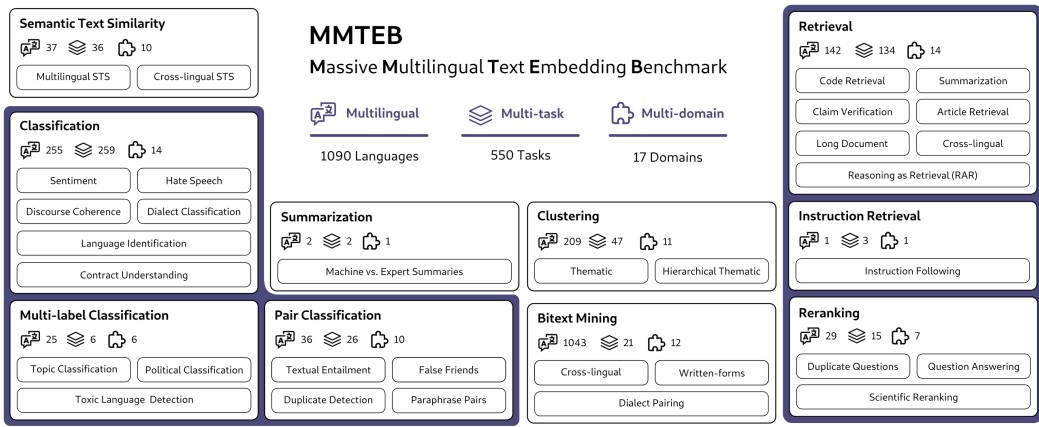

Figure 1: An overview of MMTEB. The boxes represent the overall task categories with a sample of task categories represented within each. Blue borders represent closely-related task categories.

Given the known co-occurrence of limited computational resources and low-resource languages, often referred to as the "low-resource double bind" (Ahia et al., 2021), we made it our goal to make the MMTEB benchmark accessible to low-resource communities. Evaluating models extensively is often resource-intensive. For example, evaluating a single 7B large language model (LLM) on the HELM benchmark consumes over 4,000 GPU hours (Liang et al., 2022). Similarly, the English MTEB (henceforth referred to as MTEB(eng, v1)) benchmark requires up to two days of processing on a single A100 GPU even for moderately sized LLMs (Muennighoff et al., 2023b; BehnamGhader et al., 2024). These high resource demands pose a challenge for low-resource language communities that often lack access to powerful computing resources. MMTEB addresses these challenges by expanding its coverage and optimizing the evaluation process. It significantly reduces computational cost (3.11 hours on an H100 GPU for a 7B model) by using only 2% of the original documents (6% of the original number of characters) while maintaining sensitivity as a benchmark to rank models accurately.

## 2 MMTEB CONSTRUCTION

### 2.1 OPEN SCIENCE EFFORT

To ensure the broad applicability of MMTEB across various domains, we recruited a diverse group of contributors. We actively encouraged participation from industry professionals, low-resource language communities, and academic researchers. To clarify authorship assignment and recognize desired contributions, we implemented a point-based system, similar to Lovenia et al. (2024). To facilitate transparency, coordination was managed through GitHub. A detailed breakdown of contributors and the point system can be found in Appendix A.

### 2.2 ENSURING TASK QUALITY

To guarantee the quality of the added tasks,[2] each task was reviewed by at least one of the main contributors. In addition, we required task submissions to include metadata fields. These fields included details such as annotation source, dataset source, license, dialects, and citation information. Appendix B.4 provides a comprehensive description of each field.

Furthermore, we ensured that the performance on submitted tasks fell within a reasonable range to avoid trivially low or unrealistically high performance. Therefore, we required two multilingual models to be run on the task; multilingual-e5-small (Wang et al., 2022) and MiniLM-L12 (Reimers & Gurevych, 2019). A task was examined further if the models obtained scores close to a random baseline (within a 2% margin), a near-perfect score, or if both models obtained roughly similar scores.

---

[2]A task includes a dataset and an implementation for model evaluation.

These tasks were examined for flawed implementation or poor data quality. Afterwards, a decision was made to either exclude or include the task. We consulted with contributors who are familiar with the target language whenever possible before the final decision. A task could be included despite failing these checks. For example, scores close to the random baseline might be due to the task's inherent difficulty rather than poor data quality.

## 2.3 ACCESSIBILITY AND BENCHMARK OPTIMIZATION

As detailed in Section 1, extensive benchmark evaluations often require significant computational resources. This trend is also observed in MTEB(eng, v1) (Muennighoff et al., 2023b), where running moderately sized LLMs can take up to two days on a single A100 GPU. Accessibility for low-resource communities is particularly important for MMTEB, considering the common co-occurrence of computational constraints (Ahia et al., 2021).

Below, we discuss three main strategies implemented to make our benchmark more efficient. We additionally elaborate further code optimization in Appendix C.2.

### 2.3.1 DOWNSAMPLING AND CACHING EMBEDDINGS

The first strategy involves optimizing the evaluation process by downsampling datasets and caching embeddings. Encoding a large volume of documents for tasks such as retrieval and clustering can be a significant bottleneck in evaluation. Downsampling involves selecting a representative subset of the dataset and reducing the number of documents that require processing. Caching embeddings prevents redundant encoding by using already processed documents.

**Clustering.** In MTEB, clustering is evaluated by computing the v-measure score (Rosenberg & Hirschberg, 2007) on text embeddings clustered using k-means. This process is repeated over multiple distinct sets, inevitably resulting in a large number of documents being encoded. To reduce this encoding burden, we propose a bootstrapping approach that reuses encoded documents across sets. We first encode a 4% subsample of the corpus and sample 10 sets without replacement. Each set undergoes k-means clustering, and we record performance estimates. For certain tasks, this approach reduces the number of documents encoded by $100\times$. In Appendix B.2, we compare both approaches and find an average speedup of 16.11x across tasks, while preserving the relative ranking of models (Average Spearman correlation: 0.96).

**Retrieval.** A key challenge in retrieval tasks is encoding large document collections, which can contain millions of entries Nguyen et al. (2024). To maintain performance comparable to the original datasets while reducing the collection size, we adopted the TREC pooling strategy (Buckley et al., 2007; Soboroff & Robertson, 2003), which aggregates scores from multiple models to select representative documents.[3] For each dataset, we retained the top 250 ranked documents per query, a threshold determined through initial tests that showed negligible differences in absolute scores and no changes in relative rankings across representative models (see Appendix C.1.2 for details on downsampling effects). These documents are merged to form a smaller representative collection. For datasets exceeding 1,000 queries, we randomly sampled 1,000 queries, reducing the largest datasets from over 5 million documents to a maximum of 250,000. This approach accelerated evaluation while preserving ranking performance.

**Bitext Mining.** We apply similar optimization to bitext mining tasks. Some datasets, such as Flores (Costa-jussà et al., 2022) share the same sentences across several language pairs (e.g., English sentences are the same in the English-Hindi pair and the English-Bosnian pair). By caching the embeddings, we reduce the number of embedding computations, making it linear in the number of languages instead of quadratic. For the English documents within Flores this results in a reduction of documents needed to be embedded from 410,000 in MTEB(eng, v1) to just 1,012 in our benchmark.

---

[3]We utilized a range of models: BM25 for lexical hard negatives, e5-multilingual-large as a top-performing BERT-large multilingual model, and e5-Mistral-Instruct 7B, the largest model leveraging instruction-based data.

### 2.3.2 ENCOURAGING SMALLER DATASET SUBMISSIONS

The second strategy focused on encouraging contributors to downsample datasets before submission. To achieve this, we used a stratified split based on target categories. This helped us to ensure that the downsampled datasets could effectively differentiate between candidate models. To validate the process, we compared scores before and after downsampling. For details, we refer to Appendix C.1.

### 2.3.3 TASK SELECTION

To further reduce the computation overhead we seek to construct a task subset that can reliably predict task scores outside the subset.

For task selection, we followed an approach inspired by Xia et al. (2020). We seek to estimate the model $m_i \in M$ scores $s_{t,m_i}$ on an unobserved task $t$ based on scores on observed tasks $s_{j,m_k} \in S, j \neq t$. This allows us to consider the performance of tasks as features within a prediction problem. Thus we can treat task selection as feature reduction, a well-formulated task within machine learning. Note that this formulation allows us to keep the unobserved task arbitrary, representing generalization to unseen tasks (Chollet, 2019). We used a backward selection method, where one task is left out to be predicted, an estimator[4] is fitted on the performance of all models except one, and the score of the held-out model is predicted. This process is repeated until predicted scores are generated for all models on all tasks. The most predictable task is then removed, leaving the estimators in the task subset group. Optionally, we can add additional criteria to ensure task diversity and language representation. Spearman's rank correlation was chosen as the similarity score, as it best preserved the relative ranking when applied to the MTEB(eng, v1).

### 2.4 BENCHMARK CONSTRUCTION

From the extensive collection of tasks in MMTEB, we developed several representative benchmarks, including a highly multilingual benchmark, MTEB(Multilingual), as well as regional geopolitical benchmarks, MTEB(Europe) and MTEB(Indic). Additionally, we introduce a faster version of MTEB(eng, v1) (Muennighoff et al., 2023b), which we refer to as MTEB(eng, v2). MMTEB also integrates domain-specific benchmarks like CoIR for code retrieval (Li et al., 2024) and LongEmbed for long document retrieval (Zhu et al., 2024). MMTEB also introduces language-specific benchmarks, extending the existing suite that includes Scandinavian (Enevoldsen et al., 2024), Chinese (Xiao et al., 2024b), Polish (Poświata et al., 2024), and French (Ciancone et al., 2024). For an overview of the benchmarks, we refer to Appendix H.1.

In the following section, we detail a methodology that we designed to create more targeted and concise benchmarks. This methodology includes: 1) clearly defining the initial scope of the benchmark (**Initial Scope**), 2) reducing the number of tasks by iterative task selection tasks based on intertask correlation (**Refined Scope**), and 3) performing a thorough manual review (**Task Selection and Review**). We provide an overview in Table 1.

In addition to these benchmarks, we provide accompanying code to facilitate the creation of new benchmarks, to allow communities and companies to create tailored benchmarks. In the following, we present MTEB(Multilingual) and MTEB(eng, v2) as two example cases. For a comprehensive overview of benchmark construction and the tasks included in each benchmark, we refer to Appendix H.2.

**MTEB(Multilingual)**: We select all available languages within MMTEB as the initial scope of the benchmark. This results in 550 tasks. We reduce this selection by removing machine-translated datasets, datasets with under-specified licenses, and highly domain-specific datasets such as code-retrieval datasets. This results in 343 tasks covering >250 languages. Following this selection, we evaluate this subset using a representative selection of models (See Section 3.1) and apply task selection to remove the most predictable tasks. To ensure language diversity and representation across task categories, we avoid removing a task that would eliminate a language from the respective task category. Additionally, we did not remove a task if the mean squared error between predicted

---

[4]We use the term "estimator" to differentiate between the evaluated embedding model. For our estimator, we use linear regression.

| Benchmark | Initial Scope | Refined Scope | Task Selection and Review |
|---|---|---|---|
| MTEB(Multilingual) | >500 | 343 | 132 |
| MTEB(Europe) | 420 | 228 | 74 |
| MTEB(Indic) | 55 | 44 | 23 |
| MTEB(eng, v2) | 56 | 54 | 41 |

Table 1: Number of tasks in each benchmark after each filtering step. The initial scope includes tasks relevant to the benchmark goal, notably language of interest. The refined scope further reduced the scope, e.g. removing datasets with underspecified licenses.

and observed scores exceeded 0.5 standard deviations. This is to avoid inadvertantly overindexing to easier tasks. The process of iterative task removal (Section 2.3.3) is repeated until the most predictable held-out task obtained a Spearman correlation of less than 0.8 between predicted and observed scores, or if no tasks were available for filtering. This results in a final selection of 131 diverse tasks. Finally, the selected tasks were reviewed, if possible, by contributors who spoke the target language. If needed, the selection criteria were updated, and some tasks were manually replaced with higher-quality alternatives.

**MTEB(eng, v2)**: Unlike the multilingual benchmarks which target a language group, this benchmark is designed to match MTEB(eng, v1), incorporating computational efficiencies (see Section 2.3) and reducing the intertask correlation using task selection. To prevent overfitting, we intend it as a zero-shot benchmark, excluding tasks like MS MARCO (Nguyen et al., 2016) and Natural Questions (Kwiatkowski et al., 2019), which are frequently used in fine-tuning.

We start the construction by replacing each task with its optimized variant. This updated set obtains a Spearman correlation of 0.97, $p < .0001$ (Pearson 0.99, $p < .0001$) with MTEB(eng, v1) using mean aggregation for the selected models (see Subsection 3.1). The task selection process then proceeds similarly to MTEB(Multilingual), ensuring task diversity by retaining a task if its removal would eliminate a task category. Tasks, where the mean squared error between predicted and observed performance exceeds 0.2 standard deviations, are also retained. This process continues until the most predictable held-out task yields a Spearman correlation below 0.9 between predicted and observed scores. The final selection consists of 41 tasks. We compare this with MTEB(eng, v1) (Muennighoff et al., 2023b) in Section 4.

## 3 EXPERIMENTAL SETTINGS

### 3.1 MODELS

We select a representative set of models, focusing on multilingual models across various size categories. We benchmark the multilingual LaBSE (Feng et al., 2022), trained on paraphrase corpora, English and multilingual versions of MPNet (Song et al., 2020), and MiniLM (Wang et al., 2021b) model, trained on diverse datasets. We also evaluate the multilingual e5 series models (Wang et al., 2024; 2022) trained using a two-step approach utilizing weak supervision. Additionally, to understand the role of scale as well as instruction finetuning, we benchmark GritLM-7B (Muennighoff et al., 2024) and e5-multilingual-7b-instruct (Wang et al., 2023), which are both based on the Mistral 7B model (Jiang et al., 2023).

Revision IDs, model implementation, and prompts used are available in Appendix G. We ran the models on all the implemented tasks to encourage further analysis of the model results. Results, including multiple performance metrics, runtime, $CO_2$ emissions, model metadata, etc., are publicly available in the versioned results repository.[5]

### 3.2 EVALUATION SCORES

For our performance metrics, we report average scores across all tasks, scores per task category, and weighted by task category. We compute model ranks using the Borda count method (Colombo et al.,

---

[5]https://github.com/embeddings-benchmark/results.

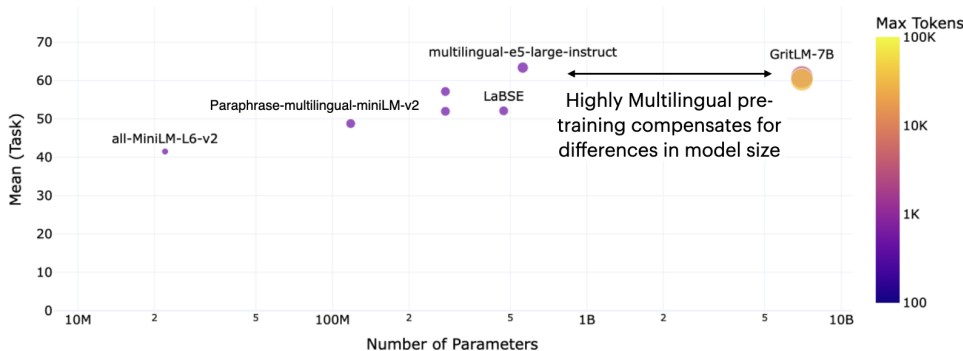

Figure 2: Mean performance across tasks on MTEB(Multilingual) according to the number of parameters. The circle size denotes the embedding size, while the color denotes the maximum sequence length of the model. To improve readability, only certain labels are shown. We refer to the public leaderboard for interactive visualization. We see that the notably smaller model obtains comparable performance to Mistral 7B and GritLM-7B, note that these overlap in the figure due to the similarity of the two models.

2022), derived from social choice theory. This method, which is also employed in election systems based on preference ranking, has been shown to be more robust for comparing NLP systems. To compute this score, we consider each task as a preference voter voting for each model, and scores are aggregated according to the Borda Count method. In the case of ties, we use the tournament Borda count method.

### 3.3 MULTILINGUAL PERFORMANCE

While MMTEB includes multiple benchmarks (see Appendix H.1), we select three multilingual benchmarks to showcase. These constitute a fully multilingual benchmark MTEB(Multilingual) and two targeting languages with varying levels of resources: MTEB(Europe) and MTEB(Indic). The performance of our selected models on these tasks can be seen in Table 2. For performance metrics per task, across domains, etc., we refer to Appendix E.

## 4 ANALYSIS AND DISCUSSION

Table 2 shows the performance across the three presented multilingual benchmarks. Two trends are clearly observable;

**Models trained with instruction-tuning perform significantly better compared to those without it.** This is especially clear when comparing the multilingual-e5-large to its instruction-tuned counterpart (multilingual-e5-large-instruct). Instruction tuning increases performance most drastically on bitext mining and clustering, though the effect remains pronounced across all task categories. Notably, this happens despite many tasks using generic prompts for the task category and no model-specific tuning of prompts per task. Surprisingly, multilingual-e5-large(-instruct) models, based on XLM-R Large (Conneau et al., 2019) generally outperform the considerably larger e5-mistral-7b-instruct and GritLM-7B, both of which are based on Mistral-7B (Jiang et al., 2023). This effect is notably pronounced for mid-to-low resource languages (<300M speaker; see Appendix E.1) and likely emerges due to differences in pre-training, with Mistral being predominantly pre-trained on English, while XLM-R targets 100 languages. All three models utilize similarly multilingual datasets for fine-tuning. However, GritLM still remains best in class for retrieval on MTEB(Multilingual), it has a higher maximum sequence length (see Figure 2) and outperforms the multilingual-e5-large-instruct on MTEB(Code) and MTEB(eng, v2).

**Discrepancies in Multilingual benchmarks ranking seem to stem from discrepancies in pre-training.** While the multilingual benchmarks obtain seemingly similar performance rankings, we see a few notable discrepancies. These discrepancies seem to mainly stem from a narrow multilingual focus (GritLM-7B, e5-mistral-7b-instruct, multilingual-mpnet-base) during training, resulting in disproportionally higher performance on the targeted languages (typically mid-high resource or Euro-

| Model (↓) | Rank (↓) Borda Count | Average Across All | Average Across Category | Average per Category Btxt | Pr Clf | Clf | STS | Rtrvl | M. Clf | Clust | Rrnk |
|---|---|---|---|---|---|---|---|---|---|---|---|
| | | | | MTEB(Multilingual) | | | | | | | |
| Number of datasets (→) | (132) | (132) | (132) | (13) | (11) | (43) | (16) | (18) | (5) | (17) | (6) |
| multilingual-e5-large-instruct | 1 (1375) | **63.2** | **62.1** | **80.1** | 80.9 | **64.9** | **76.8** | 57.1 | **22.9** | **51.5** | 62.6 |
| GritLM-7B | 2 (1258) | 60.9 | 60.1 | 70.5 | 79.9 | 61.8 | 73.3 | **58.3** | 22.8 | 50.5 | **63.8** |
| e5-mistral-7b-instruct | 3 (1233) | 60.3 | 59.9 | 70.6 | 81.1 | 60.3 | 74.0 | 55.8 | 22.2 | 51.4 | **63.8** |
| multilingual-e5-large | 4 (1109) | 58.6 | 58.2 | 71.7 | 79.0 | 59.9 | 73.5 | 54.1 | 21.3 | 42.9 | **62.8** |
| multilingual-e5-base | 5 (944) | 57.0 | 56.5 | 69.4 | 77.2 | 58.2 | 71.4 | 52.7 | 20.2 | 42.7 | 60.2 |
| multilingual-mpnet-base | 6 (830) | 52.0 | 51.1 | 52.1 | **81.2** | 55.1 | 69.7 | 39.8 | 16.4 | 41.1 | 53.4 |
| multilingual-e5-small | 7 (784) | 55.5 | 55.2 | 67.5 | 76.3 | 56.5 | 70.4 | 49.3 | 19.1 | 41.7 | 60.4 |
| LaBSE | 8 (719) | 52.1 | 51.9 | 76.4 | 76.0 | 54.6 | 65.3 | 33.2 | 20.1 | 39.2 | 50.2 |
| multilingual-MiniLM-L12 | 9 (603) | 48.8 | 48.0 | 44.6 | 79.0 | 51.7 | 66.6 | 36.6 | 14.9 | 39.3 | 51.0 |
| all-mpnet-base | 10 (526) | 42.5 | 41.1 | 21.2 | 70.9 | 47.0 | 57.6 | 32.8 | 16.3 | 40.8 | 42.2 |
| all-MiniLM-L12 | 11 (490) | 42.2 | 40.9 | 22.9 | 71.7 | 46.8 | 57.2 | 32.5 | 14.6 | 36.8 | 44.3 |
| all-MiniLM-L6 | 12 (418) | 41.4 | 39.9 | 20.1 | 71.2 | 46.2 | 56.1 | 32.5 | 15.1 | 38.0 | 40.3 |
| | | | | MTEB(Europe) | | | | | | | |
| Number of datasets (→) | (74) | (74) | (74) | (7) | (6) | (21) | (9) | (15) | (2) | (6) | (3) |
| GritLM-7B | 1 (757) | **63.0** | **62.7** | **90.4** | 89.9 | **64.7** | 76.1 | **57.1** | 17.6 | 45.3 | **60.3** |
| multilingual-e5-large-instruct | 2 (732) | 62.2 | 62.3 | 90.4 | 90.0 | 63.2 | **77.4** | 54.8 | 17.3 | **46.9** | 58.4 |
| e5-mistral-7b-instruct | 3 (725) | 61.7 | 61.9 | 89.6 | **91.2** | 62.9 | 76.5 | 53.6 | 15.5 | 46.5 | 59.8 |
| multilingual-e5-large | 4 (586) | 58.5 | 58.7 | 84.5 | 88.8 | 60.4 | 75.8 | 50.8 | 15.0 | 38.2 | 55.9 |
| multilingual-e5-base | 5 (499) | 57.2 | 57.5 | 84.1 | 87.4 | 57.9 | 73.7 | 50.2 | 14.9 | 38.2 | 53.9 |
| multilingual-mpnet-base | 6 (463) | 54.4 | 54.7 | 79.5 | 90.7 | 56.6 | 74.3 | 41.2 | 6.9 | 35.8 | 52.3 |
| multilingual-e5-small | 7 (399) | 55.0 | 55.7 | 80.9 | 86.4 | 56.1 | 71.6 | 46.1 | 14.0 | 36.5 | 54.1 |
| LaBSE | 8 (358) | 51.8 | 53.5 | 88.8 | 85.2 | 55.1 | 65.7 | 34.4 | 16.3 | 34.3 | 48.7 |
| multilingual-MiniLM-L12 | 9 (328) | 51.7 | 52.4 | 77.0 | 88.9 | 52.7 | 72.5 | 37.6 | 5.7 | 34.4 | 50.2 |
| all-mpnet-base | 10 (310) | 44.7 | 44.7 | 29.8 | 80.5 | 49.2 | 63.9 | 37.3 | 10.9 | 36.2 | 49.6 |
| all-MiniLM-L12 | 11 (292) | 44.4 | 44.1 | 32.1 | 81.5 | 49.2 | 64.2 | 36.2 | 7.6 | 32.5 | 49.2 |
| all-MiniLM-L6 | 12 (237) | 43.4 | 43.2 | 27.2 | 80.2 | 47.8 | 62.7 | 37.3 | 8.8 | 33.6 | 47.7 |
| | | | | MTEB(Indic) | | | | | | | |
| Number of datasets (→) | (23) | (23) | (23) | (4) | (1) | (13) | (1) | (2) | (0) | (1) | (1) |
| multilingual-e5-large-instruct | 1 (209) | **70.2** | **71.6** | **80.4** | 76.3 | **67.0** | **53.7** | 84.9 | | **51.7** | **87.5** |
| multilingual-e5-large | 2 (188) | 66.4 | 65.1 | 77.7 | 75.1 | 64.7 | 43.9 | 82.6 | | 25.6 | 86.0 |
| multilingual-e5-base | 3 (173) | 64.6 | 62.6 | 74.2 | 72.8 | 63.8 | 41.1 | 77.8 | | 24.6 | 83.8 |
| multilingual-e5-small | 4 (164) | 64.7 | 63.2 | 73.7 | 73.8 | 63.8 | 40.8 | 76.8 | | 29.1 | 84.4 |
| GritLM-7B | 5 (151) | 60.2 | 58.0 | 58.4 | 67.8 | 60.0 | 27.2 | 79.5 | | 28.0 | 84.7 |
| e5-mistral-7b-instruct | 6 (144) | 60.0 | 58.4 | 59.1 | 73.0 | 59.6 | 23.0 | 77.3 | | 32.7 | 84.4 |
| LaBSE | 7 (139) | 61.9 | 59.7 | 74.1 | 64.6 | 61.9 | 52.8 | 64.3 | | 21.1 | 79.0 |
| multilingual-mpnet-base | 8 (137) | 58.5 | 55.2 | 44.2 | **82.0** | 61.9 | 34.1 | 57.9 | | 32.1 | 74.3 |
| multilingual-MiniLM-L12 | 9 (98) | 49.7 | 42.2 | 15.3 | 77.8 | 57.6 | 19.8 | 48.8 | | 16.7 | 59.3 |
| all-mpnet-base | 10 (68) | 33.6 | 22.6 | 3.7 | 52.6 | 45.2 | -2.5 | 12.9 | | 4.0 | 42.6 |
| all-MiniLM-L12 | 11 (49) | 33.1 | 23.2 | 3.5 | 55.0 | 43.9 | -5.3 | 13.9 | | 3.7 | 47.6 |
| all-MiniLM-L6 | 12 (40) | 31.8 | 20.4 | 2.5 | 53.7 | 44.1 | -6.3 | 6.2 | | 3.1 | 39.2 |

Table 2: The results for three multilingual benchmarks are ranked using Borda count. We provide averages across all tasks, per task category, and weighted by task category. The task categories are shortened as follows: Bitext Mining (Btxt), Pair Classification (Pr Clf), Classification (Clf), Semantic text similarity (STS), Retrieval (Rtrvl), Multilabel Classification (M. Clf), Clustering and Hierarchical Clustering (Clust) and Reranking (Rrnk). We highlight the best score in **bold**. Note that while Instruction retrieval (Weller et al., 2024) is included in MTEB(Europe) and MTEB(Multilingual), but is excluded from the average by task category due to limited model support. For a broader model evaluation, refer to the public leaderboard.

pean ones). These are typically outperformed by the multilingually pre trained XLM-Roberta-based multilingual-e5-large-instruct on lower-resource languages in MTEB(Europe) and all languages in

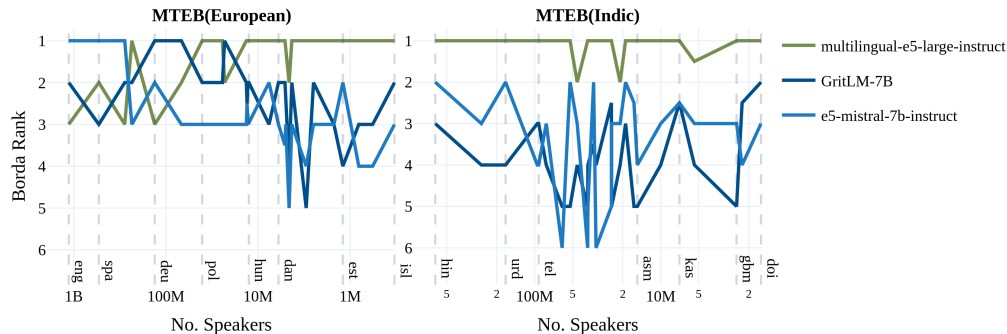

Figure 3: Performance rank of top 3 multilingual models on languages in `MTEB(Europe)` and `MTEB(Indic)` and by the number of native speakers. We see that Mistral-based models are outperformed by multilingual-e5-large-instruct on lower-resource languages, despite it having substantially fewer parameters.

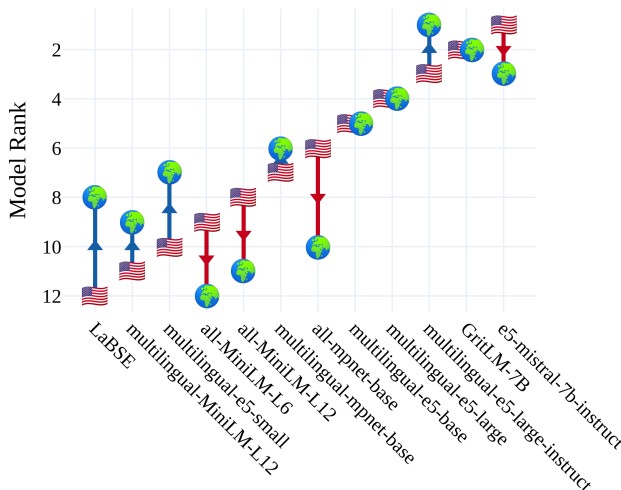

Figure 4: Performance difference on `MTEB(eng, v1)` (flag) and `MTEB(Multilingual)` (globe).

`MTEB(Indic)` (see Figure 3), despite being substantially smaller than Mistral models, the performance of which steadily decreases and becomes more volatile for languages with increasingly lower number of native speakers and this trade-off is well-known (Xue et al., 2020).

Besides these, we observe the expected detrimental performance of English models (all-MiniLM-L12, all-MiniLM-L6, all-mpnet-base) applied to non-English languages and a relatively high bitext performance of LaBSE (see Figure 4).

**MTEB(eng, v1) vs. zero-shot MTEB(eng, v2)** We compare the performance of `MTEB(eng, v1)` and `MTEB(eng, v2)` in Figure 5 obtaining a Spearman correlation of 0.90, $p < 0.0001$ (Pearson 0.96, $p < 0.0001$). For the precise scores, we refer to Subsection H.3. This includes a reduction from 56 to 40 tasks along with optimized task runtime speeding up the runtime on the benchmark (3.11 hours for GritLM-7B and 0.81 hours for all-MiniLM-L12 on an H100). We see that notably, the smaller English models (all-MiniLM-L12, all-MiniLM-L6, all-mpnet-base) perform worse on the new benchmark. This is likely because they were trained on MS MARCO and Natural questions, which were removed as part of the benchmark conversion to a zero-shot benchmark.

## 5 RELATED WORK

**Text Embedding Benchmarks.** BEIR (Thakur et al., 2021) pioneered the use of publicly available datasets from diverse information retrieval (IR) tasks and domains and evaluated 10 various retrieval

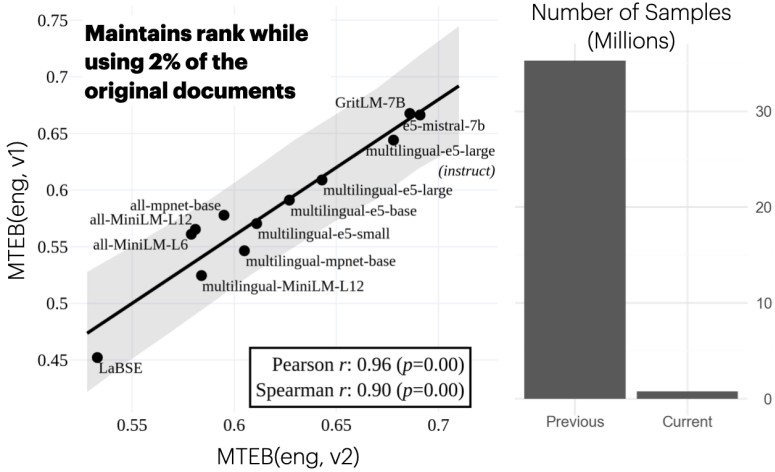

Figure 5: Performance on `MTEB(eng, v1)` and `MTEB(eng, v2)`
.

systems. MTEB (Muennighoff et al., 2023b) introduced a comprehensive text embedding benchmark that spans not only IR but also 8 other task categories, including clustering and re-ranking. MTEB covers a total of 58 tasks and 112 languages, though this multilinguality is mainly derived from machine-translated tasks or bitext mining. Its leaderboard has grown in popularity and evolved into the de facto embedding model benchmark that supports over 300 models. MIRACL (Zhang et al., 2022) supports 18 languages from different language families for monolingual retrieval. MINERS (Winata et al., 2024b) is designed to evaluate the ability of multilingual LMs in semantic retrieval tasks including classification and bitext mining tasks in more than 200 languages, including code-switching. Our work extends the number of languages to over 1000 (250 excluding bitext-mining tasks), particularly to cover more low-resource languages. We also expand the MTEB's 8 embedding tasks to 10 and the 58 datasets to over 400, significantly broadening the scope of multilingual benchmarking.

**Massive Collaborative Projects.** Open research initiatives and participatory approaches to science have been shown to stimulate innovation (Park et al., 2023), reduce negative biases (Gudowsky, 2021; Gomez et al., 2022), and increase the diversity of data sources (Hanley et al., 2020; Singh et al., 2024b; Winata et al., 2024a). By involving diverse stakeholders, these practices enhance ethical, robust, and reproducible research (Hagerty & Rubinov, 2019). Recently, the field of natural language processing has seen a growing number of community-driven collaborative projects. These can be grouped into several categories. *(a) Model creation*, such as BLOOM (BigScience Workshop et al., 2023; Muennighoff et al., 2023c), StarCoder (Li et al., 2023a; Lozhkov et al., 2024), Aya model (Üstün et al., 2024), and Cendol (Cahyawijaya et al., 2024); *(b) Dataset creation*, such as NusaX (Winata et al., 2023b), OpenAssistant (Köpf et al., 2023), NusaWrites (Cahyawijaya et al., 2023c), and Aya dataset (Singh et al., 2024b); *(c) Benchmark creation*, such as BIG-Bench (Srivastava et al., 2023), NusaCrowd (Cahyawijaya et al., 2023a), WorldCuisines (Winata et al., 2024a), HLE (Phan et al., 2025), SEACrowd (Lovenia et al., 2024), and Eval-Harnesses (Gao et al., 2021; Ben Allal et al., 2022; Biderman et al., 2024); and *(d) Other artifacts*, such as NL-Augmenter (Dhole et al., 2021), the Data Provenance Initiative (Longpre et al., 2023; 2024a;b) or the Wikibench annotation tool (Kuo et al., 2024). MMTEB expands upon earlier work within the *Benchmark creation* category. Our effort significantly differs from prior collaborative benchmarks as we focus on text embeddings, use a custom point system to incentivize contributions, and handle all communication openly via GitHub.

## 6 CONCLUSION

This work introduced the Massive Multilingual Text Embedding Benchmark (MMTEB), a large-scale open collaboration resulting in a benchmark with more than 500 tasks covering more than 1000 languages. From these, we constructed three multilingual benchmarks: one fully multi-

lingual (`MTEB(Multilingual)`) and two targeting Indic (`MTEB(Indic)`) and European languages (`MTEB(Europe)`) respectively. Acknowledging that multiple additional benchmarks can be constructed from the MMTEB additions, we propose a simple approach to constructing new benchmarks. To make these benchmarks accessible to low-resource communities, we introduced several optimizations by downsampling retrieval tasks using hard negative mining and bootstrapping clustering evaluation to re-use encoded documents across sets. This leads to a notable reduction in the number of text samples that need to be embedded.

Our findings indicate that while large (7B) LLM-based embedding models obtain state-of-the-art performance on the English benchmark, they are still outperformed in highly multilingual or low-resource settings by smaller models based on XLM-R Large, even when accounting for notable improvements like prompt-based embeddings.

## LIMITATIONS

**English Leakage.** While MMTEB filters out machine-translated datasets, it permits (human) translations. This inclusion leads to tasks like SIB200ClusteringS2S, where labels from English samples are transferred to their translations, potentially introducing bias towards English or models trained on translated content. Consequently, the benchmark may inadvertently encourage model developers to favor English or translated content by increasing their proportion in pre-training data.

**Credit Assignment for Large-scale Collaborations.** One of MMTEB's goals was to highlight the benefits of collaboration. The managing group believes the point system successfully defined contribution terms but acknowledges it isn't perfect. For instance, equal points were awarded for dataset submissions regardless of effort—some datasets were readily available, while others needed significant work like reformulation, HTML parsing, and multiple review rounds.

**Languages Representation.** While the benchmark includes over 250 languages and 500 tasks, the distribution is skewed toward high-resource languages (see Figure 6), with low-resource languages being better represented in specific task categories like bitext-mining and classification. We encourage future collaborations to fill these gaps and enhance language diversity in the collection.

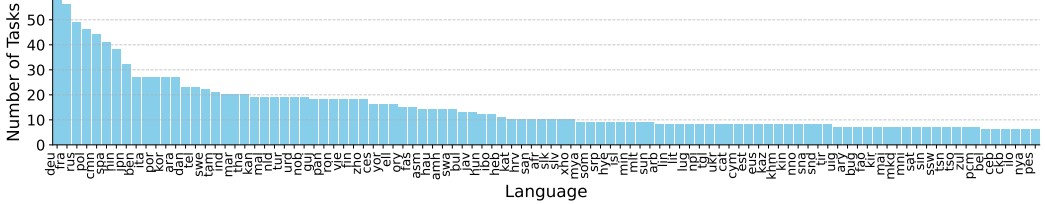

Figure 6: Number of tasks per language. For readability, we remove English (290 tasks) and only plot the 100 languages with the most tasks.

## ETHICAL CONSIDERATIONS

We acknowledge the environmental impact of the benchmark that stems from the compute needed across tasks. As such, emissions tracking is added using codecarbon (Courty et al., 2024) to measure kilograms of $CO_2$-equivalents ($CO_2eq$) and estimate the carbon footprint per task. The benchmark is a collaborative project and contains datasets of different data quality and origin. Thus, additional efforts are still required to identify and minimize biases in the benchmark datasets.

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

APPENDIX TABLE OF CONTENTS

## A  CONTRIBUTIONS

We list the contributions of every author in Table 3. The possible types of contributions and their associated points are:

- **New dataset:** A new dataset includes creating a new implementation (subclass) of a task using a new dataset. 2 points were awarded for implementing the task and 4 points for each new language introduced by the task.

- **New task:** An implementation of a new task category such as multi-label classification or instruction retrieval. 2 points were given for a new task, as well as points following adding a new dataset.

- **Annotations:** Many existing datasets were not yet annotated with proper metadata. To encourage high-quality annotations we awarded 1 point for each full dataset annotation.

- **Fixes:** These included bug fixes, usability fixes, speed improvements and more. For these, we typically awarded 2-10 points depending on the size of the contribution.

- **Running Models:** This includes both running and implementing models for MMTEB. We typically awarded 1 point per model run on a full set of relevant tasks. Relevant tasks for a specific model are limited to those pertinent to its language. For instance, a Russian model does not need to be run on French tasks.

- **Review PR:** A large part of ensuring good dataset quality comes from the dataset review. We award 2 points for a review. If a PR had multiple reviewers, 2 points were awarded to each. Often reviewers finalized dataset additions, helped with data formatting, and resolving bugs. In many cases, adding 2 points for review was considered either too low (a perfect PR with little to no corrections) or too high (lengthy discussion examining dataset quality, debugging implementations and more), however on average we believe it was appropriate.

- **Writing:** At this point many of the authors writing the paper already qualified for co-authorship and thus had reasonable experience with the MMTEB point system. Thus, it was generally possible to discuss a reasonable amount of points based on the efforts made in earlier stages.

- **Coordination:** Included Coordination of contributors and initial ideation were given points at the end of the project based on relative effort. These points were given, similar to paper writing, based on relative effort.

A total of 10 points had to be obtained to be invited as a co-author. To see each contribution mapped to specific PRs, see `https://github.com/embeddings-benchmark/mteb/tree/main/docs/mmteb/points`, where the name of JSON files corresponds to the PR id.

Table 3: Contributions by GitHub users. See Table 4 for the mapping between authors and GitHub handles

| GitHub | Total | Bug fixes | Review PR | New dataset | Dataset annotations | Paper writing | Coordination | New task | Running Models |
|---|---|---|---|---|---|---|---|---|---|
| KennethEnevoldsen | 597 | 87 | 326 | 68 | 35 | 0 | 81 | 0 | 0 |
| isaac-chung | 433 | 50 | 194 | 120 | 1 | 12 | 54 | 2 | 0 |
| imenelydiaker | 358 | 24 | 144 | 120 | 0 | 0 | 70 | 0 | 0 |
| awinml | 302 | 0 | 2 | 300 | 0 | 0 | 0 | 0 | 0 |
| x-tabdeveloping | 239 | 10 | 32 | 144 | 0 | 0 | 41 | 12 | 0 |
| davidstap | 176 | 0 | 0 | 176 | 0 | 0 | 0 | 0 | 0 |
| jaygala24 | 149 | 0 | 0 | 149 | 0 | 0 | 0 | 0 | 0 |
| wissam-sib | 144 | 4 | 6 | 134 | 0 | 0 | 0 | 0 | 0 |
| Muennighoff | 142 | 0 | 48 | 0 | 0 | 0 | 70 | 0 | 24 |
| orionw | 125 | 20 | 20 | 0 | 0 | 0 | 75 | 10 | 0 |
| dokato | 112 | 12 | 6 | 94 | 0 | 0 | 0 | 0 | 0 |
| gentaiscool | 110 | 0 | 0 | 110 | 0 | 0 | 0 | 0 | 0 |
| jupyterjazz | 108 | 0 | 0 | 108 | 0 | 0 | 0 | 0 | 0 |
| SaitejaUtpala | 102 | 0 | 0 | 102 | 0 | 0 | 0 | 0 | 0 |
| vaibhavad | 93 | 8 | 4 | 6 | 0 | 0 | 75 | 0 | 0 |
| MathieuCiancone | 88 | 0 | 0 | 88 | 0 | 0 | 0 | 0 | 0 |
| schmarion | 88 | 0 | 0 | 88 | 0 | 0 | 0 | 0 | 0 |
| GabrielSequeira | 88 | 0 | 0 | 88 | 0 | 0 | 0 | 0 | 0 |
| digantamisra98 | 71 | 0 | 0 | 71 | 0 | 0 | 0 | 0 | 0 |
| shreeya-dhakal | 62 | 0 | 8 | 54 | 0 | 0 | 0 | 0 | 0 |
| Rysias | 58 | 0 | 0 | 58 | 0 | 0 | 0 | 0 | 0 |
| Samoed | 51 | 22 | 2 | 18 | 0 | 0 | 0 | 0 | 9 |
| gowitheflow-1998 | 50 | 0 | 0 | 50 | 0 | 0 | 0 | 0 | 0 |
| sivareddyg | 50 | 0 | 0 | 0 | 0 | 0 | 50 | 0 | 0 |
| asparius | 48 | 0 | 14 | 34 | 0 | 0 | 0 | 0 | 0 |
| Akash190104 | 46 | 0 | 0 | 46 | 0 | 0 | 0 | 0 | 0 |
| MartinBernstorff | 43 | 13 | 8 | 2 | 0 | 0 | 20 | 0 | 0 |
| staoxiao | 40 | 0 | 0 | 40 | 0 | 0 | 0 | 0 | 0 |
| akshita-sukhlecha | 40 | 4 | 0 | 36 | 0 | 0 | 0 | 0 | 0 |
| rafalposwiata | 36 | 0 | 0 | 36 | 0 | 0 | 0 | 0 | 0 |
| bp-high | 36 | 0 | 0 | 36 | 0 | 0 | 0 | 0 | 0 |
| KranthiGV | 34 | 0 | 14 | 20 | 0 | 0 | 0 | 0 | 0 |
| bjoernpl | 28 | 0 | 0 | 28 | 0 | 0 | 0 | 0 | 0 |

Table 3: (Continued) Contributions by GitHub users. See Table 4 for the mapping between authors and GitHub handles

| Github Handle | Total | Bug fixes | Review PR | New dataset | Dataset annotations | Paper writing | Coordination | New task | Running Models |
|---|---|---|---|---|---|---|---|---|---|
| rasdani | 28 | 0 | 0 | 28 | 0 | 0 | 0 | 0 | 0 |
| loicmagne | 28 | 28 | 0 | 0 | 0 | 0 | 0 | 0 | 0 |
| jphme | 28 | 0 | 0 | 28 | 0 | 0 | 0 | 0 | 0 |
| ShawonAshraf | 28 | 0 | 0 | 28 | 0 | 0 | 0 | 0 | 0 |
| violenil | 26 | 0 | 0 | 26 | 0 | 0 | 0 | 0 | 0 |
| mariyahendriksen | 24 | 0 | 0 | 0 | 0 | 24 | 0 | 0 | 0 |
| dwzhu-pku | 24 | 0 | 0 | 24 | 0 | 0 | 0 | 0 | 0 |
| hgissbkh | 23 | 13 | 2 | 0 | 0 | 3 | 0 | 5 | 0 |
| jankounchained | 22 | 8 | 0 | 14 | 0 | 0 | 0 | 0 | 0 |
| taeminlee | 22 | 0 | 0 | 22 | 0 | 0 | 0 | 0 | 0 |
| tomaarsen | 22 | 0 | 2 | 0 | 0 | 0 | 20 | 0 | 0 |
| kwojtasi | 22 | 0 | 0 | 22 | 0 | 0 | 0 | 0 | 0 |
| mrshu | 21 | 0 | 4 | 16 | 1 | 0 | 0 | 0 | 0 |
| crystina-z | 21 | 0 | 0 | 21 | 0 | 0 | 0 | 0 | 0 |
| ManuelFay | 20 | 13 | 0 | 2 | 0 | 0 | 0 | 5 | 0 |
| AlexeyVatolin | 20 | 20 | 0 | 0 | 0 | 0 | 0 | 0 | 0 |
| Andrian0s | 20 | 2 | 4 | 14 | 0 | 0 | 0 | 0 | 0 |
| rbroc | 20 | 0 | 0 | 20 | 0 | 0 | 0 | 0 | 0 |
| john-b-yang | 20 | 0 | 0 | 0 | 0 | 20 | 0 | 0 | 0 |
| mmhamdy | 20 | 0 | 0 | 20 | 0 | 0 | 0 | 0 | 0 |
| manandey | 18 | 0 | 0 | 18 | 0 | 0 | 0 | 0 | 0 |
| thakur-nandan | 18 | 0 | 0 | 18 | 0 | 0 | 0 | 0 | 0 |
| PranjalChitale | 16 | 0 | 0 | 16 | 0 | 0 | 0 | 0 | 0 |
| Sakshamrzt | 16 | 0 | 4 | 12 | 0 | 0 | 0 | 0 | 0 |
| sted97 | 16 | 0 | 0 | 16 | 0 | 0 | 0 | 0 | 0 |
| dipam7 | 16 | 0 | 2 | 14 | 0 | 0 | 0 | 0 | 0 |
| artemsnegirev | 14 | 0 | 0 | 12 | 2 | 0 | 0 | 0 | 0 |
| taidnguyen | 14 | 0 | 0 | 14 | 0 | 0 | 0 | 0 | 0 |
| jordiclive | 12 | 10 | 0 | 2 | 0 | 0 | 0 | 0 | 0 |
| guenthermi | 12 | 0 | 0 | 12 | 0 | 0 | 0 | 0 | 0 |
| slvnwhrl | 12 | 0 | 0 | 12 | 0 | 0 | 0 | 0 | 0 |
| Art3mis07 | 12 | 0 | 0 | 12 | 0 | 0 | 0 | 0 | 0 |
| xhluca | 12 | 4 | 2 | 6 | 0 | 0 | 0 | 0 | 0 |
| anpalmak2003 | 12 | 0 | 0 | 9 | 3 | 0 | 0 | 0 | 0 |
| ab1992ao | 11 | 0 | 0 | 8 | 3 | 0 | 0 | 0 | 0 |
| MariyaTikhonova | 11 | 0 | 0 | 7 | 4 | 0 | 0 | 0 | 0 |
| henilp105 | 11 | 2 | 0 | 0 | 9 | 0 | 0 | 0 | 0 |
| simon-clematide | 10 | 0 | 0 | 10 | 0 | 0 | 0 | 0 | 0 |
| jimmy-lin | 10 | 0 | 0 | 0 | 0 | 0 | 10 | 0 | 0 |
| sarahooker | 10 | 0 | 0 | 0 | 0 | 10 | 0 | 0 | 0 |
| swj0419 | 10 | 0 | 0 | 10 | 0 | 0 | 0 | 0 | 0 |
| xiamengzhou | 10 | 0 | 0 | 10 | 0 | 0 | 0 | 0 | 0 |
| ABorghini | 10 | 0 | 0 | 10 | 0 | 0 | 0 | 0 | 0 |
| xu3kev | 10 | 0 | 0 | 10 | 0 | 0 | 0 | 0 | 0 |
| malteos | 10 | 0 | 0 | 10 | 0 | 0 | 0 | 0 | 0 |
| ljvmiranda921 | 10 | 0 | 0 | 10 | 0 | 0 | 0 | 0 | 0 |
| howard-yen | 10 | 0 | 0 | 10 | 0 | 0 | 0 | 0 | 0 |
| hongjin-su | 10 | 0 | 0 | 10 | 0 | 0 | 0 | 0 | 0 |
| guangyusong | 10 | 0 | 0 | 10 | 0 | 0 | 0 | 0 | 0 |
| Alenush | 10 | 0 | 0 | 6 | 4 | 0 | 0 | 0 | 0 |
| cassanof | 10 | 1 | 0 | 8 | 0 | 0 | 0 | 0 | 1 |
| HLasse | 10 | 5 | 0 | 0 | 5 | 0 | 0 | 0 | 0 |
| ZhengLiu101 | 10 | 0 | 0 | 10 | 0 | 0 | 0 | 0 | 0 |
| Ruqyai | 10 | 0 | 8 | 2 | 0 | 0 | 0 | 0 | 0 |
| izhx | 6 | 0 | 0 | 6 | 0 | 0 | 0 | 0 | 0 |
| marcobellagente93 | 6 | 0 | 0 | 6 | 0 | 0 | 0 | 0 | 0 |
| monikernemo | 2 | 0 | 0 | 2 | 0 | 0 | 0 | 0 | 0 |
| NouamaneTazi | 2 | 0 | 2 | 0 | 0 | 0 | 0 | 0 | 0 |
| MexicanLemonade | 2 | 0 | 0 | 2 | 0 | 0 | 0 | 0 | 0 |
| bakrianoo | 2 | 0 | 0 | 2 | 0 | 0 | 0 | 0 | 0 |
| PhilipMay | 2 | 0 | 2 | 0 | 0 | 0 | 0 | 0 | 0 |
| achibb | 2 | 0 | 0 | 2 | 0 | 0 | 0 | 0 | 0 |
| antoniolanza1996 | 2 | 2 | 0 | 0 | 0 | 0 | 0 | 0 | 0 |
| cslizc | 2 | 0 | 0 | 2 | 0 | 0 | 0 | 0 | 0 |
| hanhainebula | 2 | 0 | 0 | 2 | 0 | 0 | 0 | 0 | 0 |

# B  OVERVIEW AND CONSTRUCTION OF TASKS

In this appendix, we first provide an overview of existing tasks in MTEB benchmark and newly introduced tasks in our benchmark (Section B.1). We proceed by explaining how the tasks were constructed (Section B.2) from existing datasets. Lastly, we introduce newly constructed datasets specifically designed for MMTEB (Section B.3).

| GitHub | First name | Last name | Affiliations |
|---|---|---|---|
| KennethEnevoldsen | Kenneth | Enevoldsen | Aarhus University |
| x-tabdeveloping | Márton | Kardos | Aarhus University |
| imenelydiaker | Imene | Kerboua | INSA Lyon, LIRIS |
| wissam-sib | Wissam | Siblini | Individual Contributor |
| GabrielSequeira | Gabriel | Sequeira | Individual Contributor |
| schmarion | Marion | Schaeffer | Wikit |
| MathieuCiancone | Mathieu | Ciancone | Wikit |
| MartinBernstorff | Martin | Bernstorff | Aarhus University |
| staoxiao | Shitao | Xiao | Beijing Academy of Artificial Intelligence |
| ZhengLiu101 | Zheng | Liu | Beijing Academy of Artificial Intelligence |
| achibb | Aaron | Chibb | Individual Contributor |
| cassanof | Federico | Cassano | Northeastern University and Cursor AI |
| taidnguyen | Nguyen | Tai | University of Pennsylvania |
| xu3kev | Wen-Ding | Li | Cornell University |
| Rysias | Jonathan | Rystrøm | University of Oxford |
| taeminlee | Taemin | Lee | Korea University Human-Inspired AI Research |
| izhx | Xin | Zhang | Harbin Institute of Technology |
| orionw | Orion | Weller | Johns Hopkins University |
| slvnwhrl | Silvan | Wehrli | Robert Koch Institute |
| manandey | Manan | Dey | Salesforce |
| isaac-chung | Isaac | Chung | Individual Contributor |
| asparius | Ömer | Çağatan | Koç University,Turkey |
| rafalposwiata | Rafał | Poświata | National Information Processing Institute |
| rbroc | Roberta | Rocca | Aarhus University |
| awinml | Ashwin | Mathur | Individual Contributor |
| guangyusong | Guangyu | Song | Tano Labs |
| davidstap | David | Stap | University of Amsterdam |
| HLasse | Lasse | Hansen | Aarhus University |
| jaygala24 | Jay | Gala | MBZUAI |
| digantamisra98 | Diganta | Misra | Max Planck Institute for Intelligent Systems and ELLIS Institute Tübingen |
| PranjalChitale | Pranjal | Chitale | Indian Institute of Technology |
| Akash190104 | Akash | Kundu | Heritage Institute of Technology and Apart Research |
| dwzhu-pku | Dawei | Zhu | Peking University |
| ljvmiranda921 | Lester James | Miranda | Allen Institute for AI |
| Andrian0s | Andrianos | Michail | University of Zurich |
| simon-clematide | Simon | Clematide | University of Zurich |
| SaitejaUtpala | Saiteja | Utpala | Microsoft Research |
| mmhamdy | Mohammed | Hamdy | Cohere For AI Community |
| jupyterjazz | Saba | Sturua | Jina AI |
| Ruqyai | Ruqiya | Bin Safi | NaN |
| KranthiGV | Kranthi Kiran | GV | New York University |
| shreeya-dhakal | Shreeya | Dhakal | Individual Contributor |
| dipam7 | Dipam | Vasani | Individual Contributor |
| Art3mis07 | Gayatri | K | R. V. College of Engineering |
| jankounchained | Jan | Kostkan | Aarhus University |
| bp-high | Bhavish | Pahwa | Microsoft Research |
| rasdani | Daniel | Auras | ellamind, Germany |
| ShawonAshraf | Shawon | Ashraf | ellamind, Germany |
| bjoernpl | Björn | Plüster | ellamind, Germany |
| jphme | Jan Philipp | Harries | ellamind, Germany |
| malteos | Malte | Ostendorff | Occiglot |
| ManuelFay | Manuel | Faysse | CentraleSupélec and Illuin Technology |
| hgissbkh | Hippolyte | Gisserot-Boukhlef | CentraleSupélec and Artefact Research Center |
| sted97 | Simone | Tedeschi | Sapienza University of Rome |
| gentaiscool | Genta Indra | Winata | Individual Contributor |
| henilp105 | Henil | Panchal | Nirma University |
| ABorghini | Alessia | Borghini | Sapienza University of Rome |
| jordiclive | Jordan | Clive | Imperial College London |
| gowitheflow-1998 | Chenghao | Xiao | Durham University |
| mariyahendriksen | Mariya | Hendriksen | University of Amsterdam |
| dokato | Dominik | Krzemiński | Cohere For AI Community |
| Samoed | Roman | Solomatin | AI Talent Hub and ITMO University |
| Alenush | Alena | Fenogenova | SaluteDevices |
| ab1992ao | Aleksandr | Abramov | SaluteDevices |
| artemsnegirev | Artem | Snegirev | SaluteDevices |
| anpalmak2003 | Anna | Maksimova | SaluteDevices |
| MariyaTikhonova | Maria | Tikhonova | SaluteDevices and HSE University |
| vaibhavad | Vaibhav | Adlakha | Mila, McGill University and ServiceNow Research |
| sivareddyg | Siva | Reddy | Mila, McGill University and ServiceNow Research |
| guenthermi | Michael | Günther | Jina AI |
| violenil | Isabelle | Mohr | Jina AI |
| akshita-sukhlecha | Akshita | Sukhlecha | Individual Contributor |
| Muennighoff | Niklas | Muennighoff | Stanford University and Contextual AI |
| AlexeyVatolin | Aleksei | Vatolin | FRC CSC RAS |
| xhluca | Xing Han | Lù | Mila, McGill University |
| crystina-z | Xinyu | Zhang | University of Waterloo |
| tomaarsen | Tom | Aarsen | Hugging Face |
| mrshu | Marek | Suppa | Comenius University Bratislava and Cisco Systems |
| swj0419 | Weijia | Shi | University of Washington |
| xiamengzhou | Mengzhou | Xia | Princeton University |
| john-b-yang | John | Yang | Stanford University |
| thakur-nandan | Nandan | Thakur | University of Waterloo |
| loicmagne | Loic | Magne | Individual Contributor |
| sarahooker | Sara | Hooker | Cohere For AI |
| kwojtasi | Konrad | Wojtasik | Wrocław University of Science and Technology |
| jimmy-lin | Jimmy | Lin | University of Waterloo |
| hongjin-su | Hongjin | Su | University of Hong Kong |
| howard-yen | Howard | Yen | Princeton University |

Table 4: Author overview, along with their affiliations and GitHub handles.

## B.1 Introduction to benchmark tasks

**Classification** First, a train set is constructed by sampling $n$ (8-16) samples for each label. If only a test set is available, a section is split off as a training set. Both sets are then embedded and used to train a logistic regression using a maximum of 100 iterations. Afterwards, performance metrics are calculated. For robustness, this process is repeated 10 times.

**Pair classification** For two paired texts, the goal is to predict the label. Examples of such tasks include paraphrase detection or duplicate detection. The task is solved by embedding all documents and then computing the distance either using a model-specified metric, cosine, euclidean, dot product, or Manhattan. Using the best binary threshold, performance metrics are computed.

**Bitext mining** The dataset consists of matching pairs of sentences, and the goal is to find the match. All matching pairs of sentences are embedded, and the closest match is found using cosine similarity, and metrics are reported.

**Clustering and hierarchical clustering** Clustering starts with a set of documents and an associated set of labels. First we embed all documents, then take subsets of the data of size $k$ for each of 10 consecutive experiments. All the documents are embedded, and a set of size $k$ is sampled from the embedded documents. The embeddings are then clustered using K-means clustering, and performance metrics are calculated between the estimated clusters and labels. If the clustering problem is hierarchical, this procedure is repeated for each level of the hierarchy separately. Hierarchical tasks were formerly either split into multiple tasks, or later levels of the cluster hierarchy were ignored.

Note that this formulation differs from that of MTEB in that the sets are randomly sampled from the embedded documents instead of being specified a-priori. This drastically reduced runtime as one document can be used in multiple subsets without the need to embed it multiple times. The new formulation also allows us to gain a robust estimate of performance with a lower number of documents.

**Retrieval** Retrieval tasks consist of a corpus, queries, and mapping between the queries and their relevant documents. The goal is to retrieve these relevant documents. Both queries and documents are embedded using the model. We allow these to be embedded differently depending on the model. For each query, the corpus documents are ranked using a similarity score, and performance metrics are calculated based on the reference mapping.

**Multi-label classification** Classification tasks in MTEB were previously limited to utilizing only one label per document. As such, some, otherwise useful multi-label classification tasks had to be dropped or reformulated. We addressed this by introducing a multi-label classification task type Similarly to our novel clustering task, we down sample training sets for 10 experiments. We limit the training sets to include 8 instances of each unique label, and train a K Nearest-Neighbours classifier. Every classifier is then evaluated on the same test set. We opted for Accuracy, $F_1$ and Label Ranking Average Precision (LRAP) as evaluation metrics.

**Instruction retrieval** Instruction retrieval builds on the traditional retrieval task by incorporating detailed instructions alongside the queries. Unlike standard retrieval, where queries are usually brief keywords, instruction retrieval pairs each query with a comprehensive instruction that outlines the criteria for document relevance. These instructions are specific to each query and not generic to the entire dataset. Therefore, the task involves using both the query and its associated instruction to retrieve relevant documents from the corpus. For the main metric, we use Robustness@10.

**Reranking** Similar to the retrieval task, reranking includes a corpus, query, and a list of relevant and irrelevant reference texts. The aim is to rank the results according to their relevance to the query. References and queries are embedded and references are compared to the query using cosine similarity. The resulting ranking is scored for each query and averaged across all queries, and performance metrics computed. For the main metric, we use MAP@1000.

**Semantic text similarity** Semantic text similarity (STS) tasks consist of sentence pairs, where the goal is to determine their similarity. Labels are continuous scores, with higher numbers indicating more similar sentences. All sentences are embedded using the model, and the similarity of the pair is computed using various distance metrics, allowing for model-specified similarity metrics. Distances

are benchmarked with ground truth similarities using Pearson and Spearman correlations. Spearman correlation based on highest similarity serves as the main metric (Reimers et al., 2016)

## B.2 TASK CONSTRUCTION

This section outlines our approach to constructing tasks, primarily from pre-existing data. For details on the newly introduced dataset in MMTEB, we refer to Section B.3.

Task construction from existing datasets consisted of a number of steps to ensure that the task is compatible with formulations in the benchmark and matches our standards: 1. *Dataset preprocessing*: we start by applying minimal additional processing to ensure the data is in the required format. 2. *Dataset size reduction*: to maintain manageable evaluation times, we proceed by reducing dataset size whenever applicable. 3. *Relevance filtering*: To ensure the datasets are relevant for the types of tasks being evaluated, we apply relevance-based dataset filtering. 4. *Differentiation testing*: we assess the task's ability to differentiate between the performance of two candidate models.

For further details on dataset transformations for specific tasks, we refer to the `dataset_transform` method implementation for each task.

**Classification and pair classification** For both classification tasks, we used existing datasets with minimal adjustments, primarily trimming them down to more manageable sizes. For performance evaluation, we rely on such metrics as $F_1$ score, accuracy, or average precision. Whenever feasible, we align our choice of the primary metric with those used in related publications. If no specific guidance exists, we default to accuracy for general classification tasks and average precision for pairwise classification. In scenarios with significant class imbalance, the $F_1$ score is prioritized.

**Bitext mining** Bitext mining tasks were constructed using established paired datasets. Similar to the classification tasks, the primary focus was on adjusting the dataset sizes to maintain the same model rank while reducing computational load. $F_1$ scores were chosen to be the primary metric, unless specified otherwise.

**Clustering and hierarchical clustering** Clustering tasks were derived from existing corpora, such as news articles or encyclopedic entries. The source datasets typically included categories or labels assigned by their original authors or publishers. In some cases, like the SNL and VG datasets (Navjord & Korsvik, 2023), which featured hierarchical labels, we reformulated the tasks from flat to hierarchical clustering.

**Retrieval** A variety of tasks were integrated as retrieval tasks, including existing retrieval, question-answer, and news datasets. For question-answer datasets, the questions were used as queries, and the answers formed the corpus, with correct answers identified as properly retrieved documents. In news datasets, headlines were treated as queries, and both the full articles were considered part of the corpus, with matched summaries and articles serving as relevant documents. For the primary metric, we use `nDCG@10`, unless otherwise specified by the dataset publication.

**Multi-label classification** For multi-label classification, we used existing datasets that required minimal adjustments. A critical aspect of these tasks was maintaining the balance of label distributions across the training and evaluation splits. To achieve this, we employed advanced stratification techniques (Szymański & Kajdanowicz, 2017; Sechidis et al., 2011) that consider higher-order relationships between labels, ensuring balanced samples and improved classification quality. For the main metric, we use accuracy.

**Instruction Retrieval** For instruction retrieval tasks, we incorporated datasets like FollowIR (Weller et al., 2024; 2025), which consist of comprehensive narratives created by professional assessors. These datasets were initially developed for TREC shared tasks and included rich, context-heavy queries to evaluate retrieval systems' performance on more intricate retrieval problems.

**Reranking** For reranking tasks, we adapted datasets covering a range of topics and languages, including academic paper ranking, news articles (Wu et al., 2020b), QA pair relevance from online platforms, and passage ranking (Xie et al., 2023). For the primary metric, we use MAP unless otherwise specified by the dataset publication.

**Semantic text similarity** For STS tasks, we adapted well-known benchmarks like STSbenchmark (May et al., 2021) and cross-lingual STS datasets from SemEval (Agirre et al., 2015). We also adapted paraphrase datasets in various languages, such as the Russian ParaPhraser (Pivovarova et al., 2017) and the Finnish Paraphrase Corpus (Kanerva et al., 2021). As the main metric, we use Spearman correlation based on the highest similarity (Reimers et al., 2016).

## B.3 NOVEL DATASETS

This section introduces task specifically created as a part of the MMTEB contributions. For information on how existing datasets were adapted to MTEB we refer to Appendix B.

**PublicHealthQA**: This retrieval task is built on top of a novel dataset containing question-and-answer pairs in Public Health, specifically related to the COVID-19 disease. They are sourced from Q&A pages and Frequently Asked Questions (FAQ) sections of the Centers for Disease Control and Prevention (CDC) and World Health Organization (WHO) websites. They were produced and collected between 2019-12 and 2020-04.

**WebLINXReranking**: This is a novel HTML reranking task derived from WebLINX, a benchmark for training and evaluating web agents with conversational capabilities (Lù et al., 2024). Whereas the original work introduces a retrieval task with the goal of retrieving HTML elements using a conversational context, we propose the first task with the goal of reranking HTML elements based on their relevance for actions executed in web environments, including clicks, hovers, and text insertions.

**WikiClustering**: is a multilingual clustering benchmark based on Wikipedia's main topic classifications. The goal is to create a clustering benchmark that works for multiple languages.

To construct a WikiClustering dataset for a given language, we apply the following steps. First, download the wiki dump of the categories, the articles, and the category links. Second, we find the main topic classifications for all articles. The main topic classifications can be found by looking at the category page for the language[6]. We only use the first paragraph of each article to construct a paragraph-to-paragraph (P2P) task similar to other P2P tasks within MTEB. Third, we filter out articles with more than one main topic and remove any topic with only one article associated with it. This step avoids ambiguity in the clustering task. Finally, we sample 2048 articles with associated main topics.

While the WikiClustering benchmark can be extended to any language with main topic classifications, it is currently implemented for the following: Bosnian, Catalan, Czech, Danish, Basque, Manx, Ilokano, Kurdish, Latvian, Minangkabau, Maltese, Scots, Albanian, and Walloon. All code is available on GitHub.

**WikipediaRetrievalMultilingual** and **WikipediaRerankingMultilingual**: This is a multilingual retrieval and reranking dataset based on succinct queries generated by a strong multilingual LLM grounded in Wikipedia articles. The dataset was made to resemble SQuAD. Sampled Wikipedia articles of a target language were chunked and passed to GPT4-o using the following prompt:

```
"""
Your task is to anticipate possible search queries by users in the form of a question
for a given document.
- The question must be written in {{ language }}
- The question should be formulated concretely and precisely and relate to the
information from the given document
- The question must be coherent and should make sense without knowing the document
- The question must be answerable by the document
- The question should focus on one aspect and avoid using subclauses connected with
'and'
- The question should not be overly specific and should mimic a request of a user who
is just starting to research the given topic
- Do not draw on your prior knowledge
```

---

[6]for details, we refer to `https://en.wikipedia.org/wiki/Category:Main_topic_classifications`forEnglish

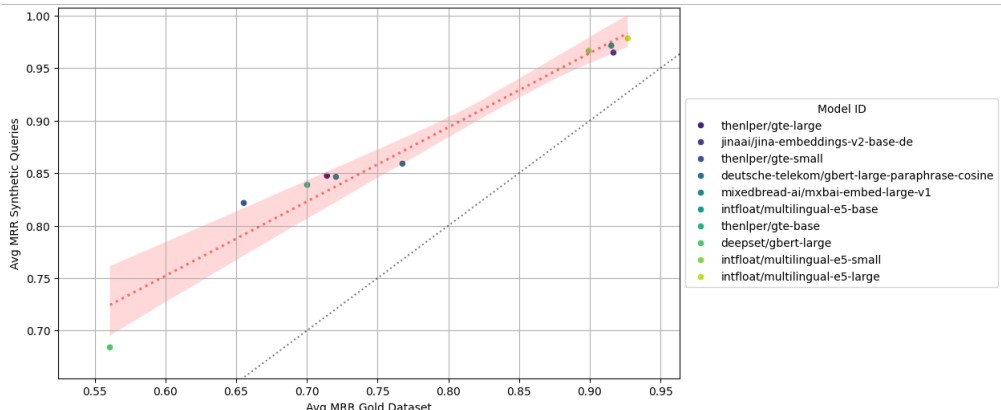

Figure 7: Comparison of MRR on synthetic retrieval and gold (GermanQuAD). The synthetic dataset was generated using GPT4-turbo.

```
Generate a question in {{ language }} for the following document:
<document>
{{ document }}
</document>

Search query:
"""
```

We filtered articles with less than 9 paragraphs and sampled 1500 articles from the top 100k viewed articles. We then selected a random window of 9 consecutive paragraphs per article and chose the middle one to be the positive context and generated a query for it with gpt-4o. The surrounding 8 paragraphs act as hard negatives. The 9 paragraphs per article are used for the reranking task with one positive and 8 negatives. The one positive, 8 hard negatives, and the remaining corpus as negatives are used in the retrieval task.

These datasets where constructed fro the following languages: "bul-Cyrl", "ben-Beng", "ces-Latn", "dan-Latn", "deu-Latn", "eng-Latn", "fas-Arab", "fin-Latn", "hin-Deva", "ita-Latn", "nld-Latn", "por-Latn", "ron-Latn", "srp-Cyrl", "dan-Latn", "nob-Latn", "swe-Latn".

To estimate the quality of these samples we compare it to the GermanQuAD (Möller et al., 2021) in Figure 7. We obtain a Spearman rank correlation of 0.93 with a 95% CI of [0.69; 1.].

## B.4   TASK METADATA

Table 5 shows the required metadata to fill before adding a task to the benchmark. We provide a detailed description of each field, along with examples and possible values.

### B.4.1   DOMAINS

For our domains, we include the following:

- **Academic**: Scholarly writing and research publications typically found in journals, theses, and dissertations.
- **Blog**: Informal or conversational posts often found on websites or personal pages, covering a wide range of topics.
- **Constructed**: Text or speech that is deliberately invented or constructed, often used for experimental purposes to target specific abilities.
- **Encyclopaedic**: Structured, reference-based texts that provide comprehensive and factual information on a wide range of subjects.

| Field | Description |
|---|---|
| Name | A concise name for the task. |
| Description | A brief explanation of the task's goals and objectives.. |
| Type | The primary task category (e.g., classification, summarization, retrieval). |
| Category | The general data structure or format of the task. This can be specified using a combination of single-letter codes (e.g., "s" for sentence, "p" for paragraph, "d" for document). For example, "s2s" indicates a sentence-to-sentence task, "s2p" indicates a sentence-to-paragraph task, and "p2p" indicates a paragraph-to-paragraph task. |
| Task Subtype | A more specific subcategory within the primary task type. This can be used to further refine the task and provide additional context. For example, "Summarization" might have subtypes like "Extractive Summarization" or "Abstractive Summarization". |
| Reference | A URL or citation to the original source material (e.g., paper, dataset repository). |
| Evaluation Splits | The specific subsets of the data used for training, validation, and testing. |
| Evaluation Languages | A list of ISO 639-3 language codes (e.g., "eng", "fra") followed by ISO 15924 script codes (e.g., "Latn", "Cyrl") for each language used in the evaluation. For example: [("eng", "Latn"), ("fra", "Latn")]. If multiple scripts are used within a single language, we specify them as a list (e.g., [("eng", ["Latn", "Grek"])]). |
| Date | The time period when the data was gathered. Specified as a tuple of two dates. |
| Main score | The primary metric used to evaluate task performance. |
| Form | The format of the data (e.g., "spoken", "written") |
| License | The licensing terms for the dataset (e.g., CC BY-SA, MIT). |
| Domains | The subject areas or fields covered by the data (e.g., medical, legal, news). One dataset can belong to multiple domains. |
| Annotation Creators | The type of the annotators. Includes "expert-annotated" (annotated by experts), "human-annotated" (annotated e.g. by mturkers), "derived" (derived from structure in the data), "LM-generated" (generated using a language model) and "LM-generated and reviewed" (generated using a language model and reviewed by humans or experts). |
| Dialect | The specific dialect or regional variation of the language. |
| Text Creation | How the text was generated. Includes "found", "created", "human-translated and localized", "human-translated", "machine-translated", "machine-translated and verified", "machine-translated and localized", "LM-generated and verified". |
| Bibtex Citation | The BibTeX format citation for the dataset. |
| Number of samples | The total number of data points in the dataset. |
| Avg. Number of characters | The average character length of the samples in the dataset. |

Table 5: Required metadata for adding a new task to MMTEB.

- **Fiction**: Narrative writing based on imaginative content, including novels, short stories, and other forms of storytelling.

- **Government**: Official documents, reports, and publications produced by governmental bodies.

- **Legal**: Documents and texts relating to laws, legal proceedings, contracts, and legal theory.

- **Medical**: Scientific and clinical literature related to healthcare, treatments, medical research, and patient care.

- **News**: Journalistic content that covers current events, politics, economy, and other topical issues.

- **Non-fiction**: Writing based on factual accounts and real-world subjects, such as biographies, essays, and documentaries.

- **Poetry**: Literary form focused on expressive language, often structured with meter, rhyme, or free verse.

- **Religious**: Texts related to religious teachings, doctrines, sacred scriptures, and spiritual discussions.

- **Reviews**: Critical evaluations of works such as books, movies, music, products, or services.

- **Social**: Written or spoken communication on social media platforms, forums, and other digital environments.

- **Spoken**: Oral communication, including speeches, dialogues, interviews, and recorded conversations.

- **Subtitles**: Textual transcriptions or translations of spoken language in films, videos, or multimedia presentations.

- **Web**: Text content found on websites, covering a wide range of subjects, often hyperlinked and multimedia-enriched.

- **Written**: General term for any form of text-based communication, whether printed or digital.
- **Programming**: Text written in programming languages to instruct computers, often for software development.

Our definition of domain aligns with that of the Universal Dependencies project (Nivre et al., 2016). We do not claim that our definition is neither precise nor comprehensive. However, and include subject fields such as "medical", "legal", and "news" and literary type such as "fiction", "non-fiction". They are not mutually exclusive.

## C  BENCHMARK OPTIMIZATIONS

### C.1  SPEEDING UP TASKS

We aim to reduce the total amount of time needed to run the complete set of MTEB task. In particular, we investigate how to drastically reduce runtime on clustering and retrieval tasks while maintaining relative model rankings. This appendix provides full details of the approach described in Section 2.3.2.

#### C.1.1  CLUSTERING

| Task | Spearman | Speedup |
|------|----------|---------|
| Biorxiv P2P | 0.9505 | 31.50x |
| Biorxiv S2S | 0.9890 | 14.31x |
| Medrxiv P2P | 0.9615 | 21.48x |
| Medrxiv S2S | 0.9560 | 8.39x |
| Reddit S2S | 0.9670 | 11.72x |
| Reddit P2P | 0.9670 | 22.77x |
| StackExchange S2S | 0.9121 | 9.55x |
| StackExchange P2P | 0.9670 | 20.20x |
| TwentyNewsgroups | 1.0000 | 5.02x |
| Average | 0.9634 | 16.11x |

Table 6: Agreement on model rankings on a selection of English clustering tasks using Spearman's correlation across the scores of 13 models of various sizes.

In the main paper, we present a down-sampled and bootstrapped version of the clustering task. We highlight the main results in Table 6 but refer to. We observe an average speedup across tasks of 16.11x while maintaining the relative ordering of models on the evaluated tasks. The largest average speed-up was seen for e5-large (16.93x), but we expect this effect to be even more pronounced among 7b or larger models.

9 single-level English clustering tasks are evaluated on 13 models across various sizes. A fraction of the documents are sampled and stratified by their target categories. At the same time, we wish to maintain robustness of the evaluation, i.e. the fast approach should be able to determine highly similar model ranking to that from the original approach. As such, we investigate the extent of agreement between the original clustering task and ours in each task on the model rankings.

The model ranking is determined from the mean of V-measure scores from evaluations, where a higher mean gives a higher model rank. Spearman's rank correlation score is then calculated based on the ranks from ours and the original approach. We additionally calculate the significant model rank which is determined by computing the significance of the given model's V-measure bootstrapped distribution based on its mean of V-measure scores using our approach against that of the original approach. Significant $S$ is then calculated based on the significant ranks from our and the original approach.

To find a balance between speedup and the robustness of the approach, 4% of the dataset is chosen as the fraction to down-sample to, with the exception of RedditS2S and StackExchange where

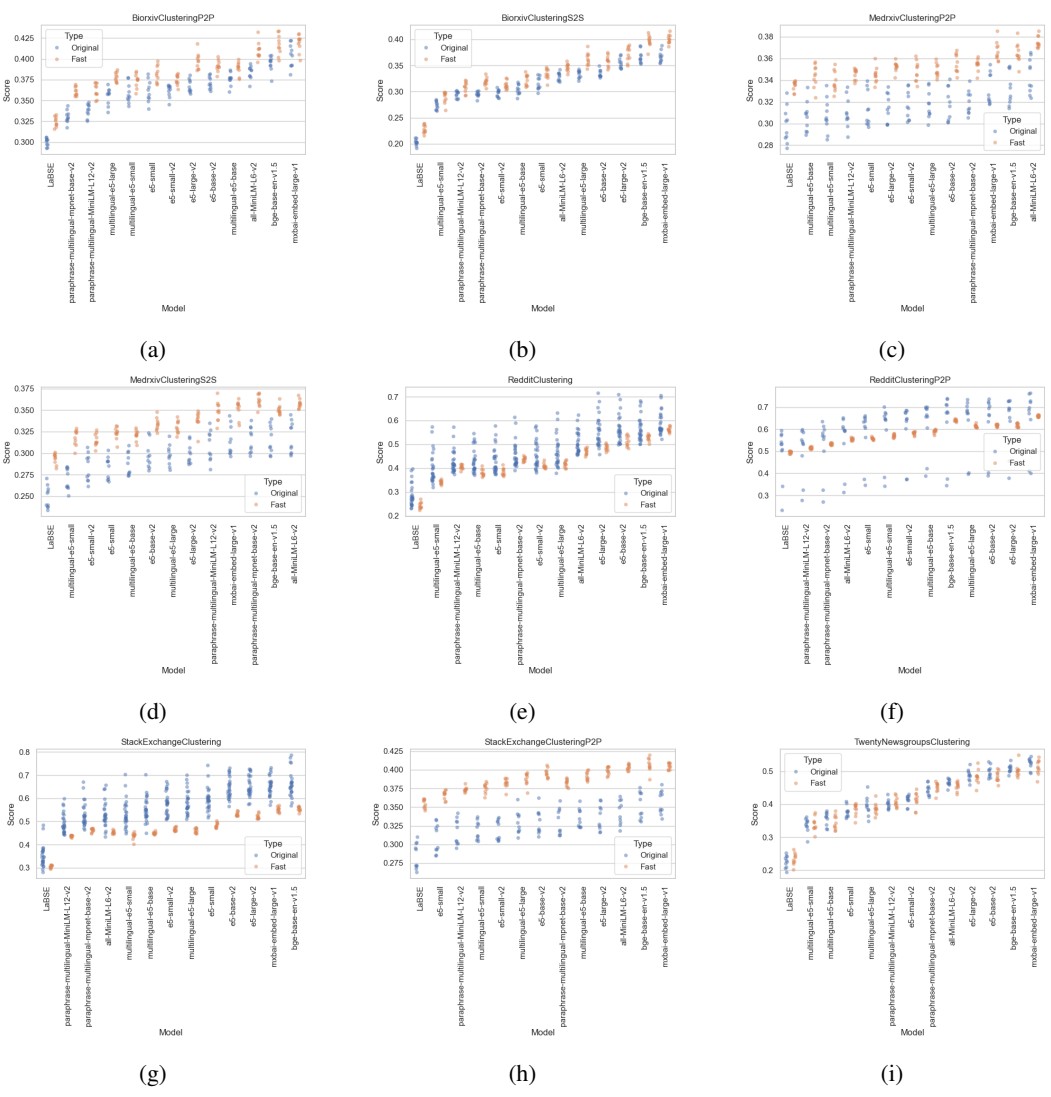

Figure 8: Distribution of scores per task across models.

| Task | Sig. $S$ |
|------|----------|
| Biorxiv P2P | 0.9390 |
| Biorxiv S2S | 0.9679 |
| Medrxiv P2P | 0.8200 |
| Medrxiv S2S | 0.9510 |
| Reddit S2S | 0.9790 |
| Reddit P2P | 0.7370 |
| StackExchange S2S | 0.9486 |
| StackExchange P2P | 0.9497 |
| TwentyNewsgroups | 0.9832 |
| Average | 0.9195 |

Table 7: Agreement on model rankings on English clustering tasks using significant Spearman's rank correlation with selected models of various sizes.

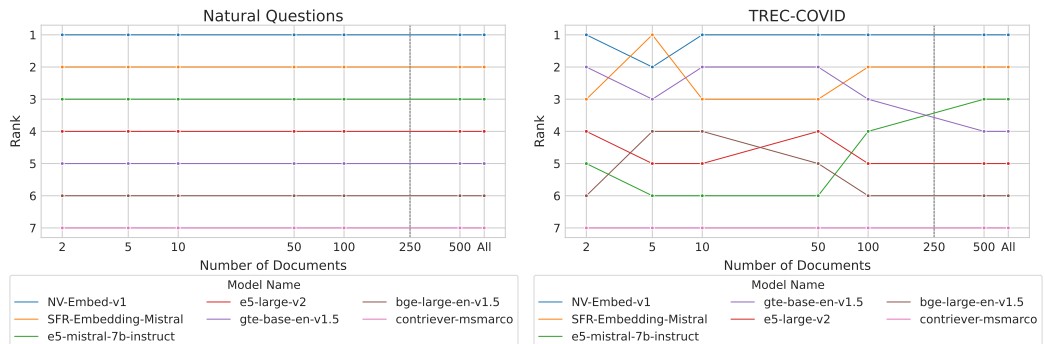

Figure 9: Ranking of different models on subsampled versions of the datasets using hard negatives. We see that NQ can be reduced to just two documents per query (relevant + 1 hard negative) while still maintaining the rank while TREC-COVID is less stable.

$n\_samples = 32768$. Table 7 shows that all evaluated datasets have very high significant Spearman's rank scores between our and the original approach. Figure 8 reports the distribution of V-measure scores obtained from evaluation per model in each dataset for the ClusteringFast and the original approach. There is generally strong agreement between the rankings from both approaches. We also observe that the ClusteringFast approach often (5 out of 9 datasets) produces a smaller spread (i.e. smaller variance) in its V-measure distributions. Reddit P2P has the lowest significant Spearman score among this set. It also has the lowest average character length for its documents.

### C.1.2 RETRIEVAL

In this section we provide details about the method used to downsample retrieval datasets.

To ensure the downsampling kept the efficacy of the evaluation we aimed to examine several axes: (1) a wide range of models to be sure that the evaluation task could still properly rank the models - just as if it were not downsampled (2) that this method works for retrieval datasets that are sparsely judged *and* densely judged and (3) seeing if it was possible to use hard negatives from a smaller set of models due to the computational expense to gather these hard negatives on the full datasets.[7]

To meet these goals we chose NQ (for sparse relevance annotations, one per query) and TREC-COVID (for dense judgements, > 500 per query). To test using a small set of hard negatives, we gather the hard negatives with e5-large-v2 only. We evaluate a wide range of models for this analysis, including the current state-of-the-art and some of the previous state-of-the-art: NV-Embed-v1 (Lee

---

[7]We also tested whether ensuring that the ground truth relevant document is present in these hard negatives made a difference - we found that it did not, as most models ranked the ground truth in the top N, so manually including it was little help as it was already included.

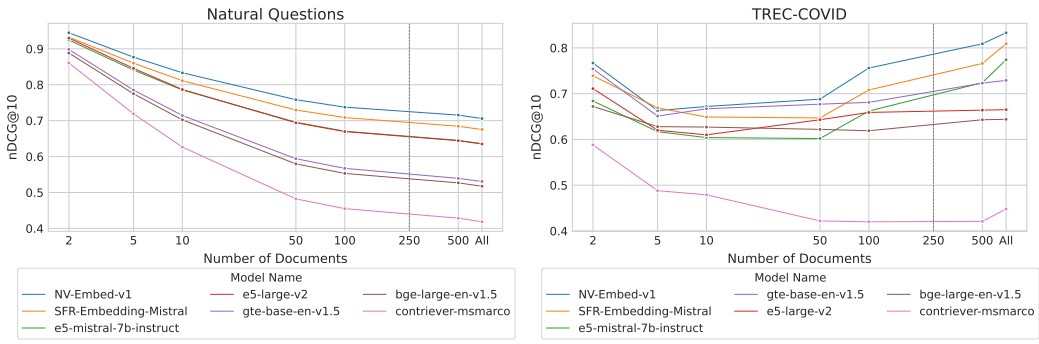

Figure 10: Absolute scores of different models on subsampled versions of the datasets using hard negatives. NQ has 1 relevant document per query while TREC-COVID has 500+ relevant documents per query which is why we see NQ scores gradually increasing whereas TREC-COVID scores vary.

et al., 2024), SFR-Embedding-Mistral (Meng et al., 2024), e5-mistral-7b-instruct (Wang et al., 2023), e5-large-v2 (Wang et al., 2022), gte-base-en-v1.5 (Li et al., 2023b), bge-large-en-v1.5 (Xiao et al., 2023), and contriever-msmarco (Izacard et al., 2021). We then evaluated the models on versions of the datasets with N hard negatives documents per query where $N \in \{2, 5, 10, 50, 100, 500, \text{all}\}$. We then compared the absolute scores and the relative rank positions to see what settings best retain the difficulty of the original task.

**Ability to rank models correctly**   For a good evaluation, it must be able to rank models correctly and determine the best model. For this we examine how the ranking of the models change when we lower the number of hard negatives. For NQ the rank remains stable even with just one hard negatives (Figure 9). For TREC-COVID the ranking becomes unstable starting at 100 hard negatives, continuing to change as the number gets smaller.

**Keeping the absolute score similar**   In an ideal case the scores for the task should remain similar and not trend towards perfect scores, remaining useful. We see that scores go very high when there are only a few hard negatives for NQ (Figure 10). For TREC-COVID it is more stable, but we see some wider swings with smaller documents. Overall, the scores are relatively similar at 100+ hard negatives.

**Summary**   Overall, we see that staying above 100 hard negatives gives similar absolute scores while maintaining the ranking ability. Thus we opted for a conservative 250 documents per query to keep these characteristics.

## C.2   CODE OPTIMIZATIONS

We here document the major code optimizations within MTEB not related to dataset scores, task reformulation

**Dataset loading** One important issue identified was about loading multilingual and cross-lingual datasets composed of numerous small files in their repositories. Even for total dataset sizes under 10MB, loading could take hours due to significant overhead from managing a high number of network requests and the improper opening and closing of gzipped files. In collaboration with the datasets team (Lhoest et al., 2021), we addressed these problems with two-side implementation improvements: the datasets library optimized the loading of a large number of requested files, and we restructured the datasets and our codebase to leverage the benefits of the newer implementation. This ultimately reduced loading times by almost a factor of 100, bringing the largely cross-lingual dataset bitext-mining loading to under a minute.

**Deduplication** Upon in-depth scrutiny of all datasets, cases with repeated samples were identified and deduplicated (e.g. MindSmallReranking). As this led to a change in scores, a second version

of the task was introduced to maintain compatible scores with existing benchmarks. To move the optimizations to existing MTEB tasks we implement a local cache to avoid encoding a sample twice.

## D TASK OVERVIEW

### D.1 TASKS

To get an overview of the all the tasks implemented in MMTEB we refer to the automatically updated tables in the documentation[8], which include the available metadata for all of the task, including license, task category, domains, etc.

### D.2 LANGUAGES

Additionally, the top 100 out of the total 1051 languages in ISO 639-3 language codes and their respective task counts are in Table 8.

| ISO Code | Language | Family | BitextMining | Classification | Clustering | InstructionRetrieval | MultilabelClassification | PairClassification | Reranking | Retrieval | STS | Speed | Summarization | Sum |
|---|---|---|---|---|---|---|---|---|---|---|---|---|---|---|
| eng | English | Indo-European | 16 | 143 | 16 | 3 | 1 | 8 | 8 | 92 | 13 | 2 | 1 | 303 |
| deu | German | Indo-European | 6 | 14 | 7 | 0 | 1 | 6 | 2 | 18 | 4 | 0 | 0 | 58 |
| fra | French | Indo-European | 7 | 13 | 8 | 0 | 1 | 5 | 3 | 15 | 4 | 0 | 1 | 57 |
| rus | Russian | Indo-European | 5 | 13 | 6 | 0 | 2 | 4 | 2 | 16 | 4 | 0 | 0 | 52 |
| pol | Polish | Indo-European | 4 | 11 | 4 | 0 | 1 | 4 | 0 | 18 | 4 | 0 | 0 | 46 |
| cmn | Mandarin Chinese | Sino-Tibetan | 4 | 10 | 4 | 0 | 0 | 3 | 4 | 10 | 9 | 0 | 0 | 44 |
| spa | Spanish | Indo-European | 4 | 13 | 4 | 0 | 1 | 2 | 2 | 13 | 4 | 0 | 0 | 43 |
| hin | Hindi | Indo-European | 9 | 12 | 2 | 0 | 0 | 1 | 2 | 10 | 2 | 0 | 0 | 38 |
| code | unknown | Programming | 0 | 0 | 0 | 0 | 0 | 0 | 0 | 37 | 0 | 0 | 0 | 37 |
| jpn | Japanese | Japonic | 5 | 8 | 3 | 0 | 0 | 1 | 3 | 13 | 2 | 0 | 0 | 35 |
| kor | Korean | Koreanic | 4 | 8 | 1 | 0 | 1 | 2 | 1 | 9 | 3 | 0 | 0 | 29 |
| ara | Arabic | Afro-Asiatic | 2 | 12 | 0 | 0 | 0 | 2 | 1 | 9 | 2 | 0 | 0 | 28 |
| ben | Bengali | Indo-European | 7 | 9 | 2 | 0 | 0 | 1 | 2 | 6 | 1 | 0 | 0 | 28 |
| ita | Italian | Indo-European | 5 | 9 | 1 | 0 | 1 | 2 | 1 | 5 | 3 | 0 | 0 | 27 |
| por | Portuguese | Indo-European | 4 | 9 | 1 | 0 | 2 | 2 | 1 | 5 | 3 | 0 | 0 | 27 |
| tel | Telugu | Dravidian | 7 | 7 | 2 | 0 | 0 | 0 | 1 | 5 | 2 | 0 | 0 | 24 |
| dan | Danish | Indo-European | 5 | 9 | 2 | 0 | 1 | 0 | 1 | 5 | 0 | 0 | 0 | 23 |
| swe | Swedish | Indo-European | 4 | 8 | 3 | 0 | 1 | 1 | 1 | 4 | 0 | 0 | 0 | 22 |
| ind | Indonesian | Austronesian | 6 | 7 | 1 | 0 | 0 | 1 | 1 | 4 | 1 | 0 | 0 | 21 |
| tam | Tamil | Dravidian | 7 | 7 | 2 | 0 | 0 | 1 | 0 | 3 | 1 | 0 | 0 | 21 |
| tha | Thai | Tai-Kadai | 4 | 8 | 1 | 0 | 0 | 1 | 1 | 6 | 0 | 0 | 0 | 21 |
| mar | Marathi | Indo-European | 7 | 6 | 2 | 0 | 0 | 1 | 0 | 2 | 2 | 0 | 0 | 20 |
| zho | Chinese | Sino-Tibetan | 2 | 2 | 1 | 0 | 0 | 1 | 1 | 13 | 0 | 0 | 0 | 20 |
| fin | Finnish | Uralic | 3 | 5 | 1 | 0 | 1 | 1 | 2 | 5 | 1 | 0 | 0 | 19 |
| kan | Kannada | Dravidian | 6 | 7 | 2 | 0 | 0 | 1 | 0 | 2 | 1 | 0 | 0 | 19 |
| mal | Malayalam | Dravidian | 7 | 7 | 2 | 0 | 0 | 0 | 0 | 2 | 1 | 0 | 0 | 19 |
| nld | Dutch | Indo-European | 6 | 6 | 1 | 0 | 1 | 0 | 1 | 2 | 2 | 0 | 0 | 19 |
| nob | Norwegian Bokmål | Unclassified | 4 | 7 | 5 | 0 | 0 | 0 | 0 | 3 | 0 | 0 | 0 | 19 |
| tur | Turkish | Turkic | 4 | 7 | 1 | 0 | 0 | 2 | 0 | 3 | 2 | 0 | 0 | 19 |
| urd | Urdu | Indo-European | 7 | 8 | 2 | 0 | 0 | 0 | 0 | 1 | 1 | 0 | 0 | 19 |

---

[8]For the latest version see `https://github.com/embeddings-benchmark/mteb/blob/main/docs/tasks.md`

| | | | | | | | | | | | | | | |
|---|---|---|---|---|---|---|---|---|---|---|---|---|---|---|
| guj | Gujarati | Indo-European | 6 | 6 | 2 | 0 | 0 | 1 | 0 | 2 | 1 | 0 | 0 | 18 |
| pan | Panjabi | Indo-European | 6 | 6 | 2 | 0 | 0 | 1 | 0 | 2 | 1 | 0 | 0 | 18 |
| ron | Romanian | Indo-European | 5 | 6 | 1 | 0 | 1 | 0 | 1 | 3 | 1 | 0 | 0 | 18 |
| vie | Vietnamese | Austroasiatic | 5 | 6 | 1 | 0 | 0 | 1 | 0 | 5 | 0 | 0 | 0 | 18 |
| fas | Persian | Indo-European | 1 | 4 | 0 | 0 | 0 | 1 | 2 | 9 | 0 | 0 | 0 | 17 |
| ces | Czech | Indo-European | 4 | 5 | 2 | 0 | 1 | 1 | 1 | 2 | 0 | 0 | 0 | 16 |
| ell | Modern Greek | Indo-European | 3 | 6 | 1 | 0 | 1 | 2 | 0 | 3 | 0 | 0 | 0 | 16 |
| yor | Yoruba | Atlantic-Congo | 4 | 5 | 3 | 0 | 0 | 0 | 1 | 3 | 0 | 0 | 0 | 16 |
| ory | Odia | Indo-European | 5 | 4 | 2 | 0 | 0 | 1 | 0 | 2 | 1 | 0 | 0 | 15 |
| swa | Swahili | Atlantic-Congo | 1 | 7 | 2 | 0 | 0 | 1 | 1 | 3 | 0 | 0 | 0 | 15 |
| amh | Amharic | Afro-Asiatic | 3 | 6 | 3 | 0 | 0 | 0 | 0 | 1 | 1 | 0 | 0 | 14 |
| asm | Assamese | Indo-European | 5 | 3 | 2 | 0 | 0 | 1 | 0 | 2 | 1 | 0 | 0 | 14 |
| hau | Hausa | Afro-Asiatic | 4 | 5 | 3 | 0 | 0 | 0 | 0 | 1 | 1 | 0 | 0 | 14 |
| bul | Bulgarian | Indo-European | 3 | 4 | 1 | 0 | 1 | 1 | 1 | 2 | 0 | 0 | 0 | 13 |
| jav | Javanese | Austronesian | 4 | 7 | 1 | 0 | 0 | 0 | 0 | 1 | 0 | 0 | 0 | 13 |
| hun | Hungarian | Uralic | 5 | 3 | 1 | 0 | 1 | 0 | 0 | 2 | 0 | 0 | 0 | 12 |
| ibo | Igbo | Atlantic-Congo | 3 | 5 | 3 | 0 | 0 | 0 | 0 | 1 | 0 | 0 | 0 | 12 |
| slk | Slovak | Indo-European | 3 | 4 | 1 | 0 | 1 | 0 | 0 | 3 | 0 | 0 | 0 | 12 |
| heb | Hebrew | Afro-Asiatic | 4 | 5 | 1 | 0 | 0 | 0 | 0 | 1 | 0 | 0 | 0 | 11 |
| afr | Afrikaans | Indo-European | 3 | 4 | 1 | 0 | 0 | 0 | 0 | 1 | 1 | 0 | 0 | 10 |
| hrv | Croatian | Indo-European | 4 | 3 | 1 | 0 | 1 | 0 | 0 | 1 | 0 | 0 | 0 | 10 |
| kat | Georgian | Kartvelian | 4 | 3 | 1 | 0 | 0 | 0 | 0 | 2 | 0 | 0 | 0 | 10 |
| san | Sanskrit | Indo-European | 5 | 3 | 1 | 0 | 0 | 1 | 0 | 0 | 0 | 0 | 0 | 10 |
| slv | Slovenian | Indo-European | 3 | 4 | 1 | 0 | 1 | 0 | 0 | 1 | 0 | 0 | 0 | 10 |
| xho | Xhosa | Atlantic-Congo | 3 | 3 | 3 | 0 | 0 | 0 | 0 | 1 | 0 | 0 | 0 | 10 |
| hye | Armenian | Indo-European | 3 | 3 | 1 | 0 | 0 | 1 | 0 | 1 | 0 | 0 | 0 | 9 |
| isl | Icelandic | Indo-European | 3 | 4 | 1 | 0 | 0 | 0 | 0 | 1 | 0 | 0 | 0 | 9 |
| min | Minangkabau | Austronesian | 3 | 4 | 2 | 0 | 0 | 0 | 0 | 0 | 0 | 0 | 0 | 9 |
| mlt | Maltese | Afro-Asiatic | 2 | 2 | 2 | 0 | 2 | 0 | 0 | 1 | 0 | 0 | 0 | 9 |
| mya | Burmese | Sino-Tibetan | 3 | 4 | 1 | 0 | 0 | 0 | 0 | 1 | 0 | 0 | 0 | 9 |
| som | Somali | Afro-Asiatic | 3 | 2 | 3 | 0 | 0 | 0 | 0 | 1 | 0 | 0 | 0 | 9 |
| srp | Serbian | Indo-European | 4 | 1 | 1 | 0 | 0 | 0 | 1 | 2 | 0 | 0 | 0 | 9 |
| sun | Sundanese | Austronesian | 3 | 4 | 1 | 0 | 0 | 0 | 0 | 1 | 0 | 0 | 0 | 9 |
| arb | Standard Arabic | Afro-Asiatic | 3 | 1 | 1 | 0 | 0 | 0 | 0 | 2 | 1 | 0 | 0 | 8 |
| cat | Catalan | Indo-European | 3 | 2 | 2 | 0 | 0 | 0 | 0 | 1 | 0 | 0 | 0 | 8 |
| cym | Welsh | Indo-European | 3 | 4 | 1 | 0 | 0 | 0 | 0 | 0 | 0 | 0 | 0 | 8 |
| est | Estonian | Uralic | 2 | 2 | 1 | 0 | 1 | 0 | 0 | 2 | 0 | 0 | 0 | 8 |
| eus | Basque | Unclassified | 3 | 2 | 2 | 0 | 0 | 0 | 0 | 1 | 0 | 0 | 0 | 8 |
| kaz | Kazakh | Turkic | 3 | 3 | 1 | 0 | 0 | 0 | 0 | 1 | 0 | 0 | 0 | 8 |
| khm | Khmer | Austroasiatic | 3 | 3 | 1 | 0 | 0 | 0 | 0 | 1 | 0 | 0 | 0 | 8 |
| kin | Kinyarwanda | Atlantic-Congo | 2 | 3 | 1 | 0 | 0 | 0 | 0 | 1 | 1 | 0 | 0 | 8 |
| lin | Lingala | Atlantic-Congo | 2 | 2 | 3 | 0 | 0 | 0 | 0 | 1 | 0 | 0 | 0 | 8 |
| lit | Lithuanian | Indo-European | 4 | 1 | 1 | 0 | 1 | 0 | 0 | 1 | 0 | 0 | 0 | 8 |
| lug | Ganda | Atlantic-Congo | 2 | 2 | 3 | 0 | 0 | 0 | 0 | 1 | 0 | 0 | 0 | 8 |
| nno | Norwegian Nynorsk | Unclassified | 4 | 3 | 1 | 0 | 0 | 0 | 0 | 0 | 0 | 0 | 0 | 8 |
| npi | Nepali | Indo-European | 4 | 2 | 1 | 0 | 0 | 0 | 0 | 1 | 0 | 0 | 0 | 8 |
| sna | Shona | Atlantic-Congo | 2 | 2 | 3 | 0 | 0 | 0 | 0 | 1 | 0 | 0 | 0 | 8 |
| snd | Sindhi | Indo-European | 4 | 2 | 1 | 0 | 0 | 0 | 0 | 1 | 0 | 0 | 0 | 8 |
| tgl | Tagalog | Austronesian | 3 | 3 | 1 | 0 | 0 | 0 | 0 | 1 | 0 | 0 | 0 | 8 |
| tir | Tigrinya | Afro-Asiatic | 2 | 2 | 3 | 0 | 0 | 0 | 0 | 1 | 0 | 0 | 0 | 8 |
| ukr | Ukrainian | Indo-European | 4 | 2 | 1 | 0 | 0 | 0 | 0 | 1 | 0 | 0 | 0 | 8 |
| ary | Moroccan Arabic | Afro-Asiatic | 1 | 3 | 1 | 0 | 0 | 0 | 0 | 1 | 1 | 0 | 0 | 7 |
| bug | Buginese | Austronesian | 2 | 4 | 1 | 0 | 0 | 0 | 0 | 0 | 0 | 0 | 0 | 7 |
| fao | Faroese | Indo-European | 3 | 2 | 1 | 0 | 0 | 0 | 0 | 0 | 1 | 0 | 0 | 7 |
| kir | Kirghiz | Turkic | 2 | 3 | 1 | 0 | 0 | 0 | 0 | 1 | 0 | 0 | 0 | 7 |
| mai | Maithili | Indo-European | 4 | 2 | 1 | 0 | 0 | 0 | 0 | 0 | 0 | 0 | 0 | 7 |
| mkd | Macedonian | Indo-European | 3 | 2 | 1 | 0 | 0 | 0 | 0 | 1 | 0 | 0 | 0 | 7 |
| mni | Manipuri | Sino-Tibetan | 4 | 2 | 1 | 0 | 0 | 0 | 0 | 0 | 0 | 0 | 0 | 7 |
| pcm | Nigerian Pidgin | Indo-European | 1 | 4 | 2 | 0 | 0 | 0 | 0 | 0 | 0 | 0 | 0 | 7 |

| sat | Santali | Austroasiatic | 4 | 2 | 1 | 0 | 0 | 0 | 0 | 0 | 0 | 0 | 0 | 7 |
| sin | Sinhala | Indo-European | 2 | 3 | 1 | 0 | 0 | 0 | 0 | 1 | 0 | 0 | 0 | 7 |
| ssw | Swati | Atlantic-Congo | 2 | 3 | 1 | 0 | 0 | 0 | 0 | 1 | 0 | 0 | 0 | 7 |
| tsn | Tswana | Atlantic-Congo | 2 | 3 | 1 | 0 | 0 | 0 | 0 | 1 | 0 | 0 | 0 | 7 |
| tso | Tsonga | Atlantic-Congo | 1 | 4 | 1 | 0 | 0 | 0 | 0 | 1 | 0 | 0 | 0 | 7 |
| uig | Uighur | Turkic | 4 | 2 | 1 | 0 | 0 | 0 | 0 | 0 | 0 | 0 | 0 | 7 |
| zul | Zulu | Atlantic-Congo | 2 | 3 | 1 | 0 | 0 | 0 | 0 | 1 | 0 | 0 | 0 | 7 |
| awa | Awadhi | Indo-European | 3 | 2 | 1 | 0 | 0 | 0 | 0 | 0 | 0 | 0 | 0 | 6 |
| bak | Bashkir | Turkic | 2 | 3 | 1 | 0 | 0 | 0 | 0 | 0 | 0 | 0 | 0 | 6 |
| bel | Belarusian | Indo-European | 4 | 1 | 1 | 0 | 0 | 0 | 0 | 0 | 0 | 0 | 0 | 6 |
| bho | Bhojpuri | Indo-European | 2 | 2 | 1 | 0 | 0 | 1 | 0 | 0 | 0 | 0 | 0 | 6 |
| bod | Tibetan | Sino-Tibetan | 3 | 1 | 1 | 0 | 0 | 0 | 0 | 1 | 0 | 0 | 0 | 6 |
| bos | Bosnian | Indo-European | 3 | 1 | 2 | 0 | 0 | 0 | 0 | 0 | 0 | 0 | 0 | 6 |
| ceb | Cebuano | Austronesian | 3 | 1 | 1 | 0 | 0 | 0 | 0 | 1 | 0 | 0 | 0 | 6 |
| ckb | Central Kurdish | Indo-European | 3 | 1 | 1 | 0 | 0 | 0 | 0 | 1 | 0 | 0 | 0 | 6 |
| ilo | Iloko | Austronesian | 2 | 1 | 2 | 0 | 0 | 0 | 0 | 1 | 0 | 0 | 0 | 6 |

Table 8: The top 100 languages across all MMTEB tasks in ISO 639-3 language codes and their respective task counts.

### D.3 EXAMPLES

Table 9 and Table 10 provide examples for each new task type introduced in MMTEB. For examples of bitext mining, classification, clustering, pair classification, reranking, retrieval, STS, and summarization datasets, we refer to the MTEB paper Muennighoff et al. (2023b).

| Dataset | Query | OG Instructions | Short query | Relevant Document |
|---|---|---|---|---|
| Robust04 | Who is involved in the Schengen agreement to eliminate border controls in Western Europe and what do they hope to accomplish? | Relevant documents will contain any information about the actions of signatories of the Schengen agreement such as: measures to eliminate border controls (removal of traffic obstacles, lifting of traffic restrictions); implementation of the information system data bank that contains unified visa issuance procedures; or strengthening of border controls at the external borders of the treaty area in exchange for free movement at the internal borders. Discussions of border crossovers for business purposes are not relevant. | Find documents that answer this question on Schengen agreement actions. | ... Schengen Space Concerning the mission traditionally performed by PAF–overseeing border traffic–the new directorate must fit into a Europe of immigration. The interior minister is therefore asking DICILC to step up its control of crossborder traffic, "particularly at the future external borders of the Schengen space." Originally scheduled in February 1994 but constantly postponed, the implementation of the agreements signed in Schengen by nine European countries (the Twelve, minus Great Britain, Ireland, and Denmark), provides for the free circulation of nationals within the space common to the territories of their nine countries... |

Table 9: Instruction Retrieval examples.

| Dataset | Text | Label |
|---------|------|-------|
| Maltese News Categories | Hi kellha 82 sena Id-dinja mużikali fl-Italja tinsab f'luttu wara l-mewt tal-attriċi u kantanta popolari Milva, li fis-snin 70 kienet meqjusa "ikona" fost it-Taljani. Milva kienet kisbet suċċess kbir, fl-istess epoka ta' Mina u Ornella Vanoni. Milva ħarġet numru kbir ta' albums tul il-karriera tagħha u ħadet sehem f'Sanremo għal xejn anqas minn 15-il darba; iżda qatt ma rebħet il-festival. Hi kellha 82 sena, u telqet mix-xena tal-ispettaklu eżatt 10 snin ilu. | [ culture(2), international(10) ] |

Table 10: Multilabel Classification examples.

```python
import mteb
from mteb.task_selection import results_to_dataframe

tasks = mteb.get_tasks(
    task_types=["Retrieval"],
    languages=["eng", "fra"],
    domains=["legal"]
)

model_names = [
    "intfloat/multilingual-e5-small",
    "intfloat/multilingual-e5-base",
    "intfloat/multilingual-e5-large",
]

models = [mteb.get_model_meta(name) for name in model_names]

results = mteb.load_results(models=models, tasks=tasks)

df = results_to_dataframe(results)
```

Figure 11: Simple example of how to obtain all scores on English (eng) and French (fra) retrieval tasks within the Legal domain for a set of models.

# E  FULL RESULTS

During this work, multiple models were evaluated on more than >500 tasks, with multiple tasks containing multiple language subsets covering more than 1000 languages. This makes a comprehensive overview unreasonable. While we have supplied scores aggregated across task categories, we realize that readers might be interested in examining scores for their specific language, domain of interest, and task. To ensure that such aggregation is available and easily accessible, we make all results available on the public and versioned results repository [9]. These results include time of run, evaluation time, and a wide set of performance metrics pr. language subset, CO2 emission, version number, and more.

To make these detailed results subject to easy analysis, we have added functionality for loading and aggregating these results within the mteb package. It is, for instance, possible to retrieve the scores for specific models on all English (eng) and French (fra) retrieval tasks within the Legal domain using the code snippet in Figure 11

We refer to the documentation[10] for the latest version of this code.

---

[9]https://github.com/embeddings-benchmark/results for the specific version of the repository used for this work see commit id 9a79f7e07542ad2f5cb47490fa1e5ac2ba57d7a8

[10]https://github.com/embeddings-benchmark/mteb

## E.1 PERFORMANCE PER NUMBER OF SPEAKERS

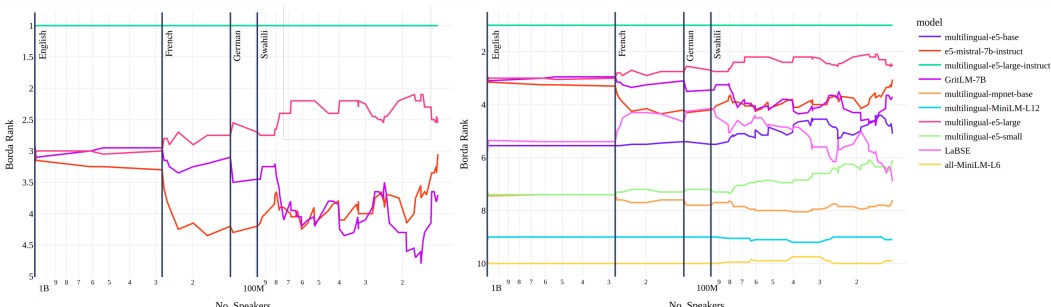

Figure 12: Models' rank on the `MTEB(Multilingual)` by the total number of speakers of a language. *Trendlines represent moving average with a window size of 10*

## F NEW METRICS

### F.1 ABSTENTION FOR RETRIEVAL AND RERANKING TASKS

In addition to the existing ranking metrics used for Retrieval and Reranking tasks (Muennighoff et al., 2023b), we propose to assess score calibration through the evaluation of model abstention ability, using the implementation of Gisserot-Boukhlef et al. (2024).

Intuitively, a model abstains on a given instance $(q, d_1, \cdots, d_k)$ (one query and $k$ candidate documents) if $c(q, d_1, \cdots, d_k) < \tau$, where $c$ is a confidence function[11] and $\tau$ is a threshold regulating abstention likelihood. Therefore, to evaluate abstention capacity on a given test set $\mathcal{S}$, an approach consists of making $\tau$ vary to achieve several abstention rates. In the case of effective abstention, the metric score increases with the abstention rate.

More formally, models' ability to abstain is evaluated by computing the normalized area under the metric-abstention curve ($n\mathsf{AUC}$). Given a confidence function $c$, a metric function $m$[12] and a labeled test dataset $\mathcal{S}$, $n\mathsf{AUC}$ is computed as follows:

1. **Multi-thresholding:** Given a model $f$ and dataset $\mathcal{D}$, we define a set of abstention thresholds $\tau_1, \ldots, \tau_n$, such that $\tau_1 < \cdots < \tau_n$. For each threshold $\tau_i$, we construct a corresponding sub-dataset $\mathcal{S}_i \subseteq \mathcal{D}$ by applying the abstention criterion. We then evaluate the model $f$ on each sub-dataset $\mathcal{S}_i$ using the metric function $m$. To quantify the model's performance across these thresholds, we compute the area under the metric-abstention curve, denoted as $\mathsf{AUC}_{model}$.

2. **Compute lower-bound:** Since $\mathsf{AUC}_{model}$ depends on the model's raw performance without abstention, we compute the effective lower bound $\mathsf{AUC}^-$. This corresponds to the area under the curve when the metric remains constant as abstention increases, representing the baseline where abstention does not improve the metric.

3. **Compute upper-bound:** To establish the upper bound, $\mathsf{AUC}^+$, we evaluate an oracle model that has access to the true labels. The oracle can selectively retain the best instances at each abstention rate, yielding the theoretical maximum area under the metric-abstention curve. This represents the optimal model performance under abstention.

4. **Compute normalized AUC:** Finally, we compute the normalized area under the curve, denoted $n\mathsf{AUC}_{model}$, by scaling $\mathsf{AUC}_{model}$ between the lower and upper bounds:

$$n\mathsf{AUC}_{model} = \frac{\mathsf{AUC}_{model} - \mathsf{AUC}^-}{\mathsf{AUC}^+ - \mathsf{AUC}^-}$$

---

[11] In our implementation, we rely on three simple confidence functions all taking the instance's query-document cosine similarity scores as input: the maximum score, the standard deviation of scores and the difference between the highest and second highest scores.

[12] We utilize the metrics initially implemented for the evaluation of Retrieval and Reranking MTEB tasks (Muennighoff et al., 2023b).

| Name in Paper | HF Name | Revision ID |
|---|---|---|
| GritLM-7B | GritLM/GritLM-7B | 13f00a0e36500c80ce12870ea513846a066004af |
| e5-mistral-7b-instruct | intfloat/e5-mistral-7b-instruct | 07163b72af1488142a360786df853f237b1a3ca1 |
| multilingual-e5-base | intfloat/multilingual-e5-base | d13f1b27baf31030b7fd040960d60d909913633f |
| multilingual-e5-large | intfloat/multilingual-e5-large | 4dc6d853a804b9c8886ede6dda8a073b7dc08a81 |
| multilingual-e5-large-instruct | intfloat/multilingual-e5-large-instruct | baa7be480a7de1539afce709c8f13f833a510e0a |
| multilingual-e5-small | intfloat/multilingual-e5-small | e4ce9877abf3edfe10b0d82785e83bdcb973e22e |
| LaBSE | s-t/LaBSE | e34fab64a3011d2176c99545a93d5cbddc9a91b7 |
| all-MiniLM-L12 | s-t/all-MiniLM-L12-v2 | a05860a77cef7b37e0048a7864658139bc18a854 |
| all-MiniLM-L6 | s-t/all-MiniLM-L6-v2 | 8b3219a92973c328a8e22fadcfa821b5dc75636a |
| all-mpnet-base | s-t/all-mpnet-base-v2 | 84f2bcc00d77236f9e89c8a360a00fb1139bf47d |
| multilingual-MiniLM-L12 | s-t/paraphrase-multilingual-MiniLM-L12-v2 | bf3bf13ab40c3157080a7ab344c831b9ad18b5eb |
| multilingual-mpnet-base | s-t/paraphrase-multilingual-mpnet-base-v2 | 79f2382ceacceacdf38563d7c5d16b9ff8d725d6 |

Table 11: Model name as it appears in the paper, its name on Huggingface Hub, and their associated revision IDs. Note: s-t stands for sentence-transformers.

.

# G  MODELS

Models used for task selection along with their revision IDs can be found in Table 11. Code for running the models, including prompts, is available within MTEB's model registry available at `https://github.com/embeddings-benchmark/mteb/tree/main/mteb/models`. Unless otherwise specified within the model implementation, the prompt is available in the file `https://github.com/embeddings-benchmark/mteb/blob/main/mteb/models/instructions.py`. As some debugging happened during the running of the models, multiple versions of MTEB were used. Due to the computational cost of running these large models on the vast amount of datasets, it was deemed unfeasible to run all the models using the exact same version. However, for each task, all models were run on the same version of the specific task. Model results can be found in JSON format in the results repository; these include additional performance metrics, model metadata, $CO_2$ emission, time of run, and exact version of MTEB used: `https://github.com/embeddings-benchmark/results/tree/9a79f7e07542ad2f5cb47490fa1e5ac2ba57d7a8`.

# H  BENCHMARK CONSTRUCTION AND OVERVIEW

## H.1  BENCHMARK CREATION

The following section introduces benchmarks created as a part of the MMTEB open contribution, which aren't introduced within the main article. MTEB additionally includes a variety of benchmark including the language-specific, notably the original English MTEB, `MTEB(eng, v2)` (Muennighoff et al., 2023b), the Scandinavian embedding benchmark `MTEB(Scandinavian)` (Enevoldsen et al., 2024), the French benchmark `MTEB(fra)` (Ciancone et al., 2024), the German benchmark `MTEB(deu)` (Wehrli et al., 2024), the Korean benchmark `MTEB(kor)`, the Chinese benchmark (Xiao et al., 2024b), the Polish benchmark `MTEB(pol)` (Poświata et al., 2024). Along with these MTEB also include an instruction based retrieval based benchmark `MTEB(FollowIR)` (Weller et al., 2024), a benchmark for law `MTEB(Law)`, the bitext section of the MINER benchmark `MINERSBitextMining` target at low resource languages (Winata et al., 2024b), and the CoIR benchmark for code retrieval `CoIR` (Li et al., 2024). For this benchmark, we refer to their associated paper and pull requests.

For an up-to-date overview of maintained benchmarks please see the benchmark registry.[13]

**MTEB(rus)** (Snegirev et al., 2024): Although Russian has approximately 258 million speakers world-wide, it was almost completely absent from the original benchmark and represented only in few multilingual datasets (e.g., MassiveIntentClassification). To address this problem, we included a number of Russian datasets in the new multilingual benchmark. For this, we selected popular Russian time-tested and community-tested datasets representing the main MMTEB tasks. Additionally, we performed data cleaning and automatic filtering, where necessary, and formatted datasets in the

---
[13]`https://github.com/embeddings-benchmark/mteb/blob/main/mteb/benchmarks.py`

MMTEB format. The final Russian part includes 18 datasets covering 7 main tasks: Classification (7 datasets), Clustering (3 datasets), MultiLabelClassification (2 tasks), PairClassification (1 task), Reranking (1 task), Retrieval (2 tasks), and STS (2 tasks). This dataset was manually constructed.

**RAR-b**: The Reasoning as Retrieval Benchmark (RAR-b) (Xiao et al., 2024a) evaluates reasoning-level understanding abilities stored in embedding models, and assesses whether correct answers to reasoning questions can be retrieved as top similar to queries, under w/ and w/o instruction settings. The benchmark provides insights into whether representations of nuanced expressions are aligned and well-encoded by current embedding models, going beyond the established reliance on evaluating with STS or traditional topical-level IR tasks.

The benchmark puts together 17 tasks made from 15 datasets (with reasoning questions from 12 datasets and 3 extra datasets to enlarge the corpus), covering 1) commonsense reasoning: WinoGrande, PIQA, SIQA, $\alpha$NLI, HellaSwag, ARC-Challenge, Quail, CSTS (Sakaguchi et al., 2021; Bisk et al., 2020; Sap et al., 2019; Bhagavatula et al., 2020; Zellers et al., 2019; Clark et al., 2018; Rogers et al., 2020; Deshpande et al., 2023), 2) temporal reasoning (Tan et al., 2023), 3) spatial reasoning: SpartQA (Mirzaee et al., 2021), 4) numerical reasoning: GSM8K, MATH (Hendrycks et al., 2021b; Cobbe et al., 2021; Yu et al., 2023), and 5) symbolic reasoning: HumanEvalPack and MBPP (Husain et al., 2019; Austin et al., 2021; Chen et al., 2021; Muennighoff et al., 2023a). The comprehensive assessment provides an early checkpoint for abilities envisioned to be necessary for next-generation embedding models (Xiao et al., 2024a).

**MTEB(Europe)**: We begin by selecting 56 official languages of the European Union, along with languages recognized by Schengen-area countries, such as Norwegian Bokmål, Icelandic, Romani, and Basque. This initial selection results in 420 tasks. We then reduce this selection by filtering out machine-translated datasets, datasets with unclear licenses, and highly specialized datasets (e.g., code retrieval datasets). Additionally, we remove tasks such as `AfriSentiClassification`, which, while containing European languages, primarily target African or Indic languages. After these exclusions, 228 tasks remain. Next, we run a representative selection of models (see Section [3.1]) and iteratively filter out the most predictable tasks (see Section [2.3.3]). To preserve language diversity and ensure fair representation across task categories, we avoid removing any task if it would eliminate a language from a particular task category. Furthermore, we retain tasks where the mean squared error between predicted and observed performance exceeds 0.5 standard deviations. This process continues until the most predictable tasks yield a Spearman correlation of less than 0.8 between predicted and observed scores, or until no further tasks can be removed. Ultimately, this results in a final selection of 96 tasks. Finally, contributors proficient in the target languages review the selected tasks, replacing some manually with higher-quality alternatives if necessary.

**MTEB(Indic)**: This benchmark is constructed similarly to the previous European benchmark but focuses on a set of Indic languages.[14] Initially, we selected 55 tasks. After manual filtering, 44 tasks remain, and following task selection and review, the final benchmark contains 23 tasks.

## H.2 BENCHMARK TASK OVERVIEW

The following tables give an overview of the tasks available within constructed benchmarks. For more information about the specific tasks, we refer to the task metadata available through the mteb package. [15]

- Table 12 and Table 13: Gives an overview of the 'MTEB(Multilingual)' benchmark
- Table 14: Gives an overview of the 'MTEB(Europe)' benchmark
- Table 15: Gives an overview of the 'MTEB(Indic)' benchmark
- Table 16: Gives an overview of the 'MTEB(eng, v2)' benchmark
- Table 17: Gives an overview of the 'MTEB(Code)' benchmark

---

[14]The following iso639-3 codes: asm, awa, ben, bgc, bho, doi, gbm, gom, guj, hin, hne, kan, kas, mai, mal, mar, mni, mup, mwr, nep, npi, ori, ory, pan, raj, san, snd, tam, tel, urd
[15]https://github.com/embeddings-benchmark/mteb

| Type | Name | Languages | Domains | Sample creation | Annotations creators | Nb samples |
|---|---|---|---|---|---|---|
| BitextMining | BUCC.v2 Zweigenbaum et al. (2017) | ['cmn', 'deu', 'eng', ...] | ['Written'] | human-translated | human-annotated | 35000 |
| | BibleNLPBitextMining Akerman et al. (2023) | ['aai', 'aak', 'aau', ...] | ['Religious', 'Written'] | created | expert-annotated | 417452 |
| | BornholmBitextMining Derczynski & Kjeldsen | ['dan'] | ['Web', 'Social', 'Fiction', ...] | created | expert-annotated | 500 |
| | DiaBlaBitextMining González et al. (2019) | ['eng', 'fra'] | ['Social', 'Written'] | created | human-annotated | 11496 |
| | FloresBitextMining Goyal et al. (2022) | ['ace', 'acm', 'acq', ...] | ['Non-fiction', 'Encyclopaedic', 'Written'] | created | human-annotated | 41908944 |
| | IN22GenBitextMining Gala et al. (2023) | ['asm', 'ben', 'brx', ...] | ['Web', 'Legal', 'Government', ...] | created | expert-annotated | 518144 |
| | IndicGenBenchFloresBitextMining Singh et al. (2024a) | ['asm', 'awa', 'ben', ...] | ['Web', 'News', 'Written'] | human-translated and localized | expert-annotated | 116522 |
| | NTREXBitextMining Federmann et al. (2022) | ['afr', 'amh', 'arb', ...] | ['News', 'Written'] | human-translated and localized | expert-annotated | 3826252 |
| | NollySentiBitextMining Shode et al. (2023) | ['eng', 'hau', 'ibo', ...] | ['Social', 'Reviews', 'Written'] | found | human-annotated | 1640 |
| | NorwegianCourtsBitextMining Tiedemann & Thottingal (2020) | ['nno', 'nob'] | ['Legal', 'Written'] | found | human-annotated | 228 |
| | NusaTranslationBitextMining Cahyawijaya et al. (2023c) | ['abs', 'bbc', 'bew', ...] | ['Social', 'Written'] | created | human-annotated | 50200 |
| | NusaXBitextMining Winata et al. (2023b) | ['ace', 'ban', 'bbc', ...] | ['Reviews', 'Written'] | created | human-annotated | 5500 |
| | Tatoeba community (2021) | ['afr', 'amh', 'ang', ...] | ['Written'] | found | human-annotated | 88877 |
| Classification | AfriSentiClassification Muhammad et al. (2023) | ['amh', 'arq', 'ary', ...] | ['Social', 'Written'] | found | derived | 18222 |
| | AmazonCounterfactualClassification O'Neill et al. (2021) | ['deu', 'eng', 'jpn'] | ['Reviews', 'Written'] | found | human-annotated | 5805 |
| | BulgarianStoreReviewSentimentClassfication Georgieva-Trifonova et al. (2018) | ['bul'] | ['Reviews', 'Written'] | found | human-annotated | 182 |
| | CSFDSKMovieReviewSentimentClassification Štefánik et al. (2023) | ['slk'] | ['Reviews', 'Written'] | found | derived | 2048 |
| | CataloniaTweetClassification Zotova et al. (2020) | ['cat', 'spa'] | ['Social', 'Government', 'Written'] | created | expert-annotated | 8051 |
| | CyrillicTurkicLangClassification Goldhahn et al. (2012) | ['bak', 'chv', 'kaz', ...] | ['Web', 'Written'] | found | derived | 2048 |
| | CzechProductReviewSentimentClassification Habernal et al. (2013) | ['ces'] | ['Reviews', 'Written'] | found | derived | 2048 |
| | DBpediaClassification Zhang et al. (2015) | ['eng'] | ['Encyclopaedic', 'Written'] | found | derived | 2048 |
| | DalajClassification Volodina et al. (2021) | ['swe'] | ['Non-fiction', 'Written'] | created | expert-annotated | 888 |
| | EstonianValenceClassification Pajupuu et al. (2023) | ['est'] | ['News', 'Written'] | found | human-annotated | 818 |
| | FilipinoShopeeReviewsClassification Riego et al. (2023) | ['fil'] | ['Social', 'Written'] | found | human-annotated | 4096 |
| | FinancialPhrasebankClassification Malo et al. (2014) | ['eng'] | ['News', 'Written', 'Financial'] | found | expert-annotated | 2264 |
| | GreekLegalCodeClassification Papaloukas et al. (2021) | ['ell'] | ['Legal', 'Written'] | found | human-annotated | 4096 |
| | GujaratiNewsClassification | ['guj'] | ['News', 'Written'] | found | derived | 1318 |
| | IndicLangClassification Madhani et al. (2023) | ['asm', 'ben', 'brx', ...] | ['Web', 'Non-fiction', 'Written'] | created | expert-annotated | 30418 |
| | IndonesianIdClickbaitClassification William & Sari (2020) | ['ind'] | ['News', 'Written'] | found | expert-annotated | 2048 |
| | IsiZuluNewsClassification Madodonga et al. (2023) | ['zul'] | ['News', 'Written'] | found | human-annotated | 752 |
| | ItaCaseholdClassification Licari et al. (2023) | ['ita'] | ['Legal', 'Government', 'Written'] | found | expert-annotated | 221 |
| | KorSarcasmClassification Kim & Cho (2019) | ['kor'] | ['Social', 'Written'] | found | expert-annotated | 2048 |
| | KurdishSentimentClassification Badawi et al. (2024) | ['kur'] | ['Web', 'Written'] | found | derived | 1987 |
| | MacedonianTweetSentimentClassification Jovanoski et al. (2015) | ['mkd'] | ['Social', 'Written'] | found | human-annotated | 1139 |
| | MasakhaNEWSClassification Adelani et al. (2023b) | ['amh', 'eng', 'fra', ...] | ['News', 'Written'] | found | expert-annotated | 6242 |
| | MassiveIntentClassification FitzGerald et al. (2022) | ['afr', 'amh', 'ara', ...] | ['Spoken'] | human-translated and localized | human-annotated | 255357 |
| | MultiHateClassification R"ottger et al. (2021) | ['ara', 'cmn', 'deu', ...] | ['Constructed', 'Written'] | created | expert-annotated | 11000 |
| | NepaliNewsClassification Arora (2020) | ['nep'] | ['News', 'Written'] | found | derived | 2048 |
| | NordicLangClassification Haas & Derczynski (2021) | ['dan', 'fao', 'isl', ...] | ['Encyclopaedic'] | found | derived | 3000 |
| | NusaParagraphEmotionClassification Cahyawijaya et al. (2023b) | ['bbc', 'bew', 'bug', ...] | ['Non-fiction', 'Fiction', 'Written'] | found | human-annotated | 5700 |
| | NusaX-senti Winata et al. (2022) | ['ace', 'ban', 'bbc', ...] | ['Reviews', 'Web', 'Social', ...] | found | expert-annotated | 4800 |
| | OdiaNewsClassification Kunchukuttan et al. (2020) | ['ory'] | ['News', 'Written'] | found | derived | 2048 |
| | PAC Łukasz Augustyniak et al. (2022) | ['pol'] | ['Legal', 'Written'] | | | 3453 |
| | PoemSentimentClassification Sheng & Uthus (2020) | ['eng'] | ['Reviews', 'Written'] | found | human-annotated | 209 |
| | PolEmo2.0-OUT | ['pol'] | ['Written', 'Social'] | | | 494 |
| | PunjabiNewsClassification Kunchukuttan et al. (2020) | ['pan'] | ['News', 'Written'] | found | derived | 157 |
| | ScalaClassification Nielsen (2023) | ['dan', 'nno', 'nob', ...] | ['Fiction', 'News', 'Non-fiction', ...] | created | human-annotated | 8192 |
| | SentimentAnalysisHindi Parida et al. (2023) | ['hin'] | ['Reviews', 'Written'] | found | derived | 2048 |
| | SinhalaNewsClassification de Silva (2015) | ['sin'] | ['News', 'Written'] | found | derived | 2048 |
| | SiswatiNewsClassification Madodonga et al. (2023) | ['ssw'] | ['News', 'Written'] | found | human-annotated | 80 |
| | SlovakMovieReviewSentimentClassification Stef'anik et al. (2023) | ['svk'] | ['Reviews', 'Written'] | found | derived | 2048 |
| | SwahiliNewsClassification Davis (2020) | ['swa'] | ['News', 'Written'] | found | derived | 2048 |
| | SwissJudgementClassification Niklaus et al. (2022) | ['deu', 'fra', 'ita'] | ['Legal', 'Written'] | found | expert-annotated | 4908 |
| | ToxicConversationsClassification cjadams et al. (2019) | ['eng'] | ['Social', 'Written'] | found | human-annotated | 2048 |
| | TswanaNewsClassification Marivate et al. (2023) | ['tsn'] | ['News', 'Written'] | found | derived | 487 |
| | TweetTopicSingleClassification Antypas et al. (2022) | ['eng'] | ['Social', 'News', 'Written'] | found | expert-annotated | 1693 |
| Clustering | AlloProfClusteringS2S.v2 Lefebvre-Brossard et al. (2023) | ['fra'] | ['Encyclopaedic', 'Written'] | found | human-annotated | 2556 |
| | ArXivHierarchicalClusteringP2P | ['eng'] | ['Academic', 'Written'] | found | derived | 2048 |
| | ArXivHierarchicalClusteringS2S | ['eng'] | ['Academic', 'Written'] | found | derived | 2048 |
| | BigPatentClustering.v2 Sharma et al. (2019) | ['eng'] | ['Legal', 'Written'] | found | derived | 2048 |
| | BiorxivClusteringP2P.v2 | ['eng'] | ['Academic', 'Written'] | created | derived | 53787 |
| | CLSClusteringP2P.v2 Li et al. (2022) | ['cmn'] | ['Academic', 'Written'] | found | derived | 2048 |
| | HALClusteringS2S.v2 Ciancone et al. (2024) | ['fra'] | ['Academic', 'Written'] | found | human-annotated | 2048 |
| | MasakhaNEWSClusteringS2S Adelani et al. (2023b) | ['amh', 'eng', 'fra', ...] | None | | | 80 |
| | MedrxivClusteringP2P.v2 | ['eng'] | ['Academic', 'Medical', 'Written'] | created | derived | 37500 |
| | PlscClusteringP2P.v2 | ['pol'] | ['Academic', 'Written'] | found | derived | 2048 |
| | RomaniBibleClustering | ['rom'] | ['Religious', 'Written'] | human-translated and localized | derived | |
| | SIB200ClusteringS2S Adelani et al. (2023a) | ['ace', 'acm', 'acq', ...] | ['News', 'Written'] | human-translated and localized | expert-annotated | 197788 |
| | SNLHierarchicalClusteringP2P Navjord & Korsvik (2023) | ['nob'] | ['Encyclopaedic', 'Non-fiction', 'Written'] | found | derived | 1300 |
| | StackExchangeClustering.v2 Geigle et al. (2021) | ['eng'] | ['Web', 'Written'] | found | derived | 2048 |
| | SwednClusteringP2P Monsen & J'onsson (2021) | ['swe'] | ['News', 'Non-fiction', 'Written'] | found | derived | 68752 |
| | WikiCitiesClustering Foundation | ['eng'] | ['Encyclopaedic', 'Written'] | found | derived | |
| | WikiClusteringP2P.v2 | ['bos', 'cat', 'ces', ...] | ['Encyclopaedic', 'Written'] | created | derived | 28672 |

Table 12: The tasks included in MTEB(Multilingual) (part 1).

## H.3 PERFORMANCE ON MTEB(eng, v2)

Table 18 show the performance of our representative set of model on MTEB(eng, v2).

## H.4 PERFORMANCE ON MTEB(Code)

Table 19 show the performance of our representative set of model on MTEB(Code).

| Type | Name | Languages | Domains | Sample creators | Annotations creators | Nb samples* |
|---|---|---|---|---|---|---|
| InstructionReranking | Core17InstructionRetrieval Weller et al. (2024) | ['eng'] | ['News', 'Written'] | found | derived | 19939 |
| | News21InstructionRetrieval Weller et al. (2024) | ['eng'] | ['News', 'Written'] | found | derived | 30985 |
| | Robust04InstructionRetrieval Weller et al. (2024) | ['eng'] | ['News', 'Written'] | found | derived | 47596 |
| MultilabelClassification | BrazilianToxicTweetsClassification Leite et al. (2020) | ['por'] | ['Constructed', 'Written'] | found | expert-annotated | 2048 |
| | CEDRClassification Sboev et al. (2021) | ['rus'] | ['Web', 'Social', 'Blog', ...] | found | human-annotated | 1882 |
| | KorHateSpeechMLClassification Lee et al. (2022) | ['kor'] | ['Social', 'Written'] | found | expert-annotated | 2037 |
| | MalteseNewsClassification Chaudhary et al. (2024) | ['mlt'] | ['Constructed', 'Written'] | found | expert-annotated | 2297 |
| | MultiEURLEXMultilabelClassification Chalkidis et al. (2021) | ['bul', 'ces', 'dan', ...] | ['Legal', 'Government', 'Written'] | found | expert-annotated | 115000 |
| PairClassification | ArmenianParaphrasePC Malajyan et al. (2020) | ['hye'] | ['News', 'Written'] | found | derived | 1470 |
| | CTKFactsNLI Ullrich et al. (2023) | ['ces'] | ['News', 'Written'] | found | human-annotated | 680 |
| | OpusparcusPC Creutz (2018) | ['deu', 'eng', 'fin', ...] | ['Spoken', 'Spoken'] | created | human-annotated | 18207 |
| | PawsXPairClassification Yang et al. (2019) | ['cmn', 'deu', 'eng', ...] | ['Web', 'Encyclopaedic', 'Written'] | human-translated | human-annotated | 28000 |
| | PpcPC Dadas (2022) | ['pol'] | ['Fiction', 'Non-fiction', 'Web', ...] | found | derived | 1000 |
| | RTE3 Giampiccolo et al. (2007) | ['deu', 'eng', 'fra', ...] | ['News', 'Web', 'Encyclopaedic', ...] | found | expert-annotated | 1923 |
| | SprintDuplicateQuestions Shah et al. (2018) | ['eng'] | ['Programming', 'Written'] | found | derived | 101000 |
| | TERRa Shavrina et al. (2020) | ['rus'] | ['News', 'Web', 'Written'] | found | human-annotated | 307 |
| | TwitterURLCorpus Lan et al. (2017) | ['eng'] | ['Social', 'Written'] | found | derived | 51534 |
| | XNLI Conneau et al. (2018) | ['ara', 'bul', 'deu', ...] | ['Non-fiction', 'Fiction', 'Government', ...] | created | expert-annotated | 38220 |
| | indonli Mahendra et al. (2021) | ['ind'] | ['Encyclopaedic', 'Web', 'News', ...] | found | expert-annotated | 2040 |
| Reranking | AlloprofReranking Lefebvre-Brossard et al. (2023) | ['fra'] | ['Web', 'Academic', 'Written'] | found | expert-annotated | 27355 |
| | RuBQReranking Rybin et al. (2021) | ['rus'] | ['Encyclopaedic', 'Written'] | created | human-annotated | 38998 |
| | T2Reranking Xie et al. (2023) | ['cmn'] | None | | | 103330 |
| | VoyageMMarcoReranking Clavié (2023) | ['jpn'] | ['Academic', 'Non-fiction', 'Written'] | found | derived | 55423 |
| | WebLINXCandidatesReranking Lù et al. (2024) | ['eng'] | ['Academic', 'Web', 'Written'] | created | expert-annotated | 5592142 |
| | WikipediaRerankingMultilingual Foundation | ['ben', 'bul', 'ces', ...] | ['Encyclopaedic', 'Written'] | LM-generated and verified | LM-generated and reviewed | 240000 |
| Retrieval | AILAStatutes Bhattacharya et al. (2020) | ['eng'] | ['Legal', 'Written'] | found | derived | 82 - 50 |
| | ArguAna Boteva et al. (2016) | ['eng'] | ['Medical', 'Written'] | | | 8674 - 1406 |
| | BelebeleRetrieval Bandarkar et al. (2023) | ['acm', 'afr', 'als', ...] | ['Web', 'News', 'Written'] | created | expert-annotated | 183488 - 338378 |
| | CUREv1 | ['eng', 'fra', 'spa'] | ['Medical', 'Academic', 'Written'] | created | expert-annotated | 1541613 - 12000 |
| | CovidRetrieval | ['cmn'] | None | | | 100001 - 949 |
| | HagridRetrieval Kamalloo et al. (2023) | ['eng'] | ['Encyclopaedic', 'Written'] | found | expert-annotated | 496 - 496 |
| | LEMBPasskeyRetrieval Zhu et al. (2024) | ['eng'] | ['Fiction', 'Written'] | found | derived | 800 - 400 |
| | LegalBenchCorporateLobbying Guha et al. (2023) | ['eng'] | ['Legal', 'Written'] | found | derived | 319 - 340 |
| | MIRACLRetrievalHardNegatives Zhang et al. (2023) | ['ara', 'ben', 'deu', ...] | ['Encyclopaedic', 'Written'] | created | expert-annotated | 2449382 - 11076 |
| | MLQARetrieval Lewis et al. (2019) | ['ara', 'deu', 'eng', ...] | ['Encyclopaedic', 'Written'] | found | human-annotated | 152379 - 173776 |
| | SCIDOCS Cohan et al. (2020b) | ['eng'] | ['Academic', 'Written', 'Non-fiction'] | | | 25657 - 1000 |
| | SpartQA Xiao et al. (2024a) | ['eng'] | ['Encyclopaedic', 'Written'] | found | derived | 1592 - 3594 |
| | StackOverflowQA Li et al. (2024) | ['eng'] | ['Programming', 'Written'] | found | derived | 19931 - 1994 |
| | StatcanDialogueDatasetRetrieval Lu et al. (2023) | ['eng', 'fra'] | ['Government', 'Web', 'Written'] | found | derived | 23628 - 9436 |
| | TRECCOVID Roberts et al. (2021) | ['eng'] | ['Medical', 'Academic', 'Written'] | | | 171332 - 50 |
| | TempReasonL1 Xiao et al. (2024a) | ['eng'] | ['Encyclopaedic', 'Written'] | found | derived | 12504 - 4000 |
| | TwitterHjerneRetrieval Holm (2024) | ['dan'] | ['Social', 'Written'] | found | derived | 262 - 78 |
| | WikipediaRetrievalMultilingual | ['ben', 'bul', 'ces', ...] | ['Encyclopaedic', 'Written'] | LM-generated and verified | LM-generated and reviewed | 216000 - 24000 |
| | WinoGrande Xiao et al. (2024a) | ['eng'] | ['Encyclopaedic', 'Written'] | found | derived | 5095 - 1267 |
| STS | FaroeseSTS Snæbjarnarson et al. (2023) | ['fao'] | ['News', 'Web', 'Written'] | found | human-annotated | 729 |
| | FinParaSTS Kanerva et al. (2021) | ['fin'] | ['News', 'Subtitles', 'Written'] | found | expert-annotated | 2000 |
| | GermanSTSBenchmark May (2021) | ['deu'] | None | | | 2879 |
| | IndicCrosslingualSTS Ramesh et al. (2022) | ['asm', 'ben', 'eng', ...] | ['News', 'Non-fiction', 'Web', ...] | created | expert-annotated | 3072 |
| | JSICK Yanaka & Mineshima (2022) | ['jpn'] | ['Web', 'Written'] | found | human-annotated | 1986 |
| | SICK-R Marelli et al. (2014) | ['eng'] | ['Web', 'Written'] | | human-annotated | 9927 |
| | STS12 Agirre et al. (2012) | ['eng'] | ['Encyclopaedic', 'News', 'Written'] | created | human-annotated | 3108 |
| | STS13 Agirre et al. (2013) | ['eng'] | ['Web', 'News', 'Non-fiction', ...] | created | human-annotated | 1500 |
| | STS14 Bandhakavi et al. (2014) | ['eng'] | ['Blog', 'Web', 'Spoken'] | created | derived | 3750 |
| | STS15 Biçici (2015) | ['eng'] | ['Blog', 'News', 'Web', ...] | created | human-annotated | 3000 |
| | STS17 Cer et al. (2017) | ['ara', 'deu', 'eng', ...] | ['News', 'Web', 'Written'] | created | human-annotated | 5346 |
| | STS22.v2 Chen et al. (2022) | ['ara', 'cmn', 'deu', ...] | ['News', 'Written'] | found | human-annotated | 3958 |
| | STSB Xiao et al. (2024b) | ['cmn'] | None | | | 2819 |
| | STSBenchmark May (2021) | ['eng'] | ['Blog', 'News', 'Written'] | machine-translated and verified | human-annotated | 1379 |
| | STSES Agirre et al. (2015) | ['spa'] | ['Written'] | | | 155 |
| | SemRel24STS Ousidhoum et al. (2024) | ['afr', 'amh', 'arb', ...] | ['Spoken', 'Written'] | created | human-annotated | 7498 |

Table 13: The tasks included in MTEB(Multilingual) (part 2). *For the number of samples, are given the total number of samples all languages included, for Retrieval tasks are given the (number of queries - number of documents).

| Type | Name | Languages | Domains | Sample creation | Annotation creators | Nb Samples* |
|---|---|---|---|---|---|---|
| BitextMining | BornholmBitextMining Derczynski & Kjeldsen | ['dan'] | ['Web', 'Social', 'Fiction', ...] | created | expert-annotated | 500 |
| | BibleNLPBitextMining Akerman et al. (2023) | ['aai', 'aak', 'aau', ...] | ['Religious', 'Written'] | created | expert-annotated | 417452 |
| | BUCC.v2 Zweigenbaum et al. (2017) | ['cmn', 'deu', 'eng', ...] | ['Written'] | human-translated | human-annotated | 35000 |
| | DiaBlaBitextMining González et al. (2019) | ['eng', 'fra'] | ['Social', 'Written'] | created | human-annotated | 11496 |
| | FloresBitextMining Goyal et al. (2022) | ['ace', 'acm', 'acq', ...] | ['Non-fiction', 'Encyclopaedic', 'Written'] | created | human-annotated | 41909944 |
| | NorwegianCourtsBitextMining Tiedemann & Thottingal (2020) | ['nno', 'nob'] | ['Legal', 'Written'] | found | human-annotated | 228 |
| | NTREXBitextMining Federmann et al. (2022) | ['afr', 'amh', 'arb', ...] | ['News', 'Written'] | human-translated and localized | expert-annotated | 3826252 |
| Classification | BulgarianStoreReviewSentimentClassfication Georgieva-Trifonova et al. (2018) | ['bul'] | ['Reviews', 'Written'] | found | human-annotated | 182 |
| | CzechProductReviewSentimentClassification Habernal et al. (2013) | ['ces'] | ['Reviews', 'Written'] | found | derived | 2048 |
| | GreekLegalCodeClassification Papaloukas et al. (2021) | ['ell'] | ['Legal', 'Written'] | found | human-annotated | 4096 |
| | DBpediaClassification Zhang et al. (2015) | ['eng'] | ['Encyclopaedic', 'Written'] | found | derived | 2048 |
| | FinancialPhrasebankClassification Malo et al. (2014) | ['eng'] | ['News', 'Written', 'Financial'] | found | expert-annotated | 2264 |
| | PoemSentimentClassification Sheng & Uthus (2020) | ['eng'] | ['Reviews', 'Written'] | found | human-annotated | 209 |
| | ToxicChatClassification Lin et al. (2023) | ['eng'] | ['Constructed', 'Written'] | found | expert-annotated | 1164 |
| | ToxicConversationsClassification cjadams et al. (2019) | ['eng'] | ['Social', 'Written'] | found | human-annotated | 2048 |
| | EstonianValenceClassification Pajupuu et al. (2023) | ['est'] | ['News', 'Written'] | found | human-annotated | 818 |
| | ItaCaseholdClassification Licari et al. (2023) | ['ita'] | ['Legal', 'Government', 'Written'] | found | human-annotated | 221 |
| | AmazonCounterfactualClassification O'Neill et al. (2021) | ['deu', 'eng', 'jpn'] | ['Reviews', 'Written'] | found | human-annotated | 5805 |
| | MassiveScenarioClassification FitzGerald et al. (2022) | ['afr', 'amh', 'ara', ...] | ['Spoken'] | human-translated and localized | human-annotated | 255357 |
| | MultiHateClassification Röttger et al. (2021) | ['ara', 'cmn', 'deu', ...] | ['Constructed', 'Written'] | created | expert-annotated | 11000 |
| | NordicLangClassification Haas & Derczynski (2021) | ['dan', 'fao', 'isl', ...] | ['Encyclopaedic'] | found | derived | 3000 |
| | ScalaClassification Nielsen (2023) | ['dan', 'nno', 'nob', ...] | ['Fiction', 'News', 'Non-fiction', ...] | created | human-annotated | 8192 |
| | SwissJudgementClassification Niklaus et al. (2022) | ['deu', 'fra', 'ita'] | ['Legal', 'Written'] | found | expert-annotated | 4908 |
| | TweetSentimentClassification Barbieri et al. (2022) | ['ara', 'deu', 'eng', ...] | ['Social', 'Written'] | found | human-annotated | 2048 |
| | CBD Ogrodniczuk & Łukasz Kobyliński (2019) | ['pol'] | ['Written', 'Social'] | found | human-annotated | 1000 |
| | PolEmo2.0-OUT | ['pol'] | ['Written', 'Social'] | | | 494 |
| | CSFDSKMovieReviewSentimentClassification Štefánik et al. (2023) | ['slk'] | ['Reviews', 'Written'] | found | derived | 2048 |
| | DalajClassification Volodina et al. (2021) | ['swe'] | ['Non-fiction', 'Written'] | created | expert-annotated | 888 |
| Clustering | WikiCitiesClustering Foundation | ['eng'] | ['Encyclopaedic', 'Written'] | found | derived | 1 |
| | RomaniBibleClustering | ['rom'] | ['Religious', 'Written'] | human-translated and localized | derived | 4 |
| | BigPatentClustering.v2 Sharma et al. (2019) | ['eng'] | ['Legal', 'Written'] | found | derived | 2048 |
| | BiorxivClusteringP2P.v2 | ['eng'] | ['Academic', 'Written'] | created | derived | 53787 |
| | AlloProfClusteringS2S.v2 Lefebvre-Brossard et al. (2023) | ['fra'] | ['Encyclopaedic', 'Written'] | found | human-annotated | 2556 |
| | HALClusteringS2S.v2 Ciancone et al. (2024) | ['fra'] | ['Academic', 'Written'] | found | human-annotated | 2048 |
| | SIB200ClusteringS2S Adelani et al. (2023a) | ['ace', 'acm', 'acq', ...] | ['News', 'Written'] | human-translated and localized | expert-annotated | 197788 |
| | WikiClusteringP2P.v2 | ['bos', 'cat', 'ces', ...] | ['Encyclopaedic', 'Written'] | created | derived | 28672 |
| Retrieval | StackOverflowQA Li et al. (2024) | ['eng'] | ['Programming', 'Written'] | found | derived | 19931 - 1994 |
| | TwitterHjerneRetrieval Holm (2024) | ['dan'] | ['Social', 'Written'] | found | derived | 262 - 78 |
| | LegalQuAD Hoppe et al. (2021) | ['deu'] | ['Legal', 'Written'] | found | derived | 200 - 200 |
| | ArguAna Boteva et al. (2016) | ['eng'] | ['Medical', 'Written'] | | | 8674 - 1406 |
| | HagridRetrieval Kamalloo et al. (2023) | ['eng'] | ['Encyclopaedic', 'Written'] | found | expert-annotated | 496 - 496 |
| | LegalBenchCorporateLobbying Guha et al. (2023) | ['eng'] | ['Legal', 'Written'] | found | derived | 319 - 340 |
| | LEMBPasskeyRetrieval Zhu et al. (2024) | ['eng'] | ['Fiction', 'Written'] | found | derived | 800 - 400 |
| | SCIDOCS Cohan et al. (2020b) | ['eng'] | ['Academic', 'Written', 'Non-fiction'] | found | derived | 25657 - 1000 |
| | SpartQA Xiao et al. (2024a) | ['eng'] | ['Encyclopaedic', 'Written'] | found | derived | 1592 - 3594 |
| | TempReasonL1 Xiao et al. (2024a) | ['eng'] | ['Encyclopaedic', 'Written'] | found | derived | 12504 - 4000 |
| | WinoGrande Xiao et al. (2024a) | ['eng'] | ['Encyclopaedic', 'Written'] | found | derived | 5095 - 1267 |
| | AlloprofRetrieval Lefebvre-Brossard et al. (2023) | ['fra'] | ['Encyclopaedic', 'Written'] | found | human-annotated | 2556 - 2316 |
| | BelebeleRetrieval Bandarkar et al. (2023) | ['acm', 'afr', 'als', ...] | ['Web', 'News', 'Written'] | created | expert-annotated | 183488 - 338378 |
| | StatcanDialogueDatasetRetrieval Lu et al. (2023) | ['eng', 'fra'] | ['Government', 'Web', 'Written'] | found | derived | 23628 - 9436 |
| | WikipediaRetrievalMultilingual | ['ben', 'bul', 'ces', ...] | ['Encyclopaedic', 'Written'] | LM-generated and verified | LM-generated and reviewed | 216000 - 24000 |
| InstructionReranking | Core17InstructionRetrieval Weller et al. (2024) | ['eng'] | ['News', 'Written'] | found | derived | 19939 |
| | News21InstructionRetrieval Weller et al. (2024) | ['eng'] | ['News', 'Written'] | found | derived | 30985 |
| | Robust04InstructionRetrieval Weller et al. (2024) | ['eng'] | ['News', 'Written'] | found | derived | 47596 |
| MultilabelClassification | MalteseNewsClassification Chaudhary et al. (2024) | ['mlt'] | ['Constructed', 'Written'] | found | expert-annotated | 2297 |
| | MultiEURLEXMultilabelClassification Chalkidis et al. (2021) | ['bul', 'ces', 'dan', ...] | ['Legal', 'Government', 'Written'] | found | derived | 115000 |
| PairClassification | CTKFactsNLI Ullrich et al. (2023) | ['ces'] | ['News', 'Written'] | found | human-annotated | 680 |
| | SprintDuplicateQuestions Shah et al. (2018) | ['eng'] | ['Programming', 'Written'] | found | derived | 101000 |
| | OpusparcusPC Creutz (2018) | ['deu', 'eng', 'fin', ...] | ['Spoken', 'Spoken'] | created | human-annotated | 18207 |
| | RTE3 Giampiccolo et al. (2007) | ['deu', 'eng', 'fra', ...] | ['News', 'Web', 'Encyclopaedic', ...] | found | expert-annotated | 1923 |
| | XNLI Conneau et al. (2018) | ['ara', 'bul', 'deu', ...] | ['Non-fiction', 'Fiction', 'Government', ...] | created | expert-annotated | 38220 |
| | PSC Ogrodniczuk & Kopeć (2014) | ['pol'] | ['News', 'Written'] | found | derived | 1078 |
| Reranking | WebLINXCandidatesReranking Lù et al. (2024) | ['eng'] | ['Academic', 'Web', 'Written'] | found | expert-annotated | 9592142 |
| | AlloprofReranking Lefebvre-Brossard et al. (2023) | ['fra'] | ['Web', 'Academic', 'Written'] | found | human-annotated | 27355 |
| | WikipediaRerankingMultilingual Foundation | ['ben', 'bul', 'ces', ...] | ['Encyclopaedic', 'Written'] | LM-generated and verified | LM-generated and reviewed | 240000 |
| STS | SICK-R Marelli et al. (2014) | ['eng'] | ['Web', 'Written'] | | human-annotated | 9927 |
| | STS12 Agirre et al. (2012) | ['eng'] | ['Encyclopaedic', 'News', 'Written'] | created | human-annotated | 3108 |
| | STS14 Bandhakavi et al. (2014) | ['eng'] | ['Blog', 'Web', 'Spoken'] | created | human-annotated | 3750 |
| | STS15 Biçici (2015) | ['eng'] | ['Blog', 'News', 'Web', ...] | created | human-annotated | 3000 |
| | STSBenchmark May (2021) | ['eng'] | ['Blog', 'News', 'Written'] | machine-translated and verified | human-annotated | 1379 |
| | FinParaSTS Kanerva et al. (2021) | ['fin'] | ['News', 'Subtitles', 'Written'] | found | expert-annotated | 2000 |
| | STS17 Cer et al. (2017) | ['ara', 'deu', 'eng', ...] | ['News', 'Web', 'Written'] | created | human-annotated | 5346 |
| | SICK-R-PL Dadas et al. (2020) | ['pol'] | ['Web', 'Written'] | human-translated and localized | human-annotated | 4906 |
| | STSES Agirre et al. (2015) | ['spa'] | ['Written'] | | | 155 |

Table 14: The tasks included in MTEB(Europe). The language column shows all the languages of the task. When running the tasks we limit it to the languages specified in the benchmark. * For the number of samples, are given the total number of samples all languages included, for Retrieval tasks are given the (number of queries - number of documents).

| Type | Name | Languages | Domains | Sample creation | Annotation creators | Nb samples* |
|---|---|---|---|---|---|---|
| BitextMining | IN22ConvBitextMining Gala et al. (2023) | ['asm', 'ben', 'brx', ...] | ['Social', 'Spoken', 'Fiction', ...] | created | expert-annotated | 760518 |
| | IN22GenBitextMining Gala et al. (2023) | ['asm', 'ben', 'brx', ...] | ['Web', 'Legal', 'Government', ...] | created | expert-annotated | 518144 |
| | IndicGenBenchFloresBitextMining Singh et al. (2024a) | ['asm', 'awa', 'ben', ...] | ['Web', 'News', 'Written'] | human-translated and localized | expert-annotated | 116522 |
| | LinceMTBitextMining Aguilar et al. (2020) | ['eng', 'hin'] | ['Social', 'Written'] | found | human-annotated | 8059 |
| Classification | BengaliSentimentAnalysis Sazzed (2020) | ['ben'] | ['Reviews', 'Written'] | found | human-annotated | 2048 |
| | GujaratiNewsClassification | ['guj'] | ['News', 'Written'] | found | derived | 1318 |
| | HindiDiscourseClassification Dhanwal et al. (2020) | ['hin'] | ['Fiction', 'Social', 'Written'] | found | expert-annotated | 2048 |
| | IndicLangClassification Madhani et al. (2023) | ['asm', 'ben', 'brx', ...] | ['Web', 'Non-fiction', 'Written'] | created | expert-annotated | 30418 |
| | MTOPIntentClassification Li et al. (2021) | ['deu', 'eng', 'fra', ...] | ['Spoken', 'Spoken'] | created | human-annotated | 30517 |
| | MalayalamNewsClassification Kunchukuttan et al. (2020) | ['mal'] | ['News', 'Written'] | found | derived | 1260 |
| | MultiHateClassification Röttger et al. (2021) | ['ara', 'cmn', 'deu', ...] | ['Constructed', 'Written'] | created | expert-annotated | 11000 |
| | NepaliNewsClassification Arora (2020) | ['nep'] | ['News', 'Written'] | found | derived | 2048 |
| | PunjabiNewsClassification Kunchukuttan et al. (2020) | ['pan'] | ['News', 'Written'] | found | derived | 157 |
| | SanskritShlokasClassification Arora (2020) | ['san'] | ['Religious', 'Written'] | found | derived | 479 |
| | SentimentAnalysisHindi Parida et al. (2023) | ['hin'] | ['Reviews', 'Written'] | found | derived | 2048 |
| | TweetSentimentClassification Barbieri et al. (2022) | ['ara', 'deu', 'eng', ...] | ['Social', 'Written'] | found | human-annotated | 2048 |
| | UrduRomanSentimentClassification Sharf (2018) | ['urd'] | ['Social', 'Written'] | found | derived | 2048 |
| Clustering | SIB200ClusteringS2S Adelani et al. (2023a) | ['ace', 'acm', 'acq', ...] | ['News', 'Written'] | human-translated and localized | expert-annotated | 197788 |
| PairClassification | XNLI Conneau et al. (2018) | ['ara', 'bul', 'deu', ...] | ['Non-fiction', 'Fiction', 'Government', ...] | created | expert-annotated | 38220 |
| Reranking | WikipediaRerankingMultilingual Foundation | ['ben', 'bul', 'ces', ...] | ['Encyclopaedic', 'Written'] | LM-generated and verified | LM-generated and reviewed | 240000 |
| Retrieval | BelebeleRetrieval Bandarkar et al. (2023) | ['acm', 'afr', 'als', ...] | ['Web', 'News', 'Written'] | created | expert-annotated | 183488 - 338378 |
| | XQuADRetrieval Artetxe et al. (2019) | ['arb', 'deu', 'ell', ...] | ['Web', 'Written'] | created | human-annotated | 2880 - 14199 |
| STS | IndicCrosslingualSTS Ramesh et al. (2022) | ['asm', 'ben', 'eng', ...] | ['News', 'Non-fiction', 'Web', ...] | created | expert-annotated | 3072 |

Table 15: The tasks included in MTEB(Indic). The language column shows all the languages of the task. When running the tasks we limit it to the Indic languages specified in the benchmark. * For the number of samples, are given the total number of samples all languages included, for Retrieval tasks are given the (number of queries - number of documents).

| Type | Name | Languages | Domains | Sample creation | Annotation creators | Nb samples* |
|---|---|---|---|---|---|---|
| Classification | AmazonCounterfactualClassification O'Neill et al. (2021) | ['deu', 'eng', 'jpn'] | ['Reviews', 'Written'] | found | human-annotated | 5805 |
| | Banking77Classification Casanueva et al. (2020) | ['eng'] | ['Written'] | found | human-annotated | 3080 |
| | ImdbClassification Maas et al. (2011) | ['eng'] | ['Reviews', 'Written'] | found | human-annotated | 25000 |
| | MTOPDomainClassification Li et al. (2021) | ['deu', 'eng', 'fra', ...] | ['Spoken', 'Spoken'] | created | human-annotated | 30517 |
| | MassiveIntentClassification FitzGerald et al. (2022) | ['afr', 'amh', 'ara', ...] | ['Spoken'] | human-translated and localized | human-annotated | 255357 |
| | MassiveScenarioClassification FitzGerald et al. (2022) | ['afr', 'amh', 'ara', ...] | ['Spoken'] | human-translated and localized | human-annotated | 255357 |
| | ToxicConversationsClassification cjadams et al. (2019) | ['eng'] | ['Social', 'Written'] | found | human-annotated | 2048 |
| | TweetSentimentExtractionClassification Maggie (2020) | ['eng'] | ['Social', 'Written'] | found | human-annotated | 3534 |
| Clustering | ArXivHierarchicalClusteringP2P | ['eng'] | ['Academic', 'Written'] | found | derived | 2048 |
| | ArXivHierarchicalClusteringS2S | ['eng'] | ['Academic', 'Written'] | found | derived | 2048 |
| | BiorxivClusteringP2P.v2 | ['eng'] | ['Academic', 'Written'] | created | derived | 53787 |
| | MedrxivClusteringP2P.v2 | ['eng'] | ['Academic', 'Medical', 'Written'] | created | derived | 37500 |
| | MedrxivClusteringS2S.v2 | ['eng'] | ['Academic', 'Medical', 'Written'] | created | derived | 37500 |
| | StackExchangeClustering.v2 Geigle et al. (2021) | ['eng'] | ['Web', 'Written'] | found | derived | 2048 |
| | StackExchangeClusteringP2P.v2 Geigle et al. (2021) | ['eng'] | ['Web', 'Written'] | found | derived | 74914 |
| | TwentyNewsgroupsClustering.v2 Lang (1995) | ['eng'] | ['News', 'Written'] | found | derived | 59545 |
| PairClassification | SprintDuplicateQuestions Shah et al. (2018) | ['eng'] | ['Programming', 'Written'] | found | derived | 101000 |
| | TwitterSemEval2015 Xu et al. (2015) | ['eng'] | ['Social', 'Written'] | found | human-annotated | 16777 |
| | TwitterURLCorpus Lan et al. (2017) | ['eng'] | ['Social', 'Written'] | found | derived | 51534 |
| Reranking | AskUbuntuDupQuestions Wang et al. (2021a) | ['eng'] | ['Programming', 'Web'] | found | human-annotated | 7581 |
| | MindSmallReranking Wu et al. (2020a) | ['eng'] | ['News', 'Written'] | found | expert-annotated | 2367791 |
| Retrieval | ArguAna Boteva et al. (2016) | ['eng'] | ['Medical', 'Written'] | | derived | 8674 - 1406 |
| | CQADupstackGamingRetrieval Hoogeveen et al. (2015) | ['eng'] | ['Web', 'Written'] | found | derived | 45301 - 1595 |
| | CQADupstackUnixRetrieval Hoogeveen et al. (2015) | ['eng'] | ['Written', 'Web', 'Programming'] | found | derived | 47382 - 1072 |
| | ClimateFEVERHardNegatives Diggelmann et al. (2021) | ['eng'] | ['Encyclopaedic', 'Written'] | found | human-annotated | 47416 - 1000 |
| | FEVERHardNegatives Thorne et al. (2018a) | ['eng'] | None | | | 163698 - 1000 |
| | FiQA2018 Thakur et al. (2021) | ['eng'] | ['Written', 'Financial'] | found | human-annotated | 57638 - 648 |
| | HotpotQAHardNegatives Yang et al. (2018) | ['eng'] | ['Web', 'Written'] | found | human-annotated | 225621 - 1000 |
| | SCIDOCS Cohan et al. (2020b) | ['eng'] | ['Academic', 'Written', 'Non-fiction'] | found | | 25657 - 1000 |
| | TRECCOVID Roberts et al. (2021) | ['eng'] | ['Medical', 'Academic', 'Written'] | | | 171332 - 50 |
| | Touche2020Retrieval.v3 Thakur et al. (2024) | ['eng'] | ['Academic'] | found | human-annotated | 303732 - 49 |
| STS | BIOSSES Soğancıoğlu et al. (2017) | ['eng'] | ['Medical'] | found | derived | 100 |
| | SICK-R Marelli et al. (2014) | ['eng'] | ['Web', 'Written'] | | human-annotated | 9927 |
| | STS12 Agirre et al. (2012) | ['eng'] | ['Encyclopaedic', 'News', 'Written'] | created | human-annotated | 3108 |
| | STS13 Agirre et al. (2013) | ['eng'] | ['Web', 'News', 'Non-fiction', ...] | created | human-annotated | 1500 |
| | STS14 Bandhakavi et al. (2014) | ['eng'] | ['Blog', 'Web', 'Spoken'] | created | derived | 3750 |
| | STS15 Biçici (2015) | ['eng'] | ['Blog', 'News', 'Web', ...] | created | human-annotated | 3000 |
| | STS17 Cer et al. (2017) | ['ara', 'deu', 'eng', ...] | ['News', 'Web', 'Written'] | created | human-annotated | 5346 |
| | STS22.v2 Chen et al. (2022) | ['ara', 'cmn', 'deu', ...] | ['News', 'Written'] | found | human-annotated | 3958 |
| | STSBenchmark May (2021) | ['eng'] | ['Blog', 'News', 'Written'] | machine-translated and verified | human-annotated | 1379 |
| Summarization | SummEvalSummarization.v2 Fabbri et al. (2020) | ['eng'] | ['News', 'Written'] | created | human-annotated | 100 |

Table 16: The tasks included in MTEB(eng, v2). The language column shows all the languages of the task. When running the tasks we limit it to the languages specified in the benchmark. * For the number of samples, are given the total number of samples all languages included, for Retrieval tasks are given the (number of queries - number of documents).

| Type | Name | Languages | Domains | Sample creation | Annotations creators | Nb Samples* |
|---|---|---|---|---|---|---|
| Retrieval | AppsRetrieval Hendrycks et al. (2021a) | ['eng', 'python'] | ['Programming', 'Written'] | found | derived | 3765 - 8765 |
| | COIRCodeSearchNetRetrieval Husain et al. (2019) | ['go', 'java', 'javascript', 'php'] | ['Programming', 'Written'] | found | derived | 52561 - 1003765 |
| | CodeEditSearchRetrieval Muennighoff et al. (2023a) | ['c', 'c++', 'go', 'java'] | ['Programming', 'Written'] | found | derived | 13000 - 13000 |
| | CodeFeedbackMT Zheng et al. (2024) | ['eng'] | ['Programming', 'Written'] | found | derived | 13277 - 66383 |
| | CodeFeedbackST Li et al. (2024) | ['eng'] | ['Programming', 'Written'] | found | derived | 31306 - 156526 |
| | CodeSearchNetCCRetrieval Li et al. (2024) | ['go', 'java', 'javascript', 'php'] | ['Programming', 'Written'] | found | derived | 52561 - 1005474 |
| | CodeSearchNetRetrieval Husain et al. (2019) | ['go', 'java', 'javascript', 'php'] | ['Programming', 'Written'] | found | derived | 6000 - 6000 |
| | CodeTransOceanContest Yan et al. (2023) | ['c++', 'python'] | ['Programming', 'Written'] | found | derived | 221 - 1008 |
| | CodeTransOceanDL Yan et al. (2023) | ['python'] | ['Programming', 'Written'] | found | derived | 180 - 816 |
| | CosQA Huang et al. (2021) | ['eng', 'python'] | ['Programming', 'Written'] | found | derived | 500 - 20604 |
| | StackOverflowQA Li et al. (2024) | ['eng'] | ['Programming', 'Written'] | found | derived | 1994 - 19931 |
| | SyntheticText2SQL Meyer et al. (2024) | ['eng', 'sql'] | ['Programming', 'Written'] | found | derived | 5851 - 105851 |

Table 17: The tasks included in MTEB(Code). * For the number of samples, are given the total number of samples all languages included, for Retrieval tasks are given the (number of queries - number of documents).

| model | Rank Borda Count | Average Across All | Category | Average by Category Pair Clf. | Clf. | STS | Retrieval | Clustering | Reranking |
|---|---|---|---|---|---|---|---|---|---|
| e5-mistral-7b-instruct | 1 (393) | 67.0 | 67.2 | 88.4 | 75.2 | 83.6 | 54.8 | 51.4 | 49.8 |
| GritLM-7B | 2 (384) | 66.4 | 66.7 | 87.3 | 77.0 | 82.5 | 53.2 | 50.8 | 49.6 |
| multilingual-e5-large-instruct | 3 (357) | 65.2 | 65.6 | 86.2 | 73.2 | 84.3 | 51.0 | 49.9 | 48.7 |
| multilingual-e5-large | 4 (270) | 62.1 | 62.4 | 84.7 | 72.8 | 80.6 | 49.0 | 42.8 | 44.7 |
| all-mpnet-base-v2 | 5 (211) | 56.0 | 58.1 | 83.0 | 56.6 | 72.2 | 41.9 | 46.6 | 48.4 |
| multilingual-e5-base | 6 (211) | 60.2 | 60.9 | 83.6 | 70.0 | 79.1 | 46.1 | 42.2 | 44.3 |
| paraphrase-multilingual-mpnet-base-v2 | 7 (188) | 57.3 | 58.8 | 81.7 | 68.6 | 79.8 | 34.1 | 43.5 | 45.2 |
| all-MiniLM-L12-v2 | 8 (172) | 54.7 | 57.0 | 82.5 | 55.8 | 70.7 | 40.7 | 44.6 | 47.5 |
| all-MiniLM-L6-v2 | 9 (149) | 54.4 | 56.7 | 82.4 | 55.4 | 70.4 | 39.8 | 44.9 | 47.1 |
| multilingual-e5-small | 10 (147) | 58.4 | 59.3 | 82.7 | 67.7 | 77.6 | 43.7 | 40.8 | 43.2 |
| paraphrase-multilingual-MiniLM-L12-v2 | 11 (109) | 55.1 | 57.0 | 80.0 | 64.4 | 77.5 | 32.8 | 41.7 | 45.4 |
| LaBSE | 12 (49) | 48.6 | 51.7 | 78.9 | 66.8 | 70.2 | 16.8 | 36.1 | 41.3 |

Table 18: Performance on MTEB(eng, v2) across task categories.

| Model | Rank Borda Count | Average Across All | Average by Language C++ | Go | Java | JavaScript | PHP | Python | Ruby |
|---|---|---|---|---|---|---|---|---|---|
| GritLM-7B | 1 (88) | 73.6 | 73.1 | 83.8 | 84.9 | 81.7 | 77.8 | 86.4 | 83.8 |
| e5-mistral-7b-instruct | 2 (74) | 69.2 | 68.3 | 83.0 | 80.9 | 79.4 | 75.6 | 83.6 | 81.1 |
| multilingual-e5-large-instruct | 3 (65) | 65.0 | 56.4 | 74.7 | 74.7 | 71.7 | 71.6 | 79.1 | 74.9 |
| multilingual-e5-large | 4 (63) | 61.7 | 46.8 | 73.4 | 72.2 | 66.6 | 69.1 | 75.7 | 73.4 |
| multilingual-e5-base | 5 (55) | 57.5 | 48.9 | 73.2 | 71.0 | 66.1 | 67.8 | 75.2 | 72.7 |
| multilingual-e5-small | 6 (53) | 58.4 | 48.4 | 70.6 | 67.9 | 65.2 | 66.6 | 73.6 | 68.1 |
| all-mpnet-base-v2 | 7 (44) | 56.4 | 46.3 | 67.4 | 62.2 | 63.1 | 61.7 | 69.0 | 65.7 |
| all-MiniLM-L6-v2 | 8 (34) | 52.7 | 48.1 | 64.4 | 57.4 | 62.2 | 60.4 | 68.1 | 66.6 |
| all-MiniLM-L12-v2 | 9 (27) | 50.2 | 46.8 | 68.1 | 57.3 | 63.6 | 62.7 | 68.7 | 67.8 |
| LaBSE | 10 (11) | 28.8 | 27.6 | 40.6 | 36.6 | 42.3 | 34.8 | 43.9 | 42.2 |

Table 19: Performance on MTEB(Code) across task categories. Because all code-related tasks are for retrieval, metrics by category are omitted.

