# OpenReview forum: "MMTEB: Massive Multilingual Text Embedding Benchmark"
_ICLR.cc/2025/Conference — ICLR 2025 Poster_

### Official Review · Reviewer_RchW · 2024-11-01

**Soundness:** 3
**Presentation:** 2
**Contribution:** 3
**Rating:** 6
**Confidence:** 5

**Summary:**

The paper introduces MMTEB, an extensive evaluation suite designed to assess text embedding models across over 1,000 languages and 500 tasks, serving as a multilingual extension to previous benchmarks like MTEB. MMTEB includes novel task categories, such as instruction following, long-document retrieval, and code retrieval. A significant contribution of MMTEB is its introduction of computational optimizations, including downsampling and hard negative sampling, which reduce compute requirements and enhance accessibility. The authors' findings reveal that smaller, instruction-tuned multilingual models outperform larger monolingual models in low-resource language settings.

**Strengths:**

(+++) MMTEB exemplifies a remarkable community-driven effort, engaging diverse contributors and fostering inclusivity.

(+++) Introduces computational optimizations like downsampling and hard negative sampling, reducing evaluation costs to 3.11 hours on a 7B model (H100 GPU), making it accessible to low-resource settings.

(++) Covers over 500 tasks across 10 categories in more than 1,000 languages, with a strong focus on low-resource languages and domains. But it lacks enough justification demonstrating the quality and value of each dataset. Provides an open-source, public leaderboard that encourages continuous contributions to advancing multilingual embedding research.

(+) Expands traditional benchmarks by including new task types like instruction following, long-document retrieval.

**Weaknesses:**

1. The study lacks a clear articulation of the specific knowledge gap that MMTEB addresses beyond what MTEB has already achieved in evaluating multi-task capabilities of embedding models. The results section suggests that multilingual scores are closely aligned with English, making it difficult to discern the unique value that MMTEB offers. Additional analysis could better exploit the benchmark's value and clarify its unique contributions.

2. While MMTEB aims to include as many relevant datasets as possible, it is unclear how these datasets were constructed or validated. Details on dataset quality, annotation methods (e.g., human vs. model-generated), and statistics (e.g., query-document ratios) would enhance transparency and reliability, especially given some datasets may be model-generated, such as FollowIR.

3. The paper mentions retaining the top 250 ranked documents per query for each dataset and model but does not specify which model(s) were used to select these hard negatives. Clarifying this would help assess the robustness of the benchmark's retrieval tasks.

4. The combination of 132 tasks makes it challenging to interpret a model's performance on specific languages or language families. While geopolitical categorization is helpful, further segmentation by language, domain, or specific capabilities could provide a more systematic and granular view of model performance. Expanding on the existing MTEB language families in Appendix H could offer researchers a clearer understanding of model weaknesses by language or domain.

**Questions:**

1. Should Section 3 be titled “Experimental Settings” instead of “Results” to better reflect its content?
2. The evaluation metrics and the main metrics for certain tasks are not described, e.g. instruction retrieval, reranking, multi-label classification.
3. Should bitext mining and STS be considered closely-related task categories in Figure 1?
4. Summarization showed minimal correlation with embedding performance in MTEB. If it is still included in MMTEB, what justifies its inclusion?
5. Does MMTEB include programming language benchmarks, such as CoIR? Additionally, what criteria of multilingualism are used to determine inclusion in the study?

---

> ### Author Response · Authors · 2024-11-20
>
> We thank the reviewer for constructive review and answer specific points below:
>
> > The study lacks a clear articulation of the specific knowledge gap that MMTEB addresses beyond what MTEB [...]
>
> Figure 3a compares multilingual performance of MTEB and MMTEB and we see that there is indeed a difference in rank even among multilingual models notably with 7B mistral-based models being outperformed by notably smaller XLM-R based multilingual-e5-large-instruct. We additionally examine this difference further in the newly added Figure 4 where we see that this difference is notably pronounced among low-resource languages.
>
> > While MMTEB aims to include as many relevant datasets as possible, it is unclear how these datasets were constructed or validated. Details on dataset quality, annotation methods (e.g., human vs. model-generated), and statistics (e.g., query-document ratios) would enhance transparency and reliability, especially given some datasets may be model-generated, such as FollowIR.
>
> We will supply descriptive statistics in Table 9-13 including number of samples (queries and documents). These are already computed within the package and accessibility as a part of the task metadata. In the metadata, we also include the Annotation creators, text creation, bibliographic reference, etc.. We have updated Table 5 with this information.
>
> > The paper mentions retaining the top 250 ranked documents per query for each dataset and model but does not specify which model(s) [...]
>
> Thank you for pointing this out! We describe more details in Appendix C.1.2 but agree this is missing, especially in the main text. To gather a wide set of hard negatives, we use lexical and neural retrieval, including SOTA ones: BM25, E5-multilingual-large (the best BERT-large sized multilingual model at the time), and E5-Mistral-Instruct 7B (due to being Mistral based and strong in many domains). This provides hard negatives at many ranges of model sizes and types. We have much more analysis justifying these choices in Appendix C.1.2 (i.e. due to strong results with even just one hard negative model on 250 documents).
>
> >  [...] While geopolitical categorization is helpful, further segmentation by language, domain, or specific capabilities could provide a more systematic and granular view of model performance. [...]
>
> We completely agree that many users will be interested in an in-depth analysis by task or by language. We have thus created a new interactive leaderboard that allows breaking down and filtering the benchmark according to task categories, languages, domains and by individual task. In addition we will add an analysis on performance by language by improving figure 9 and moving it to the main text.
>
> > Should Section 3 be titled “Experimental Settings” instead of “Results” to better reflect its content?
>
> We agree and have changed the title to Experimental Settings
>
> > The evaluation metrics and the main metrics for certain tasks are not described [...]
>
> The main metric for multi-label classification is Accuracy and for reranking it is MAP@1000. For instruction retrieval, it is Robustness@10 ([1]). We have included this in the paper as well.  However, the main metric can be task specific e.g. if a specific paper introduces a task with a given metric we ensure that the default metric remains the same (there are only few such examples). The main metric for each task is denoted in its metadata, but all metrics are computed and reported in the result objects.
>
>
> > Summarization showed minimal correlation with embedding performance in MTEB. If it is still included in MMTEB, what justifies its inclusion?
>
> SummEval is not included in mteb(multilingual), but it is included in mteb(eng), however we use “SummEvalSummarization.v2” as a bug was found in “SummEval”. The bug was in the computation of the mean score, where it included both the correlation coefficient and the p-value in the mean score. The performance on "SummEvalSummarization.v2" is as follows:
>
> - intfloat/multilingual-e5-small: 0.306
> - intfloat/multilingual-e5-base: 0.271
> - intfloat/multilingual-e5-large: 0.31
> - intfloat/e5-small-v2: 0.326
> - intfloat/e5-large-v2 0.3453
>
> To maintain comparability with existing MTEB results we do not fix this bug in the original benchmark (denoted mteb(classic)), but do issue a deprecation warning.
>
> > Does MMTEB include programming language benchmarks, such as CoIR? Additionally, what criteria of multilingualism are used to determine inclusion in the study?
>
> In the multilingual benchmark (MTEB(multilingual)) we specifically exclude programming languages. However, the released package does include the tasks from CoIR to allow construction of new benchmarks and CoIR is included in the public leaderboard.The benchmark MTEB(code) includes tasks from CoIR.
>
> ### References
>
> [1] “INSTRUCTIR: A Benchmark for Instruction Following of Information Retrieval Models” by Oh et. al. 2024

---

> > ### Comment · Reviewer_RchW · 2024-11-30
> > **Thanks for the response**
> >
> > Most of my concerns have been addressed, and I fully recognize MMTEB as a significant contribution to the embedding research community. However, as a research paper submission, I would expect more solid findings that reveal the benchmark's value and offer guidance for future research directions.
> >
> > There are a few follow-up questions I hope you can elaborate on:
> > 1. I find **the distinction between MMTEB and MTEB somewhat unclear**. Since the original MTEB already includes multilingual variants, it would be helpful to clarify the scope difference between the two in the manuscript to avoid confusion for future users. For example, in Section 2.4, the text states: "From the extensive collection of tasks in MMTEB, we developed several representative benchmarks, including a highly multilingual benchmark, MTEB(multilingual)." Why is this referred to as MTEB(multilingual) instead of MMTEB(multilingual)?
> >
> > 2. MMTEB includes 550 tasks and you have discussed four variants of MMTEB (multilingual, europe, indic, english). For the purpose of building the leaderboard, which of these settings will be considered the primary one for future model comparison? Additionally, is the multilingual variant a superset of the others, or do the subsets have distinct tasks that do not overlap?
> >
> > 3. Thank you for the clarification regarding the SummEval bug. Just to confirm, SummEval is not included in MTEB(multilingual), but it is still part of MMTEB (in English and French, as indicated in Table 8), correct? Further confirmation of this would be appreciated.

---

> > > ### Author Response · Authors · 2024-12-02
> > >
> > > Sure, we can elaborate further on these points:
> > >
> > > 1) While MTEB indeed contains multiple languages it only does so for a limited set of tasks (mostly classification and bitext mining), with the classification datasets stemming from translations. MMTEB notably expands MTEB to multiple languages across almost a much wider array of task categories. We refer to this collection as MMTEB. From this collection, multiple datasets can be constructed. We refer to a benchmark as a Massive text embedding benchmark (MTEB) and denote the target of the benchmark in parenthesis mteb(multilingual). This makes the leaderboard more transparent for new users as shorthand like MMTEB does not immediately tell the user what the benchmark targets. We welcome alternative naming approaches.
> > >
> > > 2) It would depend on the use case. If a fully multilingual model is developed we recommend using MTEB(multilingual), but users might want to target a narrower use case. The leaderboard even allow subselection datasets e.g. to construct a germanic benchmark from the current European or only examine retrieval within MTEB(multilingual). We welcome submissions for new such datasets. On the leaderboard we will default will show the multilingual or the English benchmark, but we have discussed creating an overview landing page that shows aggregate scores for a selected number of benchmarks.
> > >
> > > 3) SummEval is not a part of MTEB(multilingual) as the bug wasn't resolved when we finalized the benchmark. MTEB(classic) does contain SummEval and the French benchmark does contain a translated version SummEvalFr. We do not change these to ensure backward compatibility, but future versions of the dataset does contain the benchmark. MTEB(eng) uses SummEvalSummarization.v2.

---

> > > > ### Comment · Reviewer_RchW · 2024-12-02
> > > >
> > > > Thank you for your further clarification. I still find the naming distinction between MMTEB and MTEB somewhat confusing. Perhaps it would be clearer to reserve MTEB for English-only tasks and use MMTEB to encompass all multilingual scopes. Based on the discussion, I have adjusted my rating to 6. Thanks!

---

### Official Review · Reviewer_jHr7 · 2024-11-03

**Soundness:** 4
**Presentation:** 3
**Contribution:** 4
**Rating:** 8
**Confidence:** 4

**Summary:**

This paper introduces MMTEB, a massive multilingual text embedding benchmark that covers over 500 tasks in more than 1,000 languages. Compared to previous benchmarks, MMTEB considers the “low-resource double bind” during its construction and significantly reduces the computational resources needed for evaluation through various strategies while preserving the relative ranking of models.

**Strengths:**

1. I believe the efforts to reduce the computational resources required for evaluation are very meaningful, as they will encourage more researchers from low-resource language regions to use this benchmark. If MMTEB had simply expanded the scale of MTEB, it could be expected that most strong baseline models would originate from commercial companies with high computational resources, which could hinder the rapid development of text embedding research.
2. Each computational resource optimization strategy is described in detail, and the methods are easy to implement, which facilitates the adaptation of custom datasets.

**Weaknesses:**

1. The depth of analysis across different datasets seems inconsistent. For instance, the “Clustering” section in 2.3.1 provides an average Spearman correlation, but the “Retrieval” and “Bitext Mining” sections lack similar metrics. Moreover, as seen from the results in Appendix C.1.2, the selection of “Retrieval” strategy is based on analyses from only the NQ and TREC-COVID datasets, which may lead to biased hyperparameter selection. Although the current level of detail is already quite high, given MMTEB’s potential impact, I believe further detail would only be beneficial.
2. The abstract mentions “a diverse set of challenging, novel tasks such as instruction following, long-document retrieval, and code retrieval,” but I saw little content related to these datasets in the paper. I think the authors should clearly explain:
-  **Why were these tasks included in MMTEB?** (This is a benchmark for multilingual text embeddings, yet instruction retrieval is currently available only in one language.)
- **How were these new tasks integrated into the benchmark?** (I believe directly including long-document retrieval under the “retrieval” category should be done with caution, as it would require researchers to consider incorporating long-document-related datasets in their training data, which to some extent runs counter to the goal of addressing the “low-resource double bind.”)
- **How will these new tasks impact model performance?** (The context length limitation of models such as multilingual-e5-large-instruct could hinder their performance on tasks like long-document retrieval. In the LongEmbed evaluations [1], it performs worse than the Mistral-based version. Additionally, the results in Table 16 show that the Mistral-based models perform better on MTEB (code). Thus, claiming the exceptional performance of multilingual-e5-large-instruct in the Introduction without further clarification may mislead readers.)

[1] LongEmbed: Extending Embedding Models for Long Context Retrieval. arXiv:2404.12096

**Questions:**

1. In Lines 86 to 93, could you provide a more intuitive metric for comparing computational resources of different benchmarks, such as the time required to complete evaluations on a single A100 GPU?

---

> ### Author Response · Authors · 2024-11-20
>
> > The depth of analysis across different datasets seems inconsistent. For instance, the “Clustering” section in 2.3.1 provides an average Spearman correlation, but the “Retrieval” and “Bitext Mining” sections lack similar metrics. Moreover, as seen from the results in Appendix C.1.2, the selection of “Retrieval” strategy is based on analyses from only the NQ and TREC-COVID datasets, which may lead to biased hyperparameter selection. Although the current level of detail is already quite high, given MMTEB’s potential impact, I believe further detail would only be beneficial.
>
> It is indeed correct that we report differently for the types of speedups. For instance, we do not report a correlation metric for the bitext tasks as the speedup is mainly on reducing the number of times the same document is embedded thereby not influencing the performance score.
> We chose only two datasets for retrieval due to the computational expense of running such a large sweep of models on many different variants on the task (6 models, including 7B sized models, on 7 size variants, equivalent to 42 different runs per dataset). NQ and TREC-COVID also showcase the two extreme ends of relevance annotations (one per query, hundreds per query), and other datasets would likely fall within this range.
>
> > Why were these tasks included in MMTEB? (This is a benchmark for multilingual text embeddings, yet instruction retrieval is currently available only in one language.)
>
> We agree that coverage is not the same across the languages. Aspects such as instruction-following and long-document retrieval are important to test for and even though only available in one or few languages we believe that they are still valuable within multilingual benchmarks.
> However, since the submission the multilingual instruction following dataset mFollowIR (not yet published) have been submitted to the package. We will include it in the next iteration of the multilingual leaderboard. We see this benchmark as a continually developed leaderboard and plan to intermittently release updates to it in a versioned fashion to ensure reproducibility.
>
> > How were these new tasks integrated into the benchmark? (I believe directly including long-document retrieval under the “retrieval” category should be done with caution, as it would require researchers to consider incorporating long-document-related datasets in their training data, which to some extent runs counter to the goal of addressing the “low-resource double bind.”)
>
> As it stands each task is weighted independently, except in the “average across category”, where they are weighted such that each task category received similar weight. LEMBPasskeyRetrieval (a long document retrieval task) is indeed included within retrieval, however other tasks do as well include long documents. Would a solution to this concern be implementing a plot showing performance by average document length in a task?
> We believe that a benchmark should encourage directions of development (while maintaining accessibility). Thus including long-document tasks we argue will incentivize long docs embeddings when developing multilingual embedding models.
>
> > How will these new tasks impact model performance? [...]
>
> We agree with the reviewer that we are currently overstating the performance of the multilingual-e5-large-instruct and we will make sure to clarify cases where we see the performance discrepancy:.
> “we develop several highly multilingual benchmarks, which we use to evaluate a representative set of models. We find that while large language models (LLMs) with billions of parameters can achieve state-of-the-art performance on certain language subsets and task categories, the best-performing publicly available model is multilingual-e5-large-instruct with only 560 million parameters.”
>
> and
>
> “However, GritLM still remains best in class for retrieval on mteb(multilingual) and outperforms the multilingual-e5-large-instruct on mteb(code) and mteb(eng).”

---

### Official Review · Reviewer_CXhF · 2024-11-04

**Soundness:** 3
**Presentation:** 2
**Contribution:** 3
**Rating:** 6
**Confidence:** 3

**Summary:**

MMTEB addresses the limitations of traditional text embedding evaluations to extend the current popular MTEB benchmark to over 500 quality-controlled tasks across thousand languages, making it the largest multilingual collection for embedding model evaluation. MMTEB is a large-scale, open collaboration benchmark where the contributors have diverse backgrounds and introduce diverse and challenging tasks, such as instruction following, long-document retrieval, and code retrieval.

**Strengths:**

- Given that recent embedding models often shows the trends for optimized for MTEB benchmark tasks, it would be valuable to develop larger-scale benchmarks that include a broader range of tasks.
- Additionally, MMTEB downscales datasets and caches embeddings to help alleviate computational bottlenecks during evaluation.

**Weaknesses:**

In general, dataset information (such as sample numbers, multilingual types, etc) and relevant model benchmark numbers are missing. Find more details in questions.

**Questions:**

- How does the evaluation score compare with other embedding benchmarks? For instance, since the AIR-Benchmark doesn’t disclose its evaluation sets, does this mean there might be no overlap with MMTEB?
- Why do smaller models perform better in multilingual contexts while larger models excel on English datasets? Is this pattern unique to comparisons involving only the e5 models?
- Could you provide more MMTEB benchmark results using LLM embedding models from literature, such as SFR-Embeddings, NV-Embed, bge, and Qwen models? Since some of these models are English-based, please include their results in Table 15.
- In Table 9, could you provide the sample counts (queries and documents) for each task? Additionally, please list the 1000 languages and 500 quality-controlled evaluation tasks with examples.
- What is the sample count for each language, and is there an imbalance in sample numbers between languages? Why is it necessary to collect samples exhaustively from native speakers? Could machine translation help address sample imbalance?
- Could you provide contributor statistics, such as distribution across countries, native speakers, domains, and similar tasks?
- Since MMTEB appears to cover most of existing public embedding evaluation datasets, has there been any further data collection, annotation, or synthetic dataset creation for MMTEB? If so, please provide details.
- Paper does not properly explain the code and long-document benchmarks. Could you provide details on these benchmarks and the performance numbers for models from the literature?

---

> ### Author Response · Authors · 2024-11-20
>
> We thank the reviewer for their review and respond to the questions below:
>
> > In general, dataset information (such as sample numbers, multilingual types, etc) [...]
>
> We are not sure what is meant by multilingual types, but Tables 9-14 display languages for each dataset along with the domain and types. Additionally, each dataset is annotated with task metadata, where we have added descriptive statistics such as number of samples.
>
> > How does the evaluation score compare with other embedding benchmarks? [...]
>
> We compare our development benchmark with MTEB (denoted MTEB(classic) in the paper) in Figure 3a. We see that they do provide noticeably different results most notably among the mistral-based models (e5-mistral, gritlm) and the notably smaller XLM-R-based e5-multilingual-large-instruct.
> We additionally include retrieval datasets such as MIRACL which might be considered a benchmark in and of itself. MIRACL is included within MMTEB.
>
> We are unsure what is meant by overlap here. Since AIR does include sources such as Wikipedia and Arxiv which is also included in MMTEB. AIR does not cover a similar array of task categories and languages.
>
> > Why do smaller models perform better in multilingual contexts [...]
>
> We believe that this “likely emerges due to differences in pre-training, with Mistral being predominantly pre-trained on English, while XLM-R targets 100 languages.” (p. 6 line 309-) This includes both the e5 mistral but also the mistral-based GritLM. However, we see no reason why a large model (e.g. >7B parameters) given a highly multilingual pre-training could not outperform existing XLM-R-based models.
>
> > Could you provide more MMTEB benchmark results [...]
>
> We have started running the suggested models along with additional models (see response to reviewer PWdB) on all of the newly proposed benchmarks (including MTEB(eng)). We will include these in the public leaderboard.
>
> > In Table 9, could you provide the sample counts (queries and documents) for each task? [...]
>
> We agree with the reviewer that more information both on the tasks and languages. To facilitate this we have added D2 which provides a list of the 100 languages with the highest number of tasks in each category. We have additionally implemented functionality in the package to fetch descriptive statistics including the number of samples, average document length, etc. to allow users to easily inspect datasets. We have also updated Tables 9-13 with metadata including text creation, and number of samples.
> We agree that making datasets easily inspectable is required and have thus added Appendix D1 as well as added HuggingFace dataset links to the public leaderboard to allow for inspection.
>
>
> > What is the sample count for each language, and is there an imbalance in sample numbers between languages?
>
> There is indeed an imbalance between the number tasks pr. languages we show in Figure 5 and the newly added appendix D2, where it is shown that languages like “nya”, “pes” are represented by <10 tasks.
>
> > [...] Could machine translation help address sample imbalance?
>
> We attempt to avoid adding MT datasets as it is unclear if these would represent cultural contexts accurately and to mitigate the risk of artificially inflating multilingual model performance scores as these models are often trained on translated datasets. There is evidence that translationese, and, most importantly, machine-translationese substantially differ from natural language produced by native speakers and each other in multiple aspects (Bizzoni et al. 2020).
>
> > Could you provide contributor statistics, [...]
>
> We agree with the reviewer that sharing contributor metadata including native language would have been ideal. In Table 4 (anonymized) we provide the contributors' affiliation, but we do not have metadata on contributors' native language or country of origin, though contributors know who to contact if they are e.g. looking for a contributor proficient in French based on previous commits or assistance from other contributors.
>
> > [...] has there been any further data collection, annotation, or synthetic dataset creation for MMTEB?
>
> Many of the datasets used for MMTEB were already published allowing us to utilize validated datasets. However, the datasets require reformatting to fit within a unified framework, we describe this in section B2. This practice is common among widely used and highly influential benchmarks such as (glue (Wang et al., 2018), superglue (Wang et al., 2019)).
>
> We also introduce five novel datasets we extend B.3, with a description of these datasets.
>
> > Paper does not properly explain the code and long-document benchmarks [...]
>
> We agree that this is unclear and have added the descriptions to clarify e.g.:
>
> “MMTEB also integrates domain-specific benchmarks like CoIR for code retrieval and LongEmbed for long document retrieval.”
>
> We also include a manually curated benchmark mteb(code) on which more information can be found in H2 and H4.

---

> > ### Author Response · Authors · 2024-11-20
> > **References**
> >
> > - Wang, A., Singh, A., Michael, J., Hill, F., Levy, O., & Bowman, S.R. (2018). GLUE: A Multi-Task Benchmark and Analysis Platform for Natural Language Understanding. BlackboxNLP@EMNLP.
> > - Wang, A., Pruksachatkun, Y., Nangia, N., Singh, A., Michael, J., Hill, F., Levy, O., & Bowman, S.R. (2019). SuperGLUE: A Stickier Benchmark for General-Purpose Language Understanding Systems. ArXiv, abs/1905.00537.
> > - Yuri Bizzoni, Tom S Juzek, Cristina España-Bonet, Koel Dutta Chowdhury, Josef van Genabith, and Elke Teich. 2020. How Human is Machine Translationese? Comparing Human and Machine Translations of Text and Speech. In Proceedings of the 17th International Conference on Spoken Language Translation, pages 280–290, Online. Association for Computational Linguistics.

---

> > > ### Comment · Reviewer_CXhF · 2024-11-27
> > >
> > > Thanks for the detailed clarifications and response. I will keep my score.

---

### Official Review · Reviewer_PWdB · 2024-11-04

**Soundness:** 3
**Presentation:** 3
**Contribution:** 4
**Rating:** 8
**Confidence:** 4

**Summary:**

This paper introduces a new set of benchmarks called "massive multilingual text embedding benchmark". This benchmark includes more than 500 tasks and covers a lot of low resource languages as well. They also introduce downsampling technique such that the resources required for the evaluation is minimized.

**Strengths:**

- The paper is well written and the main points are clearly communicated
- The dataset is a great extension to the MTEB and would be a good resource to research community towards building largescale multilingual embedding models
 - The coverage of the dataset is great

**Weaknesses:**

Based on table9, one limitation is that most of the crowd submissions are already based on existing public datasets from multiple language domains and not particularly for this dataset construction effort.

**Questions:**

- How do the top models on MTEB leaderboard do on this new dataset and whether this new dataset changes the ranking of the leaderboard?

---

> ### Author Response · Authors · 2024-11-20
>
> We thank the reviewer for the kind words and review. Here are some responses to the points raised:
>
> > Based on table 9, one limitation is that most of the crowd submissions are already based on existing public datasets from multiple language domains and not particularly for this dataset construction effort.
>
> While it is indeed correct that many of the datasets were publicly available. This allows us to utilize existing high-quality validated datasets for novel evaluation. An expansive evaluation such as the one presented would not have been possible had we started from scratch. This practice is common among widely used and highly influential benchmarks such as (glue (Wang et al., 2018), and superglue (Wang et al., 2019)).
>
> However, we introduce 5 novel datasets, WebLINXReranking, PublicHealthQA, WikiClustering, WikipediaRetrievalMultilingual, and WikipediaRerankingMultilingual of which the last two utilize LLM-based synthetic queries. We show in Figure 4 that these approximate existing Wikipedia-based retrieval datasets such as GermanQuAD. We have extended the description of these datasets in Appendix B3
>
> > How do the top models on MTEB leaderboard do on this new dataset and whether this new dataset changes the ranking of the leaderboard?
> In figure 2a we compare existing models’ performance on MTEB(classic) and MTEB(multilingual) of a representative sample of multilingual models along with a few monolingual models. We see that they do provide noticeably different results most notably among the mistral-based models (e5-mistral, GritLM) and the notably smaller XLM-R based e5-multilingual-large-instruct.
>
> In addition to these, we have also started running:
>
> - gte-Qwen2-7B-instruct
> - BAAI/bge-large-en-v1.5
> - gritlm-8x7B
> - salesforce/SFR-Embeddings-2R
> - snowflake/arctic-embed-m-v1.5
> - WhereIsAI/UAE-Large-V1
> - stella_en_1.5B_v5
> - stella_en_400M_v5
> - openai/text-embedding-large
> - openai/text-embedding-small
> - intfloat/e5-base-v2
> - intfloat/e5-large-v2
> - intfloat/e5-small-v2
> - jina/jina-embeddings-v3
> - mixedbread-ai/mxbai-embed-v1
> - nomic-ai/nomic-embed-text-v1.5
> - NV-embed-v2
>
> And will include these in the public leaderboard.
>
> ### References
>
> - Wang, A., Singh, A., Michael, J., Hill, F., Levy, O., & Bowman, S.R. (2018). GLUE: A Multi-Task Benchmark and Analysis Platform for Natural Language Understanding. BlackboxNLP@EMNLP.
> - Wang, A., Pruksachatkun, Y., Nangia, N., Singh, A., Michael, J., Hill, F., Levy, O., & Bowman, S.R. (2019). SuperGLUE: A Stickier Benchmark for General-Purpose Language Understanding Systems. ArXiv, abs/1905.00537.

---

### Meta-Review · Area_Chair_VVAN · 2024-12-20

**Metareview:**

This paper introduces the Massive Multilingual Text Embedding Benchmark (MMTEB), a comprehensive initiative that expands the existing MTEB to include over 500 quality-controlled evaluation tasks across more than 1,000 languages, offering the largest multilingual collection of tasks for text embedding models. The study reveals important findings on the performance of LLMs across various tasks and languages, highlighting the efficiency of the notably smaller multilingual model, e5-large-instruct, across multiple languages. The reviewers are unanimous in appreciating the depth of the evaluation, the methodological innovations for reducing computational demands, and the benchmark's potential impact on the field, leading to a collective recommendation for accepting the paper.

**Additional Comments On Reviewer Discussion:**

Nil

---

### Decision · Program_Chairs · 2025-01-22

Accept (Poster)